# Changes in semantic memory structure support successful problem-solving and analogical transfer
Théophile Bieth [1,2] ✉, Yoed N. Kenett [3], Marcela Ovando-Tellez [1], Alizée Lopez-Persem [1], Célia Lacaux [1,4], Marie Scuccimarra[1], Inès Maye[1], Jade Sénéchal[1], Delphine Oudiette [1,4,5] ✉ & Emmanuelle Volle [1,5] ✉

Creative problem-solving is central in daily life, yet its underlying mechanisms remain elusive. Restructuring (i.e., reorganization of problem-related representations) is considered one problem-solving mechanism and may lead to an abstract problem-related representation facilitating the solving of analogous problems. Here, we used network science methodology to estimate participants' semantic memory networks (SemNets) before and after attempting to solve a riddle. Restructuring was quantified as the difference in SemNets metrics between pre- and post-solving phases. Our results provide initial evidence that problem-related SemNets restructuring may be associated with the successful solving of the riddle and, subsequently, an analogous one. Solution-relevant concepts and semantically remote concepts became more strongly related in solvers. Only changes in semantically remote concepts were instrumental in actively solving the riddle while changes in solution-relevant concepts may reflect a pre-exposure to the solution.

In our daily life, we constantly deal with problems, ranging from the most mundane (e.g., what to cook for dinner given the ingredients at our disposal), to professional activities (e.g., how to reorganize our current plans to meet a new deadline), up to major societal challenges (e.g., how to find innovative solutions against global warming). How do we find new solutions to problems? While the ability to solve problems is a critical skill for adapting to new situations and innovating, the mechanisms underlying the problem-solving process remain largely unknown.

Among the new problems we face each day, some are well-defined (e.g., playing a jigsaw puzzle). The initial state (i.e., the number of independent pieces) and goal state (i.e., assembling the pieces so it looks like the picture model) are clear, and the solver can apply a set of operations (i.e., interlocking the pieces as a function of their shape) to reach the goal. However, for many of our problems (e.g., organizing work activities during the COVID-19 pandemic), the problem space is ambiguous. No heuristics or existing rules could be applied to transform the initial state into the goal state[1]. Such "ill-defined" problems[2] thus require additional mental processes, which have been tightly linked to creative thinking[3–5]. Ill-defined problem-solving (or creative problem-solving) is often referred to as insight solving, where the solution comes to mind suddenly and effortlessly, with a

"Eureka" phenomenon[6–9]. According to the Representational Change Theory[10], solving such problems involves restructuring the initial problem mental representational space[5,9], which presumably entails combining elements related to the problem in a new way. In theory, restructuring allows one to change perspective, reframe the problem, or escape its implicitly imposed constraints[11], leading to creative associations[6,9]. For instance, consider the following problem: "*A man walks into a bar and asks for a glass of water. The bartender points a shotgun at the man. The man says, 'Thank you,' and walks out*"[12]. The problem is ill-defined because the path to finding the solution is to be discovered, and the goal state is vague. Solving this problem first requires asking the right question: in which context would a *shotgun* and a *glass of water* help somebody? Rather than relying on obvious associations (e.g., *a glass of water* is related to thirst), solvers must fill the missing link between the relevant elements of the problem (a *shotgun* induces fear, and fear can be a remedy for hiccups, as can drinking a *glass of water*). Hence, restructuring the initial representation of a given problem would allow one to see this link and find its solution.

A separate field of research suggests that such reorganization of mental representations could be useful, not only for solving a given problem, but also for solving future, different problems that share some structural

[1]Sorbonne University, Institut du Cerveau - Paris Brain Institute -ICM-, Inserm, CNRS, AP-HP Hôpital de la Pitié-Salpêtrière, DMU Neuroscience, Paris, France. [2]Neurology department, Pitié-Salpêtrière hospital, AP-HP, F-75013 Paris, France. [3]Faculty of Data and Decision Sciences, Technion – Israel Institute of Technology, Haifa 3200003, Israel. [4]Sleep center, Pitié-Salpêtrière hospital, AP-HP, F-75013 Paris, France. [5]These authors contributed equally: Delphine Oudiette, Emmanuelle Volle. ✉e-mail: theo_bieth@hotmail.fr; delphine.oudiette@gmail.com; emmavolle@gmail.com

similarities with the one presented (i.e., an analogous problem). Extracting an abstract schema from an initial problem is hypothesized to be the core process of analogical reasoning and transfer[13–17], which are cognitive abilities usually associated with problem-solving[18–23]. Several studies proposed that restructuring was key in extracting such an abstract representation of the problem[19,24–29]. Alternatively, restructuring, which is supposed to combine remote concepts in memory, may facilitate the detection of structural links between a problem and its analogous. Hence, higher abilities to form remote associations may be more conducive to constructing a broader representation, that, in turn, would facilitate solving an analogous problem[19].

Overall, although the restructuring hypothesis for problem-solving is intuitively satisfying, and the reorganization of mental representation into an abstract schema is central in analogical reasoning theories, empirical evidence supporting these hypotheses is surprisingly scarce[9,12,30–33]. The lack of existing measures to assess such restructuring may explain this gap in the literature. For instance, in insight problem-solving studies, the assumption that solvers restructured their internal representation of the problem often stems from the mere fact that they found the solutions (circular argument)[34]. This assumption came from theories of insight problem-solving that highlighted restructuring as a prerequisite for insight solving[10,35–38] (although it is debated, see refs. 33,39,40). Alternatively, some studies have proposed a measure of restructuring using individual subjective ratings during problem-solving. These ratings assessed how much problem elements (that could be related to the solution or not) were relevant to consider for solving the problem[30,33]. They showed that problem elements that were objectively relevant for solving the problem (compared to objectively non-relevant ones) were rated progressively as more relevant during the thinking time in solvers. Yet, these studies did not explore this effect in non-solvers. In addition, their restructuring measure reflected the relationship between problem elements and the solution without considering how problem elements became more related to each other. These limitations could be addressed using computational network science methods that have been established to represent semantic memory as a semantic memory network (SemNet).

Network science is based on mathematical graph theory, offering quantitative tools to represent complex systems as networks[41]. A growing amount of research has been applying network science methods to study cognitive systems, mainly focusing on memory and language[41,42]. Specifically, SemNets represent elements of knowledge as nodes related to each other with edges of various strengths[43–45]. They have been used to explore the organization of semantic memory[41,44,46,47], and how this organization relates to creativity[42,48–56]. In a theoretical paper, Schilling proposed that insight solving occurs as a result of a recombination of the SemNet associations through the creation of new or unexpected links between remote nodes in semantic memory[35]. This hypothesis echoes the concept of restructuring developed by the tenants of the Representational Change Theory[10], but extends it more broadly to the formation of any atypical association without necessitating the concepts to be directly relevant to solving the problem at hand. This idea aligns with current theories on the dynamic nature of semantic memory[57], but only scarce empirical studies tested it (but see ref. 58).

To date, only one study has related insight problem-solving to SemNets restructuring[12]. Durso et al. used the riddle presented above and SemNets to demonstrate a restructuring of the solution-relevant semantic associations in SemNets of participants who solved the riddle compared to those who did not[12]. However, SemNet is a dynamic system that can be modulated by context[57,58]. In the context of a problem-solving task, it is unclear whether SemNets organization changes reflect the awareness of the solution (e.g., processes akin to a priming effect induced by the solution), or active processes related to the mental restructuring of problem representation. In their study, Durso et al. showed that edges between nodes that are decisive to solving the riddle were gradually identified as critical before problem-solving, suggesting that restructuring occurred before solving the problem[12]. In addition, they found that SemNets of non-solvers to whom the solution was given showed more similarities with SemNets of non-solvers than with

those of solvers. This study pioneered the domain but had several limitations that we aim to address. First, although restructuring referred to a change in problem representation, the authors approximated restructuring using only one SemNet computed after attempting to solve a problem without considering a baseline reflecting the initial problem representation. Second, they explored SemNets at the group level (solvers vs. non-solvers), missing out on how SemNets change at the individual level. Third, restructuring was explored only based on the relevance of word associations for solving the problem. If Schilling's theory[35] is correct, restructuring related to problem-solving should also target semantically remote conceptual associations (not just the solution-relevant ones). Finally, no replication of this study has been published since 1994 and it is unclear whether this restructuring effect, shown using a single riddle, would be generalized to other problems.

Here, we aimed to explore restructuring as a cognitive mechanism underlying successful problem-solving by replicating and extending Durso et al. results[12]. We propose a method to quantify problem restructuring that measures the changes in the organization of problem-related concepts by building SemNets in solvers and non-solvers using tools from network science. We estimated individual SemNets at two different time points: once before the presentation of a problem, operationalized as a riddle (four different riddles, such as the one described above), and after an attempt to solve it. We assumed that measuring changes in the organization of problem-related concepts could capture restructuring and be used to investigate the relationship between restructuring and solving. In contrast to previous studies that used the SemNets method to relate semantic memory structure to creative behaviors at the inter-individual level[48,49,52,53,59–61], we propose here that the SemNets method can be used in a different way to describe problem-solving processes at the individual level. Thus, unlike in our previous studies that related global SemNets properties to creativity as a trait[52,53,61], here we built problem-related SemNets and analyzed their local changes depending on whether a given problem was solved, independently of whether individuals were inherently creative or not. In addition, we examined whether these SemNets changes also relate to analogy transfer, i.e., to solving an analogous problem. Importantly, we explored two types of SemNets changes: those that occurred in solution-relevant links (i.e., solution-relevant restructuring) and those bringing closer together initially remote problem-related word pairs (i.e., remoteness-based restructuring).

Based on previous studies., we initially hypothesized that changes in SemNets targeting solution-relevant words would predict its successful solving, as well as the successful solving of an analogous problem. Yet, in that case, we could not disambiguate whether these solution-based SemNets changes reflected a restructuring leading to the discovery of the solution (instrumental role), or whether they reflected the consequences of finding the solution (i.e., priming effect). Thus, we also tested an alternative hypothesis based on Schilling's theory, which assumes that solving an insight problem implies local SemNets changes, involving semantically remote words[35]. In this view, combining remote concepts would reflect an active restructuring process that is less influenced by solution-induced priming effect. Finally, to disentangle solution priming from solving-related restructuring, we ran an additional study in which the solution was given to the participants. We hypothesized that participants who are given the solution would show solution-induced SemNets changes but not remoteness-based changes.

## Methods

The three studies were not preregistered. Gender was determined based on information provided by participants. We did not collect information about ethnicity. Data distribution was assumed to be normal, but this was not formally tested (Figs. S1, S2, and S3).

### Participants

In Study 1, ninety-nine native French speakers aged between 18 and 38 years old (mean age = 26 years, standard error of the mean, SEM = 2.60 years; 69 women and 30 men) were included. All participants were healthy adults with no history of neurological and/or psychiatric illness and no

psychoactive substance abuse. This sample was a sample of convenience. An approved ethics committee approved the study (CPP Ile-de-France III, 2019-A00562-55). All the participants gave their written informed consent and received 10€/hour as monetary compensation. After preprocessing, we excluded 19 participants' data resulting in a final sample of 80 participants used in statistical analyses. Details about exclusions are available in each relevant section.

## Experimental procedure

Participants underwent a 4h-experiment during which they had to solve four riddles (labeled "Zoe", "Daniel", "Car", and "Bar" riddles) and underwent a relatedness judgment task (RJT) in order to build individual-based SemNets. The four riddles were made of two pairs of analogous riddles, i.e., the problem and its solution of one riddle of the pair shared a common global schema with the second one (Zoe/Car riddles and Bar/Daniel riddles).

Participants were asked to solve the two pairs of riddles, one at a time, according to the following procedure (Fig. 1A). They first attempted to solve one riddle of the pair (e.g., Zoe riddle) for 10-min. Before and after this solving phase, participants performed the RJT. Then, the solution to the riddle was provided to the participants, followed by an independent task. Finally, they attempt to solve the second riddle of the pair (i.e., Car riddle) for 4-min. After a 15-min break followed by various creativity tasks, participants were presented with the other pair of riddles (i.e., Bar and Daniel riddles) following the same procedure. The order of riddles within each pair was counterbalanced among participants. The order in which the riddles were presented to the participants defined the *naive* condition (i.e., first presented riddle, without an analogous one before) or the *transfer* condition (i.e., second presented riddle, preceded by an analogous one). In addition, the order of the pairs of riddles within the experimental design was counterbalanced among participants to ensure that neither positive (training) nor negative (fatigue) effects related to which riddle came first interfered

with the results (four possible combinations: Zoe/Car then Daniel/Bar, $n = 25$; Daniel/Bar then Zoe/Car, $n = 24$; Bar/Daniel then Car/Zoe, $n = 25$; and Car/Zoe then Bar/Daniel, $n = 25$).

All tasks were computed using the Psychopy software[62] running on individual computers in a classroom dedicated to cognitive experiments (https://prisme.institutducerveau-icm.org). Task-related instructions were initially explained and repeated before the beginning of each task.

## Problem-solving

To test whether problem-solving was related to a restructuring of semantic associations, we needed to construct problems that: (i) were verbal (to allow building SemNets from the RJT ratings with words that made sense to the problem); (ii) had a unique solution (to facilitate the classification of participants as solvers or non-solvers); (iii) were difficult enough (to be able to measure a change after a solving phase); (iv) were likely to necessitate creative thinking (to maximize our chances to observe a restructuring of semantic associations) and (v) could be used to build analogous problems. Problems requiring creative thinking are often ill-defined (i.e., no specific heuristics or usual rules could be used to solve the problem) and putatively prompt a usual representation that one needs to overcome in order to solve the problem. Riddles like the one found in ref. 12. satisfy all of those criteria. We thus selected the Durso riddle and translated it into French (the "Bar" riddle). As we could not find any other similar riddles in the scientific literature, we picked one relevant riddle displayed on different riddle websites and rated as tricky in a pilot study (the "Car" riddle). Finally, we created two novel riddles in such a way that the problem situation/solution was analogous to the "Bar" riddle (the "Daniel" riddle; in both riddles, a person with a threatening behavior is actually helpful) and to the "Car' riddle (the "Zoe" riddle; in both riddles, a person apparently involved in a real-life situation is actually playing a game). We ensured that initial and analogous riddles were in distinct semantic domains or contexts. The riddles in French and with their English translation are available in Tables S1 and S2. For

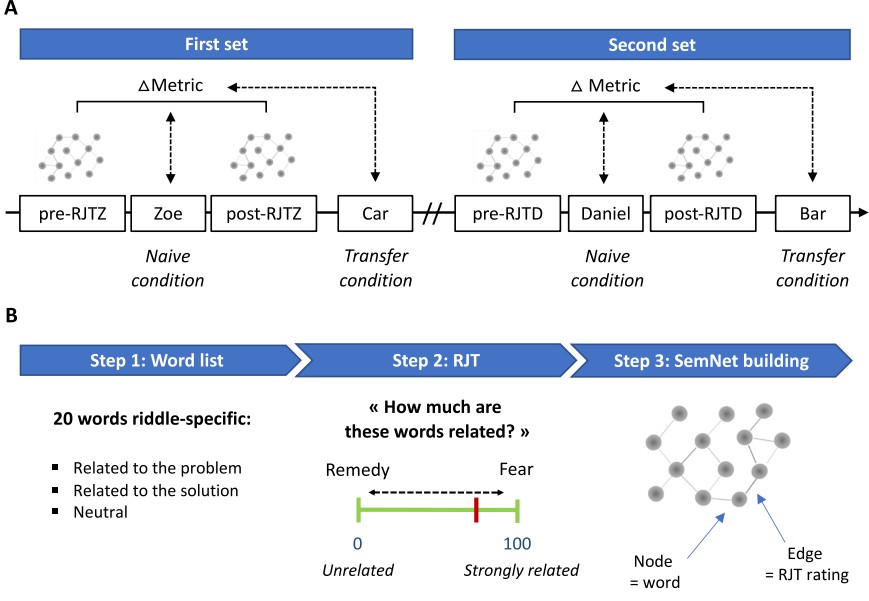

**Fig. 1 | Experimental procedure. A** Experimental design. Each participant had to sequentially solve two pairs of riddles (e.g., first set Zoe/Car then second set Daniel/Bar) separated by a 15-min break followed by various creativity tasks (not shown in the figure). Within each pair of riddles, the *naive* condition corresponded to the first presented riddle and the *transfer* condition to the second one. Each riddle-specific Relatedness Judgment Task (RJT) was performed before and after the solving phase related to the riddles in the *naive* condition. Pre- and post-RJT were similar for a given riddle (RJTZ referred to RJT specific to Zoe riddle, and RJTD to RJT specific to Daniel riddle). They were used to build individual semantic networks (SemNets) and assess changes in their organization during the solving phase (ΔMetric). The

ΔMetric was used to predict the successful problem-solving of a problem and an analogous one. **B** SemNets construction based on RJT. First (step 1), 20 words were specifically selected for each riddle and thus differed between riddles' RJTs. Then (step 2), participants had to rate the strength of relatedness (RJT) between two words on a visual scale from 0 (unrelated) to 100 (strongly related). For instance, they had to assess the degree to which the word 'remedy' is related to the word 'fear'. Word pairs composing riddle-specific RJT trials were all possible combinations of these 20 words ($n = 190$ pairs). Finally (step 3), individual SemNets were built using the word list and the RJT ratings. SemNets metrics were then computed for each individual, both in the pre and post conditions, for each riddle.

example, the Zoe riddle states the following: "*Zoe throws a stone that lands in the sky. How is it possible and in which context?*" and the Bar riddle is the one used by ref. 12 (described in the introduction).

After each riddle presentation, participants had up to 10 min in the *naive* condition and 4 min in the *transfer* condition to search for the solution. During this time, the riddle remained displayed at the center of the screen. Participants were instructed to report all ideas that came to mind, even if they judged them bizarre or irrelevant. They could press the space button anytime to propose a response but had to do it only if they knew what to write (to avoid getting additional thinking time). Pressing the space bar stopped the timer. Participants then had up to 30 s to write their ideas using the keyboard. The number of ideas a participant could propose was not limited, and no feedback on the response correctness was provided. For each response they gave, participants were first asked to indicate the confidence they had in their response on a visual scale from 0 ("not sure at all") to 100 ("completely sure"). Afterwards, a new screen displayed "Eureka?", and the participants indicated whether their response came to their mind with an insight or Eureka phenomenon by pressing a "yes" or "no" button[63]. Participants were told that a Eureka is "the subjective experience you may have when you solve a problem, and the solution comes to mind suddenly, is not the direct result of cognitive effort, and you are not able to report the mental steps leading to this solution". It was opposed to analytic solving where "you have a strategy and the feeling of gradually getting closer to the solution". We also told participants that these two solving methods were not exclusive and instructed them to consider only the few seconds before the idea came to their mind. Once participants replied to the Eureka question, the riddle was displayed again, and the timer restarted. Every two minutes, we probed participants' attentional focus by asking them what they were thinking about. Participants answered the question by choosing between four options (« Focused on the riddle », « Distracted by the environment », « Thoughts unrelated to the riddle », and « No thoughts ») using the keyboard (a predetermined numerical key corresponding to each option).

The solving phase of the riddle in the *transfer* condition followed the same experimental procedure. Participants were not informed of the relationship between the two analogous problems.

For each riddle, participants were assigned to either the solver or non-solver group, depending on their success in solving the corresponding riddle. We computed the solving rate for the *naive* condition (i.e., first presented riddle), and the *transfer* condition (i.e., second riddle presented). The solving rate corresponded to the percentage of participants who gave the correct solution anytime during the time allowed (i.e., the number of solvers divided by the number of participants who worked on this riddle), excluding participants who knew the riddle beforehand. Of note, responses of 12 participants who were already familiar with a riddle were excluded (Bar: $n = 5$ in *naive* condition and $n = 5$ in *transfer* condition; Car: $n = 1$ in *naive* condition and $n = 1$ in *transfer* condition).

## Relatedness judgments task – RJT

Participants' SemNets estimation was achieved via a computational method based on the RJT. In this task, participants rated the relatedness of all possible pairs of words. The RJT is based on previous research that showed how semantic distance in a semantic memory network corresponds to subjective relatedness ratings[64,65]. This method estimates an individual's SemNet structure based on relatedness judgments to all possible pairs of a set of cue words[48], and has been applied across different languages and cultures[48,52,59]. The RJT serves as a proxy of the organization of these words in an individual's SemNet. An n × n matrix is constructed, in which n represents the number of words used in the RJT, and each cell represents the relatedness rating given by the participant for these two words. This matrix represents a participant's individual SemNet.

We built verbal material for the RJT that was specific to each riddle and consisted of a list of 20 different words (four different lists for Zoe, Daniel, Car, and Bar riddles; see Table S3). The same list was used in the RJT performed before (pre-RJT) and after (post-RJT) working on a given riddle. To create those lists, we used a two-step procedure. First, three of the co-

authors, experts in creativity research (TB, DO, EV), independently proposed up to 30 words for each riddle (blind procedure). Each of them adopted the approach used by Durso et al.[12], consisting of selecting words that were explicitly stated in the problem (e.g., *bar* and *shotgun* in the Bar riddle), related to the solution (e.g., *remedy* and *relieve* in the Bar riddle), or usually associated with the problem but not with the solution (e.g., *drunk* and *loaded* in the Bar riddle). Words could be verbs, nouns, or adjectives. To ensure that selected words could be easily accessible in an individual's SemNet, we made sure that only frequent words were listed, with a lexical frequency higher than one million occurrences in the lexicon database[66] (http://www.lexique.org/). Second, the three experts shared their respective lists and reached a consensus for 20 words per riddle (including, on average, $4 \pm 0.8$ words explicitly related to the problem, $6 \pm 1$ words related to the solution, and $10 \pm 1$ words loosely associated with the problem but unrelated to the solution). During this selection process, we were careful to avoid words that were too closely related to the solution (e.g., *game* in the Zoe riddle), too strongly associated with other ones (e.g., *bar* and *barman* in the Bar riddle), or too distant from all of the other words within the same list. We retained 20 words (rather than 14 in the Durso et al. study[12]) to obtain a larger SemNet (larger graphs are ideal for finding subtle differences in global metrics). As a side note, for the Bar riddle, we used 10 out of the 14 words used in the Durso et al. study[12] along with ten novel words. We discarded four words (*paper bag*, *pretzel*, *man*, and *barman*) from Durso's original list based on pilot experiments, which showed that they were not related to the problem (*paper bag*) for French people or were isolated from the other words (*pretzel*), or were too strongly related to bar (*man* and *barman*) thus biasing the SemNets metrics.

During the RJT, participants were presented with all possible combinations of 20 word pairs ($n = 190$) based on a riddle-specific 20-word list. On each trial, a different word pair was displayed on the screen, and participants were asked to rate the strength of semantic association or relatedness between the two words on a visual scale from 0 ("unrelated") to 100 ("strongly related"), using a slider (Fig. 1B). The visual scale was displayed below the word pair on each trial and stayed on the screen until the participant responded. Participants had up to 4.5 s to respond using the computer mouse and validate their rating with a left click. The order of trials was initially pseudo-randomized and then fixed across subjects. We used the Mix software[67] to generate a pseudo-random order where each word appeared equally on the right and left sides of the screen and did not repeat in two consecutive trials. Before starting the task, participants were instructed to answer as quickly and spontaneously as possible and completed 25 practice trials. Before each first riddle-specific RJT, all the words that composed the pairs were successively displayed on the screen.

## Impact rating and semantic distance variables

To examine whether changes occurred in a specific part of participants' SemNets, we created two variables: the *impact rating* and the *semantic distance*.

The *impact rating* variable represented how much each edge and node were relevant in solving a riddle. We asked nine independent and external judges to rate how helpful or misleading the link between the words of each of the 190 pairs was in solving a given riddle. The judges scored each riddle separately. They first read the riddle and were given the solution. Then, they rated the importance of each word pair to solve the riddle using a visual scale ranging from −50 ("misleading") to 50 ("helpful"), centered on 0 ("neutral"). Each word pair was successively displayed on the screen above the visual scale in the same way as during the RJT procedure described above. There was no time limit to respond. By averaging the ratings across the judges and then z-scoring across the 190 word pairs, we obtained the *impact rating*, which quantifies the relevance of each word pair ($n = 190$) for solving the riddle. We also computed an indirect impact score determining the importance of each word to solve the riddle. This indirect score was calculated by averaging the *impact rating* of the pairs involving this word ($n = 19$ pairs per word) for each word. Then, we z-scored the obtained values across the 20 riddle-specific words. This procedure was repeated for each

riddle-specific word list. These *impact ratings* allowed us to weigh word pairs (edges) and words (nodes) according to their relevance for solving the problem: the higher the score, the more we considered the pair or the word helpful. The *impact rating* allowed us to explore a solution-based restructuring corresponding to SemNets changes toward an optimal representation that integrated the problem and its solution.

Intraclass coefficient (ICC) was higher than 0.80 for all riddles (Zoe: ICC = 0.93; Car: ICC = 0.84; Bar: ICC = 0.90; and Daniel: ICC = 0.94), suggesting good inter-judge reliability. Distributions of *impact rating* show that our riddle-specific RJT material captures word associations of varying degrees of relevance for solving the riddle (Figs. S2A and S3A).

The *semantic distance* variable represented how much two words were semantically distant in general, independently of the problem-solving context. We used the ratings collected during the first RJT (i.e., before the problem presentation) from all participants included in the three experiments. For each riddle, we computed a riddle-specific *semantic distance* corresponding to the median value of the participants' ratings for each word pair (Zoe: $n = 193$; Car: $n = 38$; Bar: 34; Daniel: 34 participants). We then z-scored the *semantic distance* across the 190 median values obtained. As we did for the *impact rating*, we also computed an indirect semantic score determining the semantic isolation of a word in relation to the others. This indirect score was calculated by averaging the *semantic distance* of the pairs involving this word ($n = 19$ pairs per word) for each word. Then, we z-scored the obtained values across the 20 riddle-specific words. This procedure was repeated for each riddle-specific word list. These *semantic distances* allowed us to weigh word pairs (edges) and words (nodes) according to the semantic remoteness: the higher the score, the more we considered a word pair semantically distant or a word semantically isolated. The *semantic distance* allowed us to explore a remoteness-based restructuring that corresponded to SemNets changes targeting problem-related associations that were semantically remote.

We provided the distribution of each riddle-specific *semantic distance* in Figs. S2A and S3A and the relationship between *impact rating* and *semantic distance* in Figs. S2B and S3B. We observed that solution-relevant edges or nodes were usually also semantically remote (and conversely), but the correlation was weak. Thus, some edges could be solution-relevant but semantically close (for instance *to write – ground* for the Zoe riddle, or *relieved – remedy* for the Bar riddle), and other edges could be solution-irrelevant but semantically distant (for instance *to fly – space* for the Zoe riddle, or *to die – bar* for the Bar riddle). This suggests that *impact rating* and *semantic distance* variables could capture different effects.

### Individual-based semantic memory networks

For each participant and each riddle independently, we built two SemNets based on RJT ratings collected before the presentation of the problem (Pre-SemNet) and after attempting to solve the problem (Post-SemNet) (Fig. 1A). These networks were represented as a $20 \times 20$ adjacency matrix (one word per row and column) containing all values of the individual RJT ratings. Based on previous studies using this approach[48,49,52,53,59,61], we applied a weighted undirected network method.

In these estimated weighted undirected networks, the relation between node *a* and node *b* is equal to the relation between node *b* and node *a* (i.e., symmetrical adjacency matrix), all edges are kept in the network, and edges are weighted based on the judgements (without any transformation) provided by each participant during the RJT[48]. The benefit of this methodology is that it avoids any arbitrary thresholding of edges for network filtering. This is critical for capturing the possible weaker connections of semantic relationships in lexicons[52,59].

For each SemNet (pre-RJT and post-RJT), we computed SemNets metrics (adapted for weighted undirected network method) that quantify the connectivity properties of a network using the Brain Connectivity Toolbox[68] (version 2019-03-03) running with Matlab R2020. We used a limited set of SemNets metrics, commonly used in cognitive research and previously linked to creative abilities as a trait[41,43,44,68–70]. We hypothesized that they could thus potentially also explain the creative process as well.

These metrics all quantified a different characteristic of SemNets (edge metric, node metric, centrality metric).

We computed several metrics for each node ($n = 20$) and edge ($n = 190$) in the SemNets, including: (1) the *weight*, that is the brute strength of association extracted from RJT ratings (the higher the *weight* is, the more associated two nodes are) and is thus easily interpretable; (2) the *efficiency*, that is the inverse shortest path length between two nodes (the higher the *efficiency* is, the more efficient the connection between two nodes is) and has been shown to be a more suitable measure than path length in individual-based SemNets such as the ones we analyzed in our study[59]; (3) the *clustering coefficient*, that is the degree to which neighbor nodes are also connected to one another (the higher the *clustering coefficient* is, the more interconnected words are); and (4) the *eigenvector centrality*, that is a self-referential measure of centrality that considers the centrality of neighbor (the higher the *eigenvector centrality* is, the more central and influent the node is) and is the centrality measure with the highest reliability compared to other centrality measures in cognitive networks[66]. In previous studies, *efficiency*, *clustering coefficient*, and *eigenvector centrality* have all shown promising correlations with creativity measures[52,53,61,71,72]. For example, more creative individuals often exhibit a more connected (higher *clustering coefficient*) and efficient (higher *efficiency*) SemNet than less creative individuals[48,49,51,52,60]. This suggests that ideas in more creative people were more interconnected and closer, leading to more flexible and efficient spontaneous association of ideas.

Then, we compared pre- and post-SemNets local metrics to explore changes in the properties of SemNets that could be related to creative problem-solving. For this purpose, we calculated an individual difference for each metric between the SemNet built after and before a riddle ($\Delta \text{Metric} = \text{Metric}_{\text{Post-SemNet}} - \text{Metric}_{\text{Pre-SemNet}}$). The higher the $\Delta \text{Metric}$, the larger the increase in the considered metric after working on the riddle.

We excluded one RJT of two participants from SemNets metrics analyses because of technical issues during this specific task (Bar: $n = 1$; Zoe: $n = 1$). In addition, we excluded the RJT in which participants did not respond or rated the word pairs as zero in more than 10% of all trials in one RJT (Zoe: $n = 12$; Bar: $n = 10$; Daniel: $n = 15$; Car: $n = 11$). We considered that these participants were not sufficiently engaged in the RJT and that missing or zero ratings would importantly bias the network with missing links. Non-responses were substituted by a zero for metrics that cannot be computed if missing values are included. The final sample included 22 solvers (Zoe: $n = 10$; Daniel: $n = 3$; Bar: $n = 2$; Car: $n = 7$) and 120 non solvers (Zoe: $n = 26$; Daniel: $n = 31$; Bar: $n = 32$; Car: $n = 31$) distributed over 80 different participants (see Tables S4 and S5 for more details).

### Creativity tasks

In addition, we explored how the ability to solve the riddles related to creative abilities measured with several tasks including an adaptation of the remote associate task[73,74], the short version of the Torrance test of creative thinking[75], and the inventory of creative activities and achievements (ICAA)[76]. All the methodology and results related to these tasks are detailed in the Supplementary Note 1. The creativity tasks did not interfere with the riddle as they were semantically unrelated.

### Statistical analyses

To explore the impact of analogical transfer in problem-solving, we tested whether the solving rate (all riddles confounded) differed between *naive* and *transfer* conditions using chi-square analyses (corrected with Yates method if needed). The same analyses at the riddle level were provided in the Supplementary Note 2. In addition, we compared the averaged response time of correct responses (all riddles confounded) between *naive* and *transfer* conditions using non-parametric paired Wilcoxon tests. We ran additional chi-square tests (i) to ensure that the solving rate was not dependent of the experimental design (first or second presentation order of the pair of riddles), and (ii) to explore whether the solving rates of riddles in the *transfer* condition differed whether the initial riddle was correctly solved or not.

Then, we investigated the relationships between solving success and insight solving by exploring the proportion of Eureka reports in correct versus incorrect responses. To this end, we ran a chi-square test for the *naive* and *transfer* conditions separately, grouping all the riddles and all the participants together, excluding missing values.

Finally, additional analyses were conducted to explore participants' response confidence when giving correct responses (versus incorrect responses), and their attentional level during the solving phases using the probes. The methodology and results of these analyses are detailed in Supplementary Note 3.

We used a nonlinear mixed-effects model to explore how changes in SemNets related to a problem could predict its solving. In this model, successful solving was the dependent variable (isSolved, binary variable), and the independent variables were the difference between the two SemNets of a given metric ($\Delta$Metric, continuous variable), the *impact rating* (IR, continuous variable), the *semantic distance* (SD, continuous variable), and the interaction factor between the three ($\Delta$Metric × IR, $\Delta$Metric × SD, $\Delta$Metric × IR × SD, continuous variables). $\Delta$Metric, *impact rating*, and *semantic distance* variables were z-scored across the whole group. Participants were entered as a random-effect factor in the model on both the intercept (1|Subject) and slope ($\Delta$Metric|Subject). By default, we considered that the two random effects were independent as we have no argument to claim that subject would differ similarly on both slope and intercept. These random effects allow us to take into account the repeated measures across subjects as a random-effect factor (Subject, maximum two riddles per subject) and inter-individual variability. The model can be formalized as follows:

$$
\begin{aligned}
\text{isSolved} = &\ \beta_0 + \beta_1 \times \Delta\text{Metric} + \beta_2 \times \text{IR} + \beta_3 \times \text{SD} + \beta_4(\Delta\text{Metric} \times \text{IR}) \\
&+ \beta_5(\Delta\text{Metric} \times \text{SD}) + \beta_6(\text{IR} \times \text{SD}) + \beta_7(\Delta\text{Metric} \times \text{IR} \times \text{SD}) \quad (1) \\
&+ (1|\text{Subject}) + (-1 + \Delta\text{Metric}|\text{Subject})
\end{aligned}
$$

We designed similar nonlinear mixed-effects models to explore how changes in SemNets related to a problem could predict the solving of an analogous one. Here, analogous problem-solving was the dependent variable (isTransfer, binary variable). The independent variables were the same as in the previous model (i.e., $\Delta$Metric *impact rating*, *semantic distance*, and all possible interactions). $\Delta$Metric, *impact rating*, and *semantic distance* variables were z-scored. Participants were entered as a random-effect factor in the model. The models can be formalized as follows:

$$
\begin{aligned}
\text{isTransfer} = &\ \beta_0 + \beta_1 \times \Delta\text{Metric} + \beta_2 \times \text{IR} + \beta_3 \times \text{SD} + \beta_4(\Delta\text{Metric} \times \text{IR}) \\
&+ \beta_5(\Delta\text{Metric} \times \text{SD}) + \beta_6(\text{IR} \times \text{SD}) + \beta_7(\Delta\text{Metric} \times \text{IR} \times \text{SD}) \\
&+ (1|\text{Subject}) + (-1 + \Delta\text{Metric}|\text{Subject})
\end{aligned}
$$

$$(2)$$

By adding both *impact rating* and *semantic distance* in statistical models, the common variance between the two variables is left out in the model residues, allowing us to distinguish changes in SemNets that are purely driven by the solution and those likely to capture more specifically the creative restructuring process (because they reflect how initially remote concepts are combined).

In all these models, we were particularly interested in the $\Delta$Metric effect, the $\Delta$Metric by *impact rating* interaction effect, and the $\Delta$Metric by *semantic distance* interaction effect. A $\Delta$Metric effect represents whether a restructuring at the global (SemNet) level (i.e., average of $\Delta$Metric across edges/nodes at the individual level) was associated with successful problem-solving. A $\Delta$Metric × *impact rating* interaction effect represents a local restructuring in parts of the SemNet that have been assessed as relevant in solving the problem (i.e., a solution-based restructuring). A $\Delta$Metric × *semantic distance* interaction effect represents a local restructuring targeting the most semantically distant word pairs in the SemNet (i.e., a remoteness-based restructuring). Finally, a $\Delta$Metric × *impact rating* x *semantic distance* interaction effect explores whether solution-based and remoteness-based restructuring have a synergistic or opposite effect.

We replicated the same model for each metric (*weight*, *efficiency*, *clustering coefficient*, and *eigenvector centrality*). Hence, we applied a correction for multiple comparisons adapted for correlated variables[77]. We calculated the effective number of tests ($M_{eff}$) among our four correlated variables (*weight*, *efficiency*, *clustering coefficient*, and *eigenvector centrality*) with the following formula:

$$
M_{eff} = 1 + \left[ (k - 1) \times \left( 1 - \text{var(lambda)}/k \right) \right] \quad (3)
$$

In this formula, k represents the number of correlated variables (in our case, $k = 4$), and lambda is a vector of eigenvalues of length k for the variables of interest. To compute lambda, we built a correlation matrix of $\Delta$Metric between the four variables using Spearman correlation (Table S6). We first averaged $\Delta$Metric at the individual level because we did not have the same number of $\Delta$Metric values for each variable (190 for edge metrics, and 20 for node metrics). The vector of eigenvalues was computed using *eigen* function in Rstudio. The resulting $M_{eff}$ indicated a correction for 2.67 tests, resulting in a significant *p* value threshold of 0.0187 (i.e., 0.05/$M_{eff}$).

All nonlinear mixed models were run on Rstudio (v 1.4.1717) with *glmer* function. For each significant model, we calculated the *balanced accuracy*. It measures the probability that the model correctly classifies the variables according to the condition to be explained (i.e., solver or non-solver), considering its unbalanced distribution. Finally, to ensure that the significant relationships that we observed between SemNets changes and problem-solving were not influenced by individual creative abilities, we ran the same models with each creativity score added as a predictor. These control analyses investigated whether the results remained significant after adding creative measures to the statistical models (see Supplementary Note 1).

## Replication study (Study 2)

Since our approach is novel, we ran a replication study in an independent sample to ensure the validity of our results. This second sample comprised 151 participants (mean age = 22 years, SEM = 1.80 years; 112 women and 39 men), recruited under the same inclusion criteria and ethical approval as the initial study. Participants followed the same experimental procedure (RJT before the riddle presentation and after a 10-min solving phase), except that they only had to solve one riddle, the same for everyone (we chose the Zoe riddle because it had the largest solving rate in our initial study). We arbitrarily chose to include about 150 participants to verify that the effects of the initial study were replicated. This sample size triples the number of individuals compared to the initial sample (49 individuals had to solve Zoe riddle). We used the same material (Zoe riddle and its 20 words list) and the same SemNets estimation procedure detailed above (in particular, we used the same SemNets metrics).

We excluded five participants because they already knew the riddle, and one additional participant because of a technical issue during the experiment. For the SemNets statistical analyses, 26 participants (four solvers and 22 non-solvers) were excluded following the same criteria used for SemNets cleaning as in the initial study (>10% zero ratings or missing values in the RJT). We also excluded one participant who found the solution after the post-RJT since we did not know if the solution came to his/her mind in the midst of performing the RJT (potentially influencing the rating halfway through the RJT). The final sample comprised 118 participants, including 27 solvers (23%) and 91 non-solvers (77%).

We used similar statistical analyses to explore the relationship between SemNets changes and problem-solving. Because of the new data format (i.e., one riddle per individual), we removed the participants' random-effect factor on the intercept from the nonlinear mixed-effects models. This random effect was added in the initial models to capture the repeated measures across subjects, which is no longer relevant here. However, we kept the participants' random-effect factor on the $\Delta$Metric. The model used in

the replication sample can be formalized as follows:

$$
\begin{aligned}
\text{isSolved} = &\; \beta_0 + \beta_1 \times \Delta\text{Metric} + \beta_2 \times \text{IR} + \beta_3 \times \text{SD} + \beta_4(\Delta\text{Metric} \times \text{IR}) \\
&+ \beta_5(\Delta\text{Metric} \times \text{SD}) + \beta_6(\text{IR} \times \text{SD}) + \beta_7(\Delta\text{Metric} \times \text{IR} \times \text{SD}) \\
&+ (-1 + \Delta\text{Metric}|\text{Subject})
\end{aligned} \quad (4)
$$

Finally, we explored if SemNets changes were different whether participants solved the problem with a Eureka (i.e., insight problem-solving) or not. This analysis could not be done in Study 1 alone because of the low solving rate of the riddles. Hence, we combined the solvers of Zoe riddle from Studies 1 ($n = 10$, including 9 with Eureka) and 2 ($n = 26$, including 20 with Eureka). Note that one participant was removed from Study 2 because of missing data concerning the Eureka report. We used similar statistical analyses to explore the relationship between SemNets changes and insight problem-solving (isInsight, binary variable) as above. The model can be formalized as follows:

$$
\begin{aligned}
\text{isInsight} = &\; \beta_0 + \beta_1 \times \Delta\text{Metric} + \beta_2 \times \text{IR} + \beta_3 \times \text{SD} + \beta_4(\Delta\text{Metric} \times \text{IR}) \\
&+ \beta_5(\Delta\text{Metric} \times \text{SD}) + \beta_6(\text{IR} \times \text{SD}) + \beta_7(\Delta\text{Metric} \times \text{IR} \times \text{SD}) \\
&+ (-1 + \Delta\text{Metric}|\text{Subject})
\end{aligned} \quad (5)
$$

The statistical results were not corrected for multiple comparisons in the analyses of Study 2 because our choice of SemNets metrics was hypotheses-driven, following the results of Study 1.

### Control study (Study 3)

We ran an additional study to better understand how restructuring (measured by changes in SemNets organization) promoted problem-solving. Specifically, Study 3 explored whether changes in SemNets organization differed between participants who found the solution by themselves (solver group) from those to whom the solution was given (solution group).

This third sample comprised 47 new participants (mean age = 26 years, SEM = 0.73 years; 27 women and 20 men), recruited under the same inclusion criteria and ethical approval as the previous studies. Participants followed the same experimental procedure as in Study 2 (RJT before the riddle presentation and after a 10-min solving phase, only Zoe riddle, no riddle in *transfer* condition). We used the same material (Zoe riddle and its 20 RJT words list) and the same SemNets estimation procedure detailed above (in particular, we used the same SemNets metrics). The only difference concerned the non-solvers: at the end of the 10-min allotted time to solve the riddle, the solution was given to participants who failed in solving the riddle before completing the second RJT.

For the SemNets analyses, nine participants (two solvers and seven non-solvers) were excluded following the same exclusion criteria based on the SemNets cleaning as in the initial study (>10% zero ratings or missing values in the RJT). Three additional solvers were excluded because the experimenter accidentally gave them the solution before completing the second RJT. The final sample included 35 participants, with ten in the solver group (29%) and 25 in the solution group (71%). To increase our statistical power, we combined the ten solvers of Study 3 with all the solvers of Zoe riddle from Studies 1 and 2. Thus, in total, the solver group was composed of 47 participants (Study 1: $n = 10$; Study 2: $n = 27$; Study 3: $n = 10$).

We used the same statistical models as before to explore whether SemNets changes were different whether participants found the solution themselves or not. In the model, the dependent variable (isSolved) was a binary variable representing whether the participant solved the problem alone. It can be formalized as follows:

$$
\begin{aligned}
\text{isSolved} = &\; \beta_0 + \beta_1 \times \Delta\text{Metric} + \beta_2 \times \text{IR} + \beta_3 \times \text{SD} + \beta_4(\Delta\text{Metric} \times \text{IR}) \\
&+ \beta_5(\Delta\text{Metric} \times \text{SD}) + \beta_6(\text{IR} \times \text{SD}) + \beta_7(\Delta\text{Metric} \times \text{IR} \times \text{SD}) \\
&+ (-1 + \Delta\text{Metric}|\text{Subject})
\end{aligned} \quad (6)
$$

The statistical results were not corrected for multiple comparisons as our analyses and SemNets metrics were predetermined by Study 1.

## Results

### Solving a problem with and without analogical transfer (Study 1)

The solving rate (all riddles confounded) was 15.6% ($n = 30/192$ solvers, 11 with Eureka) in the *naive* condition, and 34% ($n = 65/191$ solvers, 49 with Eureka) in the *transfer* condition (see Table S7 for details by riddle). Note that non-solvers were given the solution to the first riddle before the transfer condition. The solving rate was significantly higher in the *transfer* condition ($\chi^2(1) = 17.39$, $p = 3.04 \times 10^{-5}$) compared to the *naive* condition. Analyses at the riddle level are provided in Supplementary Note 2. In addition, the response times of correct responses were, on average, shorter in the *transfer* condition (75 s, SEM = 10 s) compared to the *naive* condition (210 s, SEM = 33 s, W = 1400, $p = 6.77 \times 10^{-4}$, $r = -0.35$).

The solving rate did not statistically differ whether the riddle was the first or the second presented riddle, be it in the *naive* condition (11.3% in the first position and 20% in the second position, $\chi^2(1) = 2.73$, $p = 0.10$) or the *transfer* condition (33.3% in the first position and 34.7% in the second position, $\chi^2(1) = 1.52$, $p = 0.22$). Finally, the solving rate in the *transfer* condition did not statistically differ whether the initial riddle was correctly solved or not (*transfer* condition: 45.5% and 31% respectively, $\chi^2(1) = 3.71$, $p = 0.054$).

Correct responses were more often associated with a Eureka experience than incorrect responses in the *naive* condition (36.7% and 21.3% respectively, $\chi^2(1) = 4.10$, $p = 0.04$), and in the *transfer* condition (75.4% and 18.8% respectively, $\chi^2(1) = 94.05$, $p = 3.06 \times 10^{-22}$).

Importantly, we explored how the ability to solve the riddles related to creative abilities, measured with several assessments. We found that creativity scores including measures of convergent and divergent thinking positively and significantly correlated with the solving rate of the four riddles (see Supplementary Note 1 and Fig. S4).

Overall, we found a behavioral signature of analogy transfer (higher solving rate, faster response time) in problem-solving. In addition, we provided evidence suggesting that solving our riddles involves creativity-related processes (correlation with other measures of creativity and a positive association between correct responses and Eureka report).

### Local changes in problem-related SemNets relate to successful solving (Study 1)

In the *naive* condition, we explored whether the ability to solve problems could be reflected in changes in SemNets organization, either at the global (i.e., SemNet) or local (i.e., node or edge) level. We used nonlinear mixed-effects models to predict problem-solving based on the difference between individual-based SemNets metrics over time ($\Delta\text{Metric} = \text{Metric}_{\text{PostSemNet}} - \text{Metric}_{\text{PreSemNet}}$), the *impact rating* (measuring the importance of each node or edge to solve the problem), the *semantic distance* (measuring the semantic remoteness of each node or edge), and all possible interaction effects between these factors. We expected to find positive SemNets metric changes in parts of the SemNet that were solution-relevant and semantically remote in solvers compared to non-solvers.

We found two major results (all statistical results are in Table 1). First, we found a significant positive effect of the interaction between $\Delta\text{Metric}$ and *impact rating* on problem-solving for three of our metrics including *weight* ($\beta = 0.07$, $p = 0.01$, confidence interval 95%, CI95 = [0.02;0.13]), *efficiency* ($\beta = 0.09$, $p = 0.004$, CI95 = [0.03;0.14]), and *eigenvector centrality* ($\beta = 0.25$, $p = 0.009$, CI95 = [0.06;0.44]) (Fig. 2A). These interaction effects indicate that larger positive changes in *weight*, *efficiency*, and *eigenvector centrality* in pairs/words judged as more helpful to solve the riddles were associated with a higher solving success rate. In other words, helpful word pairs were rated in the post-RJT as more related and connected (*weight* and *efficiency*) after successfully solving the problem than before the problem presentation, and helpful nodes became more central/influential in the networks (*eigenvector centrality*). This interaction effect was not significant for *clustering coefficient* ($\beta = 0.04$, $p = 0.84$, CI95 = [−0.31;0.37]).

**Table 1 | Results of nonlinear mixed model analyses in Study 1 (problem-solving)**

| | | β | SE | z value | *p* value | CI 95% |
|---|---|---|---|---|---|---|
| Weight (BA = 0.76) | ΔMetric | 0.08 | 0.11 | 0.70 | 0.48 | −0.15;0.30 |
| | IR | 1.15e-03 | 0.03 | 0.04 | 0.95 | −0.05;0.02 |
| | SD | −0.01 | 0.03 | −0.53 | 0.59 | −0.07;0.04 |
| | ΔMetric × IR | 0.07 | 0.03 | 2.49 | **0.0127\*** | 0.02;0.13 |
| | ΔMetric × SD | 0.02 | 0.03 | 0.72 | 0.47 | −0.04;0.09 |
| | IR × SD | 6.15e-03 | 0.03 | 0.23 | 0.82 | −0.05;0.06 |
| | ΔMetric × IR × SD | 0.05 | 0.03 | 1.52 | 0.13 | −0.01;0.12 |
| Efficiency (BA = 0.78) | ΔMetric | 0.12 | 0.11 | 1.07 | 0.28 | −0.11;0.35 |
| | IR | 5.20e-03 | 0.03 | 0.20 | 0.84 | −0.05;0.06 |
| | SD | −8.28e-03 | 0.03 | −0.32 | 0.75 | −0.06;0.04 |
| | ΔMetric × IR | 0.09 | 0.03 | 2.87 | **4.14e-03\*** | 0.03;0.14 |
| | ΔMetric × SD | 0.08 | 0.03 | 2.37 | **0.0176\*** | 0.01;0.15 |
| | IR × SD | 0.01 | 0.03 | 0.44 | 0.66 | −0.04;0.06 |
| | ΔMetric × IR × SD | 0.07 | 0.03 | 2.00 | **0.0457** | 0.002;0.13 |
| Clustering coefficient | ΔMetric | 0.16 | 0.56 | 0.29 | 0.77 | −0.93;1.16 |
| | IR | 0.12 | 0.11 | 1.11 | 0.27 | −0.09;0.33 |
| | SD | −0.18 | 0.11 | −1.62 | 0.11 | −0.41;0.04 |
| | ΔMetric × IR | 0.04 | 0.18 | 0.21 | 0.84 | −0.31;0.37 |
| | ΔMetric × SD | −0.15 | 0.18 | −0.82 | 0.41 | −0.49;0.21 |
| | IR × SD | −0.07 | 0.11 | −0.66 | 0.51 | −0.29;0.15 |
| | ΔMetric × IR × SD | 0.11 | 0.19 | 0.60 | 0.55 | −0.25 0.48 |
| Eigenvector centrality (BA = 0.74) | ΔMetric | −0.02 | 0.09 | −0.21 | 0.84 | −0.20;0.16 |
| | IR | 9.80e-03 | 0.08 | 0.12 | 0.91 | −0.15;0.17 |
| | SD | 8.17e-03 | 0.09 | 0.10 | 0.92 | −0.16;0.17 |
| | ΔMetric × IR | 0.25 | 0.10 | 2.62 | **8.87e-03\*** | 0.06;0.44 |
| | ΔMetric × SD | 0.07 | 0.10 | 0.71 | 0.48 | −0.12;0.26 |
| | IR × SD | −8.15e-03 | 0.09 | −0.09 | 0.93 | −0.18;0.17 |
| | ΔMetric × IR × SD | 0.03 | 0.10 | 0.30 | 0.76 | −0.17;0.23 |

ΔMetric represents the difference between the considered metric between the PreSemNet and the PostSemNet. *Impact rating* (IR) corresponds to the importance of edges or nodes for solving the problem based on independent assessment. *Semantic distance* (SD) represents the semantic remoteness of edges or nodes based on all participants' baseline pre-RJT before the presentation of the problem. We ran a mixed model for each metric. Significant results (*p* < 0.05) are in bold, and highlighted with a star if they remained significant after correction for multiple comparisons. Balanced accuracy (BA) is indicated for significant models.

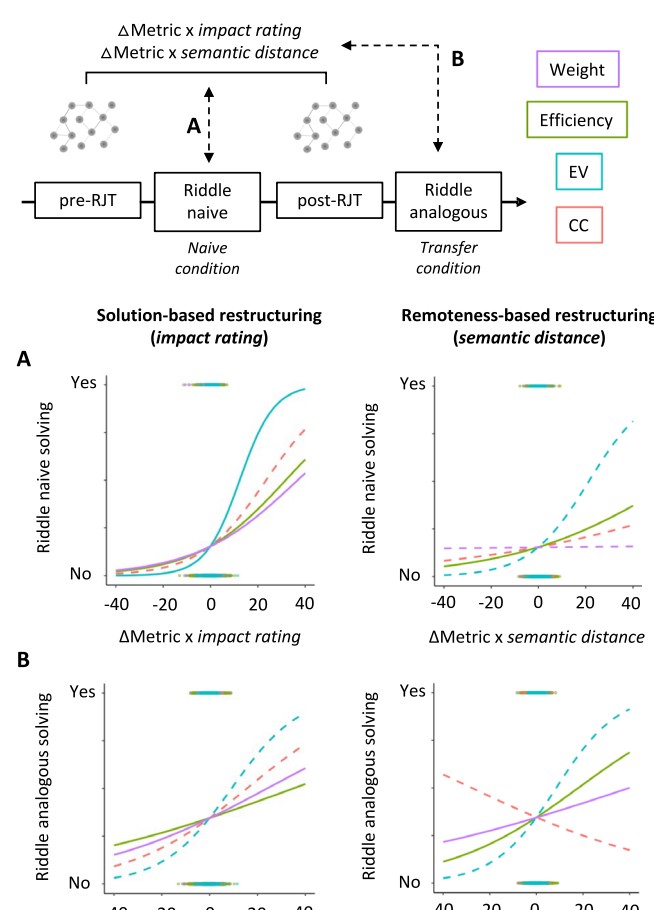

**Fig. 2 | Local SemNets changes were associated with problem-solving in both *naive* and *transfer* conditions.** The analysis design is schematized in the top panel. The pre- and post-RJT used words related to the riddle in the *naive* condition (i.e., Riddle naive). Riddles in the *naive* condition (Riddle naive) were in a different domain than the ones in the *transfer* condition (i.e., Riddle analogous). Graphs represent the ΔMetric weighted by the *impact rating* (ΔMetric x *impact rating* – solution-based SemNets changes) or by the *semantic distance* (ΔMetric × *semantic distance* – remoteness-based SemNets changes) as a function of solving success. Lines represent the fitting curves of the interaction effect (ΔMetric × *impact rating* on the left, and Δmetric x *semantic distance* on the right) in predicting problem-solving in *naive* (**A**, *n* = 23,560 observations for edge metric, *n* = 2480 observations for node metric) and *transfer* (**B**, *n* = 26,980 observations for edge metric, *n* = 2840 observations for node metric) conditions in the four mixed models computed for local SemNets metrics (*weight* – in purple, *efficiency* – in green, *eigenvector centrality* – EV in blue, and *clustering coefficient* – CC in red). A solid line indicates a significant effect (*p* < 0.05).

Second, we found a significant positive effect of the interaction between ΔMetric and *semantic distance* on problem-solving for *efficiency* (**β** = 0.08, *p* = 0.02, CI95 = [0.01;0.15]), suggesting that larger positive changes in *efficiency* in semantically remote word pairs were associated with a higher solving success rate. In other words, bringing closer together initially semantically remote associations related to successful problem-solving. This interaction effect was not significant for the other metrics (*weight*: **β** = 0.02, *p* = 0.72, CI95 = [−0.04;0.09]; *clustering coefficient*: **β** = −0.15, *p* = 0.41, CI95 = [−0.49;0.21]; and *eigenvector centrality*: **β** = 0.07, *p* = 0.48, CI95 = [−0.12;0.26]) (Fig. 2A).

Note that the *balanced accuracies* of the significant models ranged between 0.74 and 0.76, indicating that the models were powerful enough to predict problem-solving (Table 1). Importantly, we did not find statistically significant differences in metric values at baseline (i.e., Metric$_{PreSemNet}$) between solvers and non-solvers (see Supplementary Note 4). In addition, the significant relationship that we found between SemNets changes and problem-solving remained significant when creative measures were added to the model (see Supplementary Note 1, Table S8, and Fig. S7), indicating that our findings reflect the creative process rather than individual creative traits.

Overall, these findings are consistent with the hypothesis that solving success implies local changes of SemNets targeting the most helpful (ΔMetric × *impact rating* interaction effect) or the most semantically distant (ΔMetric × *semantic distance* interaction effect) nodes and edges[35].

**Table 2 | Results of nonlinear mixed model analyses in Study 1 (analogous problem-solving)**

| | | β | SE | z value | *p* value | CI 95% |
|---|---|---|---|---|---|---|
| Weight (BA = 0.82) | ΔMetric | 0.04 | 0.07 | 0.59 | 0.55 | −0.09;0.17 |
| | IR | 3.29e-04 | 0.02 | 0.02 | 0.99 | −0.04;0.04 |
| | SD | −2.05e-03 | 0.02 | −0.09 | 0.93 | −0.05;0.04 |
| | ΔMetric × IR | 0.05 | 0.02 | 2.21 | **0.02736** | 0.01;09 |
| | ΔMetric × SD | 0.08 | 0.03 | 2.96 | **3.13e-03*** | 0.03;0.13 |
| | IR × SD | −3.77e-05 | 0.02 | −2.00e-03 | 1.00 | −0.05;05 |
| | ΔMetric × IR × SD | −0.01 | 0.03 | −0.57 | 0.57 | −0.07;04 |
| Efficiency (BA = 0.82) | ΔMetric | 0.03 | 0.08 | 0.44 | 0.66 | −0.11;18 |
| | IR | −5.39e-03 | 0.02 | −0.24 | 0.81 | −0.05;0.04 |
| | SD | −2.40e-03 | 0.02 | −0.11 | 0.91 | −0.05;0.04 |
| | ΔMetric × IR | 0.06 | 0.02 | 2.50 | **0.01237*** | 0.01;0.10 |
| | ΔMetric × SD | 0.07 | 0.03 | 2.68 | **7.27e-03*** | 0.02;0.11 |
| | IR × SD | 7.19e-03 | 0.02 | 0.31 | 0.76 | −0.04;0.05 |
| | ΔMetric × IR × SD | 0.03 | 0.03 | 1.01 | 0.31 | −0.02;07 |
| Clustering coefficient | ΔMetric | 0.13 | 0.46 | 0.28 | 0.78 | −0.79;1.06 |
| | IR | −0.05 | 0.09 | −0.60 | 0.55 | −0.23;0.12 |
| | SD | 0.05 | 0.09 | 0.54 | 0.59 | −0.13;0.23 |
| | ΔMetric × IR | 0.10 | 0.12 | 0.81 | 0.42 | −0.14;0.34 |
| | ΔMetric × SD | −0.07 | 0.12 | −0.57 | 0.57 | −0.31;0.17 |
| | IR × SD | −0.04 | 0.09 | −0.48 | 0.63 | −0.22;0.13 |
| | ΔMetric × IR × SD | −0.07 | 0.13 | −0.56 | 0.58 | −0.33;0.18 |
| Eigenvector centrality | ΔMetric | 0.03 | 0.08 | 0.44 | 0.66 | −0.11;0.18 |
| | IR | 3.92e-03 | 0.07 | 0.06 | 0.96 | −0.13;0.14 |
| | SD | 3.66e-03 | 0.07 | 0.05 | 0.96 | −0.13;0.14 |
| | ΔMetric × IR | 0.14 | 0.08 | 1.67 | 0.09 | −0.02;0.30 |
| | ΔMetric × SD | 0.08 | 0.07 | 1.04 | 0.30 | −0.07;0.22 |
| | IR × SD | −0.06 | 0.07 | −0.78 | 0.44 | −0.20;0.09 |
| | ΔMetric × IR × SD | −0.06 | 0.08 | −0.74 | 0.46 | −0.22;0.10 |

ΔMetric represents the difference between the considered metric between the PreSemNet and the PostSemNet. *Impact rating* (IR) corresponds to the importance of edges or nodes for solving the problem based on independent assessment. *Semantic distance* (SD) represents the semantic remoteness of edges or nodes based on all participants' baseline pre-RJT before the presentation of the problem. We ran a mixed model for each metric to predict the solving of the analog problem. Significant results (*p* < 0.05) are in bold, and highlighted with a star if they remained significant after correction for multiple comparisons. Balanced accuracy (BA) is indicated for significant models.

## Changes in a problem-related SemNets relate to successfully solving an analogous problem (Study 1)

As for the *naive* condition, we used nonlinear mixed-effect models to predict problem-solving in the *transfer* condition (analogous problem) based on the changes in SemNets ΔMetrics, *impact rating,* and *semantic distance* that were related to the naive problem. We expected to find more positive SemNets metric changes in parts of the network that were solution-relevant and semantically remote in solvers of the analogous problem compared to non-solvers, as both types of changes can lead to a broader representation of the problem.

We found two primary results (all statistical results are in Table 2). First, we found a significant positive interaction effect of ΔMetric and *impact rating* on analogous problem-solving for *efficiency* (β = 0.06, *p* = 0.01, CI95 = [0.01;0.10]), suggesting that solving an analogous problem was related to a higher connection of helpful edges in the mental representation of the initial problem (Fig. 2B). This interaction was also significant for *weight* (β = 0.05, *p* = 0.03, CI95 = [0.01;0.09]) but did not survive correction for multiple comparisons and was not statistically significant for *clustering coefficient* (β = 0.10, *p* = 0.42, CI95 = [−0.14;0.34]) and *eigenvector centrality* (β = 0.14, *p* = 0.09, CI95 = [−0.02;0.30]).

Second, we found a significant positive interaction effect of ΔMetric and *semantic distance* on analogous problem-solving for *weight* (β = 0.08, *p* = 0.003, CI95 = [0.03;0.13]), and *efficiency* (β = 0.07, *p* = 0.007, CI95 = [0.02;0.11]), suggesting that combining remote concepts when attempting to solve a first riddle helped solving an analogous one (Fig. 2B). This interaction was not statistically significant for *clustering coefficient* (β = −0.07, *p* = 0.57, CI95 = [−0.31;0.17]), or *eigenvector centrality* (β = 0.08, *p* = 0.30, CI95 = [−0.07;0.22]).

Note that the *balanced accuracies* of the significant models were 0.82 indicating that the models were powerful enough to predict analogous problem-solving (Table 2). In addition, we ran similar analyses on a non-analogous problem as a control (Fig. S5). Using similar statistical models, we did not find a statistically significant relationship between changes in a given problem-related SemNet and the solving of another problem that had no analogical similarities with the initial riddle (see Supplementary Note 5 and Table S9).

Overall, these results show that the restructuring of the mental representation of a problem (approximated by local SemNets changes) facilitates the solving of a different but analogous one.

## Replicating the relationship between SemNet changes and problem-solving (Study 2)

To assess the reliability of our results, we conducted a replication study with 151 independent participants. Participants underwent the exact same experimental procedure as in our first study, except that they only had to solve a single riddle (Zoe, which had the largest solving rate in Study 1) and

**Table 3 | Results of nonlinear mixed model analyses in Study 2 (problem-solving)**

| | | β | SE | z value | p value | CI 95% |
|---|---|---|---|---|---|---|
| Weight (BA = 0.52) | ΔMetric | 2.26e−03 | 0.05 | 0.05 | 0.96 | −0.09;0.10 |
| | IR | −4.36e−03 | 0.02 | −0.25 | 0.81 | −0.04;0.03 |
| | SD | 0.01 | 0.02 | 0.75 | 0.45 | −0.02;0.05 |
| | ΔMetric × IR | 9.95e−03 | 0.02 | 0.53 | 0.60 | −0.03;0.05 |
| | ΔMetric × SD | 0.07 | 0.02 | 3.49 | **4.91e−04** | 0.03;0.12 |
| | IR × SD | 7.90e−03 | 0.02 | 0.50 | 0.62 | −0.02;0.04 |
| | ΔMetric × IR × SD | −2.16e−03 | 0.02 | −0.12 | 0.90 | −0.04;0.03 |
| Efficiency (BA = 0.57) | ΔMetric | 0.01 | 0.09 | 0.15 | 0.88 | −0.17;0.19 |
| | IR | −3.81e−03 | 0.02 | −0.21 | 0.83 | −0.04;0.03 |
| | SD | 0.02 | 0.02 | 1.23 | 0.22 | −0.01;0.06 |
| | ΔMetric × IR | 0.03 | 0.02 | 1.56 | 0.12 | −0.008;0.07 |
| | ΔMetric × SD | 0.08 | 0.02 | 3.93 | **8.39e−05** | 0.04;0.12 |
| | IR x SD | 6.13e−03 | 0.02 | 0.38 | 0.70 | −0.03;0.04 |
| | ΔMetric × IR × SD | 0.02 | 0.02 | 1.13 | 0.26 | −0.01;0.06 |
| Clustering coefficient (BA = 0.76) | ΔMetric | 0.08 | 0.38 | 0.21 | 0.84 | −0.66;0.81 |
| | IR | 0.06 | 0.10 | 0.59 | 0.56 | −0.14;0.27 |
| | SD | −0.02 | 0.10 | −0.17 | 0.87 | −0.22;0.19 |
| | ΔMetric × IR | 0.31 | 0.15 | 2.00 | **0.045** | 0.01;0.61 |
| | ΔMetric × SD | −0.11 | 0.15 | −0.75 | 0.46 | −0.40;18 |
| | IR × SD | −8.27e−03 | 0.11 | −0.08 | 0.94 | −0.22;0.21 |
| | ΔMetric × IR × SD | 0.31 | 0.17 | 1.86 | 0.06 | −0.02;0.63 |
| Eigenvector centrality | ΔMetric | 9.51e−03 | 0.06 | 0.15 | 0.88 | −0.12;0.14 |
| | IR | −9.50e−03 | 0.08 | −0.12 | 0.90 | −0.16;0.14 |
| | SD | 4.10e−03 | 0.08 | 0.054 | 0.96 | −0.15;0.15 |
| | ΔMetric × IR | 0.07 | 0.08 | 0.86 | 0.39 | −0.08;0.22 |
| | ΔMetric × SD | 0.14 | 0.07 | 1.90 | 0.0578 | −0.005;0.29 |
| | IR × SD | 9.60e−03 | 0.08 | 0.12 | 0.91 | −0.15;0.17 |
| | ΔMetric × IR × SD | 0.07 | 0.08 | 0.82 | 0.41 | −0.09;0.22 |

ΔMetric represents the difference between the considered metric between the PreSemNet and the PostSemNet. *Impact rating* (IR) corresponds to the importance of edges or nodes for solving the problem based on independent assessment. *Semantic distance* (SD) represents the semantic remoteness of edges or nodes based on all participants' baseline pre−RJT before the presentation of the problem. We ran a mixed model for each metric to predict the solving of the problem. Significant results ($p < 0.05$) are in bold. Balanced accuracy (BA) is indicated for significant models.

that there was no *transfer* condition. In this new sample, 22% of participants found the solution (32/145, 23 with a Eureka). As in Study 1, most solvers (31/32) found the solution during the solving phase that directly followed the riddle presentation. One solver found the solution after the second RJT (post-RJT).

We used nonlinear mixed-effects models to explore whether global or local SemNets changes could predict successful problem-solving. We partially replicated our results from the first study (all statistical results are in Table 3). We found a significant positive interaction effect of ΔMetric and *impact rating* on problem-solving for *clustering coefficient* ($\beta = 0.31$, $p = 0.045$, CI95 = [0.01;0.61]) but not the other metrics (*weight*: $\beta = 0.01$, $p = 0.60$, CI95 = −0.03;0.05; *efficiency*: $\beta = 0.03$, $p = 0.12$, CI95 = −0.01;0.07; and *eigenvector centrality*: $\beta = 0.07$, $p = 0.39$, CI95 = [−0.08;0.22]). In addition, we found a significant positive interaction effect of ΔMetric and *semantic distance* on problem-solving for *weight* ($\beta = 0.07$, $p = 4.91 \times 10^{-4}$, CI95 = [0.03;0.12]) and *efficiency* ($\beta = 0.08$, $p = 8.39 \times 10^{-5}$, CI95 = [0.04;0.12]), but not for *clustering coefficient* ($\beta = -0.11$, $p = 0.46$, CI95 = [−0.40;0.18]) nor *eigenvector centrality* ($\beta = 0.14$, $p = 0.06$, CI95 = [−0.01;0.29]) (Fig. S6).

Note that the *balanced accuracies* of the significant models ranged between 0.52 and 0.76 indicating that the models were reasonably powerful to predict problem-solving (Table 3). In addition, we did not find statistically significant differences in metric values at baseline (i.e., Metric$_{preSemNet}$) between solvers and non-solvers (see Supplementary Note 4). In summary,

we partially replicated Study 1, where local changes in SemNets based on the impact rating and the semantic distance (as measured by the weight, the efficiency, and the clustering coefficient) were associated with successful problem-solving.

## Changes in problem-related SemNets relate to insight solving (Studies 1 and 2)

Thanks to this additional dataset, we could explore how solving with a Eureka related to SemNets changes. Such analysis was not possible using each dataset alone due to the small number of solvers. By combining solvers of Zoe riddle from Studies 1 and 2, we investigated whether solving the riddle with a Eureka (i.e., insight problem-solving) was associated with higher SemNets local changes than solving it without a Eureka. Among the 37 solvers, 29 reported a Eureka when solving Zoe riddle and seven did not (one missing data). Using the same methodology as before, we ran nonlinear mixed-effect models to explore whether global or local SemNets changes were associated with insight problem-solving (all statistical results are in Table 4).

We found a significant positive interaction effect of ΔMetric and *semantic distance* on insight solving for *weight* ($\beta = 0.10$, $p = 0.004$, CI95 = [0.03;0.16]), *clustering coefficient* ($\beta = 0.67$, $p = 0.03$, CI95 = [−0.02; 1.35]), and *eigenvector centrality* ($\beta = 0.25$, $p = 0.04$, CI95 = [0.01;0.49]), but not for *efficiency* ($\beta = 0.01$, $p = 0.73$, CI95 = [−0.05;0.08]). In addition, we found a significant positive, triple interaction effect of ΔMetric, *semantic distance*,

**Table 4 | Results of nonlinear mixed model analyses in Study 2 (insight problem-solving)**

| | | β | SE | z value | *p* value | CI 95% |
|---|---|---|---|---|---|---|
| Weight | ΔMetric | 0.01 | 0.09 | 0.15 | 0.88 | −0.17;0.20 |
| | IR | 9.00e−03 | 0.03 | 0.03 | 0.97 | −0.06;0.06 |
| | SD | −3.13e−03 | 0.03 | −0.11 | 0.91 | −0.06;0.05 |
| | ΔMetric × IR | −0.04 | 0.03 | −1.11 | 0.27 | −0.10;0.03 |
| | ΔMetric × SD | 0.10 | 0.03 | 2.86 | **4.21e−03** | 0.03;0.16 |
| | IR × SD | 3.42e−03 | 0.03 | 0.13 | 0.89 | −0.05;0.05 |
| | ΔMetric × IR × SD | −0.03 | 0.03 | −1.09 | 0.27 | −0.09;0.03 |
| Efficiency | ΔMetric | 0.06 | 0.20 | 0.31 | 0.75 | −0.33;0.47 |
| | IR | 4.31e−03 | 0.03 | 0.15 | 0.88 | −0.05;0.06 |
| | SD | 3.15e−03 | 0.03 | 0.11 | 0.91 | −0.05;0.06 |
| | ΔMetric × IR | −0.04 | 0.04 | −1.19 | 0.23 | −0.11;0.03 |
| | ΔMetric × SD | 0.01 | 0.03 | 0.34 | 0.73 | −0.05;0.08 |
| | IR × SD | 0.01 | 0.03 | 0.41 | 0.68 | −0.04;0.06 |
| | ΔMetric × IR × SD | −0.07 | 0.03 | −2.17 | **0.0297** | −0.12;−0.006 |
| Clustering coefficient | ΔMetric | −0.33 | 0.87 | −0.38 | 0.71 | −2.29;1.62 |
| | IR | 0.04 | 0.17 | 0.25 | 0.80 | −0.31;0.39 |
| | SD | −6.10e−03 | 0.17 | −0.04 | 0.97 | −0.35;0.34 |
| | ΔMetric × IR | −0.32 | 0.31 | −1.03 | 0.30 | −1.00;0.36 |
| | ΔMetric × SD | 0.67 | 0.31 | 2.14 | **0.0321** | −0.02;1.35 |
| | IR × SD | 0.13 | 0.18 | 0.75 | 0.45 | −0.23;0.50 |
| | ΔMetric × IR × SD | 0.07 | 0.33 | 0.21 | 0.83 | −0.59;0.78 |
| Eigenvector centrality | ΔMetric | 0.04 | 0.10 | 0.39 | 0.70 | −0.17;0.25 |
| | IR | −0.03 | 0.12 | −0.26 | 0.80 | −0.28;0.21 |
| | SD | 0.03 | 0.12 | 0.25 | 0.81 | −0.21;0.28 |
| | ΔMetric × IR | −0.04 | 0.12 | −0.34 | 0.73 | −0.29;0.20 |
| | ΔMetric × SD | 0.25 | 0.12 | 2.05 | **0.04** | 0.01;0.49 |
| | IR × SD | 2.69e−03 | 0.13 | 0.02 | 0.98 | −0.26;0.26 |
| | ΔMetric × IR × SD | 0.04 | 0.13 | 0.32 | 0.75 | −0.21;0.29 |

ΔMetric represents the difference between the considered metric between the PreSemNet and the PostSemNet. *Impact rating* (IR) corresponds to the importance of edges or nodes for solving the problem based on independent assessment. *Semantic distance* (SD) represents the semantic remoteness of edges or nodes based on all participants' baseline pre-RJT before the presentation of the problem. We ran a mixed model for each metric to predict if the problem was correctly solved with or without an insight measured as the Eureka report. Significant results (*p* < 0.05) are in bold.

and *impact rating* on insight solving for *efficiency* (β = −0.07, *p* = 0.03, CI95 = [−0.12;−0.01]). As interaction effects of ΔMetric × *semantic distance* and ΔMetric × *impact rating* for *efficiency* on insight solving were in opposite directions (positive for *semantic distance*, β = 0.01 and negative for *impact rating*, β = −0.04), the significant three-way interaction effect suggests that the type of SemNets changes differently impacted insight solving (although the two-way interactions were not significant: *p* = 0.73 for ΔMetric × *semantic distance* and *p* = 0.23 for ΔMetric × *impact rating*; Fig. 3A). The three-way interaction effect was not statistically significant for the other metrics (*weight*: β = −0.03, *p* = 0.27, CI95 = [−0.09;0.03]; *clustering coefficient*: β = 0.07, *p* = 0.83, CI95 = [−0.59;0.78]; and *eigenvector centrality*: β = 0.04, *p* = 0.75, CI95 = [−0.21;0.29]). These results indicate that insight problem-solving, as opposed to non-insight problem-solving, was associated with larger local SemNets changes.

In addition, we showed that insight problem-solving (defined by the report of a Eureka) was predicted by local remoteness-based SemNets changes, suggesting that bringing remote concepts closer together reflects a restructuring that promotes insight solving.

**Remoteness-based and solution-based SemNets changes dissociated active solving from solution exposure (Study 3)**

In Studies 1 and 2, we found that both solution- and remoteness-based SemNets changes were associated with solving a problem and an analogous one. We ran an additional study to test whether these two types of

SemNets changes were actually instrumental for solving the problem or whether they reflected the effect of solving itself (e.g., making solution-related associations more salient in the semantic networks due to a mere pre-exposure to the solution). We recruited 47 new participants who were asked to solve the Zoe riddle with the same design as in Study 2. At the end of the 10-min allotted time to solve the riddle, participants who failed to solve it were given the solution before completing the second RJT. We compared SemNets changes between participants who found the solution by themselves and those to whom it was given. Fifteen participants (32%, 14 with a Eureka) found the solution during the solving phase.

We used nonlinear mixed-effects models to explore if global or local SemNets changes differed between participants who found the solution by themselves (solver group) and those to whom the solution was given (solution group). To increase our statistical power that was limited by the low number of solvers (*n* = 10; 5 solvers excluded because they did not fit the criteria for SemNets building), we added in the solver group participants from Studies 1 and 2 who found the solution of the Zoe riddle by themselves (Study 1, *n* = 10; Study 2, *n* = 27), leading to a total of 47 participants in the solver group, and 25 in the solution group.

We found a significant negative interaction effect of ΔMetric and *impact rating* on problem-solving for *weight* (β = −0.05, *p* = 0.01, CI95 = [−0.09;−0.01]) and *efficiency* (β = −0.05, *p* = 0.04, CI95 = [−0.09;−0.002]). This result indicates that solution-based SemNets

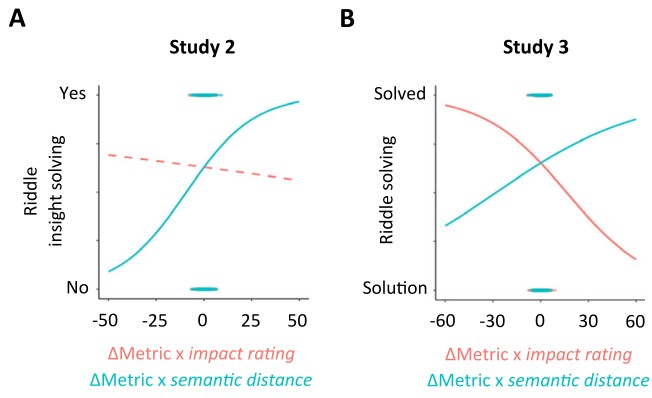

**Fig. 3 | *Impact rating* and *semantic distance* capture different SemNets changes related to problem-solving. A** Three-way interaction effects of ΔMetric, *impact rating*, and *semantic distance* on insight problem-solving (compared to non-insight problem-solving) for *efficiency* (Study 2). Graphs represent the ΔMetric weighted by the *impact rating* (ΔMetric × *impact rating* – solution-based restructuring, in red) or the ΔMetric weighted by the *semantic distance* (ΔMetric × *semantic distance* – remoteness-based restructuring, in blue) as a function of insight report when solving the riddle ($n = 7030$ observations). **B** Three-way interaction effects of ΔMetric, *impact rating*, and *semantic distance* on problem-solving (compared to non-solvers with solution) for *weight* (Study 3). Graphs represent the ΔMetric weighted by the *impact rating* (ΔMetric × *impact rating* – solution-based restructuring, in red) or the ΔMetric weighted by the *semantic distance* (ΔMetric × *semantic distance* – remoteness-based restructuring, in blue) as a function of solving success ($n = 13,680$ observations). Lines represent the fitting curves of the interaction effect (ΔMetric × *impact rating* in red, and ΔMetric × *semantic distance* in blue) in predicting insight problem-solving. A solid line indicates that a significant effect ($p < 0.05$) was found.

changes (i.e., based on the *impact rating*) were higher in the solution group compared to the solver group. This interaction effect was not statistically significant for *clustering coefficient* ($\beta = -0.13$, $p = 0.52$, CI95 = $[-0.57;0.31]$), or *eigenvector centrality* ($\beta = -0.11$, $p = 0.23$, CI95 = $[-0.28;0.07]$).

We also found a significant positive interaction effect of ΔMetric and *semantic distance* on problem-solving for *weight* ($\beta = 0.06$, $p = 0.02$, CI95 = $[0.01;0.10]$). Contrary to the previous result, this finding indicates that remoteness-based SemNets changes (i.e., based on the *semantic distance*) were higher in the solver group compared to the solution group. This interaction effect was not statistically significant for the other metrics (*efficiency*: $\beta = -7.56 \times 10^{-4}$, $p = 0.97$, CI95 = $[-0.05;0.05]$; *clustering coefficient*: $\beta = 0.13$, $p = 0.49$, CI95 = $[-0.28;0.55]$; and *eigenvector centrality*: $\beta = 0.03$, $p = 0.75$, CI95 = $[-0.14;0.19]$).

Importantly, we found a significant three-way interaction effect of ΔMetric, *semantic distance* and *impact rating* on problem-solving for *weight* ($\beta = -0.04$, $p = 0.048$, CI95 = $[-0.08;-0.0004]$). This result suggests a dissociation in how participants change their semantic associations between the two groups (Fig. 3B). Participants who found the solution by themselves showed remoteness-based SemNets changes, bringing closer remote semantic associations. In contrast, participants who were given the solution showed solution-based SemNets changes, with their SemNets resembling the optimal problem-solution representation. This three-way interaction effect was not statistically significant for the other metrics (*efficiency*: $\beta = -0.02$, $p = 0.43$, CI95 = $[-0.06;0.02]$; *clustering coefficient*: $\beta = 0.20$, $p = 0.34$, CI95 = $[-0.26;0.65]$; and *eigenvector centrality*: $\beta = -0.04$, $p = 0.68$, CI95 = $[-0.22;0.14]$) (all statistical analyses are available in Table 5). An illustration of the results is provided for summarizing our findings (Fig. 4), showing the different patterns of SemNets changes according to the group (non solver, solver, and solution groups), the word pair relevance for problem-solving (*impact rating*), and the word pair semantic remoteness (*semantic distance*).

## Discussion

We explored how semantic memory restructuring,—supposedly reflecting a reorganization of the mental representation of the riddles,—could serve as a cognitive mechanism underlying problem-solving. By building individual SemNets before and after solving a problem, we characterized and quantified the changes in the organization of semantic associations between riddle-related concepts. We found that local SemNets changes were related to successful problem-solving (see Table 6 for a summary). In addition, we found that similar local SemNets changes were also associated with the solving success of a semantically distinct, analogous problem, suggesting a link between restructuring and analogical transfer (Studies 1 and 2). We also showed that solving with insight was related to a remoteness-based restructuring (Studies 1 and 2). Finally, we demonstrated that solution-based and remoteness-based SemNets changes dissociated participants who solved the problem by themselves from those who were given the solution (Study 3). Together, our findings provide evidence for the role of local semantic memory restructuring as a cognitive mechanism of problem-solving and analogical transfer.

Previous research has demonstrated that computational network science methodology allows exploring how concepts are organized in individual minds via SemNets[41,43,44,47]. Investigating global semantic memory structure using SemNets has recently been validated as a tool to empirically explore individual differences in creative abilities and behavior[48-53,59-61]. In contrast with these previous works, here we built problem-specific SemNets and developed an original SemNet approach to quantitatively measure changes that reflected restructuring of the problem representation. SemNets changes associated with problem-solving success were related to local effects that focused separately on both solution-based and remoteness-based concepts (Studies 1 and 2). We found that successful problem-solving was linked with a local increase in *efficiency and weight* in the edges between words relevant to the solution and words that were initially semantically distant. In studies 1 and 2, we did not find three-way interaction effects that might have suggested a synergic or opposite effect of solution-based and remoteness-based SemNets changes on problem solving success. These results may suggest that the restructuring of solution-relevant words may not necessarily involve the most semantically remote ones explaining the missing three-way interaction. However, our significant two-way interaction effects may reflect that solution-based and remoteness-based SemNets changes captured distinct processes. These findings can be interpreted in two, not mutually exclusive ways. First, solution-based SemNets changes could suggest a restructuring that better integrates the solution into the mental representation of the problem, while remoteness-based changes indicate a restructuring that brings closer together concepts that were initially remote. Alternatively, these SemNets changes could reflect a restructuring that moves away word pairs that were solution-irrelevant and word pairs that were initially close. Indeed, we could hypothesize that strong but misleading word associations should be inhibited in order to create a new link[11,78-80] and solve the problem. The inhibition of non-relevant and obvious associations of ideas (that often lead to a mental impasse) has been proposed as the first necessary step to initiate restructuring[78,81]. Further work is needed to characterize more comprehensively the cognitive processes underlying the restructuring evidenced in our study.

Our work complements and extends a pioneering study[12], which used a pathfinder scaling algorithm to show at the group level that the SemNets of participants who solved the Bar riddle was different from the SemNets of participants who did not solve the riddle, and from the SemNets of those who were given the solution. Our findings highlighted that the restructuring of a problem-related SemNets was associated with the solving of an independent analogous problem. It is unlikely that the solving of the analogous problem was impacted by performing the RJT since each problem and its analog belong to distinct semantic fields. It is also unlikely that SemNets changes related to the naive problem depended on working on the analogous one since the analogous problem was presented after the two RJTs (and naive problem; see Fig. 1A). Thus, solving the analogous problem could not

**Table 5 | Results of nonlinear mixed model analyses in Study 3**

| | | β | SE | z value | *p* value | CI 95 |
|---|---|---|---|---|---|---|
| Weight | ΔMetric | −0.05 | 0.07 | −0.65 | 0.51 | −0.19 0.10 |
| | IR | 6.37e−03 | 0.02 | 0.32 | 0.75 | −0.03 0.05 |
| | SD | 0.02 | 0.02 | 0.85 | 0.39 | −0.02 0.06 |
| | ΔMetric × IR | −0.05 | 0.02 | −2.44 | **0.0148** | −0.09 −0.01 |
| | ΔMetric × SD | 0.06 | 0.02 | 2.30 | **0.0214** | 0.008 0.10 |
| | IR × SD | 4.61e−03 | 0.02 | 0.26 | 0.80 | −0.03 0.04 |
| | ΔMetric × IR × SD | −0.04 | 0.02 | −1.98 | **0.0479** | −0.08 −0.0004 |
| Efficiency | ΔMetric | −0.05 | 0.11 | −0.46 | 0.64 | −0.29 0.18 |
| | IR | 0.02 | 0.02 | 0.78 | 0.44 | −0.02 .06 |
| | SD | 8.69e−03 | 0.02 | 0.42 | 0.68 | −0.03 .05 |
| | ΔMetric × IR | −0.05 | 0.02 | −2.04 | **0.0417** | −0.09 −0.002 |
| | ΔMetric × SD | −7.56e−04 | 0.02 | −0.03 | 0.97 | −0.05 0.05 |
| | IR × SD | 0.01 | 0.02 | 0.61 | 0.54 | −0.02 0.05 |
| | ΔMetric × IR × SD | −0.02 | 0.02 | −0.78 | 0.43 | −0.06 0.02 |
| Clustering coefficient | ΔMetric | −0.69 | 0.54 | −1.28 | 0.20 | −1.88 0.41 |
| | IR | 0.02 | 0.12 | 0.19 | 0.85 | −0.22 0.26 |
| | SD | 0.03 | 0.11 | 0.30 | 0.76 | −0.20 0.27 |
| | ΔMetric × IR | −0.13 | 0.20 | −0.64 | 0.52 | −0.57 0.31 |
| | ΔMetric × SD | 0.13 | 0.19 | 0.69 | 0.49 | −0.28 0.55 |
| | IR × SD | −0.02 | 0.12 | −0.20 | 0.84 | −0.28 0.23 |
| | ΔMetric × IR × SD | 0.20 | 0.21 | 0.95 | 0.34 | −0.26 0.65 |
| Eigenvector centrality | ΔMetric | 0.03 | 0.07 | 0.46 | 0.65 | −0.11 0.18 |
| | IR | 2.68e−03 | 0.09 | 0.03 | 0.98 | −0.17 0.18 |
| | SD | −8.65e−04 | 0.09 | −0.01 | 0.99 | −0.17 0.17 |
| | ΔMetric × IR | −0.11 | 0.09 | −1.21 | 0.23 | −0.28 0.07 |
| | ΔMetric × SD | 0.03 | 0.08 | 0.32 | 0.75 | −0.14 0.19 |
| | IR × SD | 0.01 | 0.09 | 0.12 | 0.91 | −0.17 0.19 |
| | ΔMetric × IR × SD | −0.04 | 0.09 | −0.42 | 0.68 | −0.22 0.14 |

ΔMetric represents the difference between the PreSemNet and the PostSemNet for the considered metrics. *Impact rating* (IR) corresponds to the importance of edges or nodes for solving the problem based on independent assessment. *Semantic distance* (SD) represents the semantic remoteness of edges or nodes based on all participants' baseline pre-RJT before the presentation of the problem. We ran a mixed model for each metric to predict whether participants solved the problem by themselves or were given the solution after failing to do so. Significant results ($p < 0.05$) are in bold.

be driven by a mere semantic priming or post-solution memory triggered by the naive riddle concepts or its related RJT. Instead, our results on analogous problems support our hypothesis that restructuring preceded solving and are consistent with the mechanistic role of restructuring in analogical transfer, which had been proposed by several studies[19,24,26–29]. Restructuring of the initial problem may lead to the formation of an abstract representation of the problem and its solution, which may in turn facilitate the analogical transfer[13–17].

In addition, we also replicated the classical analogy transfer effect shown in previous studies[18–23] with new riddles that are different from the ones used in the classic tumor-fortress problem paradigm[18–22]. Our design allowed us to identify a novel behavioral signature of analogy transfer, evidenced by shorter solving times for the analogous compared to the naive problem. Our findings converge with the literature suggesting that transfer to novel problems depends on the structure of the mental representation of the initial problem. Our results further extend this literature by providing empirical evidence that this restructured representation better integrates concepts related to the solution and/or bridges closer previously remote elements of the problem. Overall, future studies are needed to clarify the mechanisms of analogy transfer and elucidate whether the specific SemNets changes that we measured led to the formation of an analogy schema or an abstract representation that facilitated analogical transfer, or whether these SemNets changes reflect a higher individual ability to reorganize one's semantic associations.

Our work extends Durso et al.'s results[12] by several other means. We tested four different riddles instead of one and showed, using nonlinear mixed models, that the results generalize across the four riddles. Importantly, we provided strong evidence of the relationship between the ability to solve the four riddles and other creative abilities, suggesting that our riddles capture some aspects of creative ability and can be considered as creative problems. Another major difference is that we captured intra-individual changes within the SemNets, instead of comparing the average SemNets of solvers and non-solvers. It allowed us to consider changes in SemNets local properties as a marker of restructuring. Finally, we used several classical network metrics that provide valuable information about how individual SemNets structures were modified during the problem-solving process. Our study demonstrates that a problem-related restructuring occurs locally in specific parts of the SemNets. We introduced two variables that characterized each node and edge in the SemNets regarding their relevance for solving the riddle (i.e., *impact rating*), or their semantic remoteness (i.e., *semantic distance*). Both solution-based and remoteness-based SemNets changes were individually associated with problem-solving, suggesting that participants who solved the riddle showed SemNets changes toward a solution-relevant problem representation and brought closer together semantically remote problem-related concepts. Additionally, we distinguished the local SemNets changes depending on whether participants found the solution themselves (solver group) or were given the solution before the post-RJT (solution group). The solver group showed remoteness-

**Fig. 4 | SemNets changes at the group level regarding *impact rating* and *semantic distance* of word pairs.** Each row represents a word pair (*n* = 190) that are sorted by increasing *impact rating* (**A**) or *semantic distance* (**B**). ΔWeight are averaged across participants according to the group (non solver, solver, or solution groups). We combined the three study and used only the Zoe riddle. Color code indicates averaged ΔWeight value from negative (in red) to positive value (in blue). We used *weight* for this figure as it is less transformed (and in turn more easily understandable) metric.

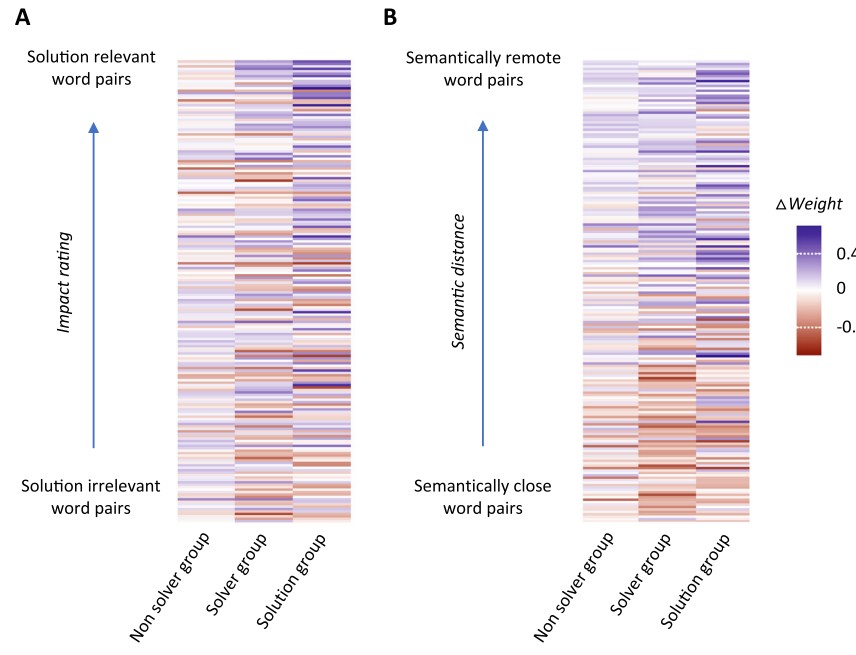

## Table 6 | Summary of the results from the three studies

| | | Solution-based restructuring (ΔMetric × *IR*) | Remoteness-based restructuring (ΔMetric × *SD*) | |
|---|---|---|---|---|
| Exp. 1 | PS | W, Eff, EV | Eff | ▪Strengthening/centralizing high-*IR* edges/nodes is positively related to PS.<br>▪Strengthening high-*SD* edges is positively related to PS. |
| | APS | W, Eff | W, Eff | ▪Strengthening high-*IR* or high-*SD* edges is positively related to APS. |
| Exp. 2 | PS | CC | W, Eff | ▪Interconnecting high-*IR* nodes is positively related to PS.<br>▪Strengthening high-*SD* edges is positively related to PS. |
| Exp. 1 + 2 | IPS | - | W, CC, EV | ▪Strengthening/interconnecting/centralizing high-*SD* edges/nodes is positively related to IPS. |
| Exp. 3 | PS | - | W | ▪Strengthening high-*SD* edges is positively related to PS. |
| | Solution | W, Eff | - | ▪Strengthening high-*IR* edges is positively related to giving the solution<br>▪These effects for high-SD and high-IR edges were significantly opposite. |

For each analysis, we report the dependent variables (PS: naive problem-solving, APS: analogous problem-solving, IPS: insight problem-solving) and the metrics (W: *weight*, Eff: *efficiency*, CC: *clustering coefficient*, EV: *eigenvector centrality*) for which the interaction effect of ΔMetric and *impact rating* (IR, solution-based restructuring) or △Metric and *semantic distance* (SD, remoteness-based restructuring) was positive and significant.

based SemNets changes suggesting that this type of SemNets changes reflected an active process of combining remote semantic concepts in memory, which was instrumental for successful problem-solving. This interpretation echoes associative theories of creativity[54,56]. In contrast, in the solution group, we observed solution-based SemNets changes that better align solution-relevant concepts with the problem representation and can reflect a prior exposure to the solution and/or the activation of a category of solutions. Thus, our findings support a dynamic view of semantic memory[57] and differentiate between active and passive dynamics of semantic memory (see also[58]).

Our results provide empirical support for an oft-cited theory in the creativity literature, which postulates that the restructuring of semantic associations pertaining to a given problem predicts the successful solving of this problem[3–5,9]. Such restructuring idea has often been associated with the notion of insight[38,82,83], i.e., when the solution to a problem comes suddenly and effortlessly. A seminal theory has proposed that cognitive insight may entail the forging or changing of nodes and/or edges in one's semantic memory network[7,8,11,35,78]. Such changes supposedly create a 'shortcut' in the network that allows decreasing the distance between different

representations and results in a cascade of new connections. Here, we found that problem-solving with insight was linked with a local increase in *efficiency, clustering,* and *weight* in the edges and nodes that are initially semantically distant. This finding substantiates Schilling's hypothesis of a fast decrease in path length within individual's network representations when solving a problem with insight .

## Limitations
Some limitations must be mentioned. First, our methodology quantifying SemNets restructuring at two different time points did not allow us to characterize the full mechanisms leading to SemNets changes, nor to provide a full picture of the dynamic changes occurring during creative problem-solving and whether these changes represented long-term structural changes.

Second, one might argue that some word pairs during the RJT primed the riddle's solution and gave participants hints. This is unlikely given the low solving rate and since only one participant among the two first samples (*n* = 250) found the solution after the post-RJT (0.4%). In addition, among the 190 word pairs used during the RJT, most of them were either neutral or

https://doi.org/10.1038/s44271-024-00100-w **Article**

misleading for solving the riddle (see distributions of *impact rating* in Figs. S2 and S3), making helpful word pairs less salient.

Third, the solving rate of the riddles in the *naive* condition was overall low. It may be partially due to the difficulty of the riddle or to the additional cognitive load induced by participants having to report their ideas in real time and answering questions about their attentional focus during the reflection period. The low solving rate led to an unbalanced statistical design that may have impacted our results. However, our main results (Study 1) were replicated in an independent larger study (Study 2), reinforcing their reliability. In addition, we checked that the *balanced accuracy* of significant models was high, indicating that our models remained powerful when considering the unbalanced design.

Fourth, we only explored problems that are supposed to involve restructuring to be solved. Further studies with a larger variety of riddles, including a control riddle that does not require restructuring, would be helpful to conclude on the specificity of SemNets restructuring for creative problem-solving.

Fifth, it is not entirely clear why some SemNets metrics reached significance in predicting solving success and others did not (see Table 6). The differences in the results of Studies 1 and 2 could be explained because *semantic distance* was computed based on a larger dataset for the Zoe riddle and thus may be more reliable than for the other riddles. In most of our analyses, significant results were mainly found for edge metrics (*weight* and *efficiency*), and less consistently for node metrics (*eigenvector centrality* and *clustering coefficient*). This might be due to a lower statistical power for node metrics, which were based on far less data points ($n = 20$ points/participant) than edge metrics ($n = 190$ point/participants). However, we found significant prediction of insight solving (versus non-insight solving) mainly with node metrics (*clustering coefficient* and *eigenvector centrality*) whereas significant prediction of solving (versus non-solving) was significant with edge metrics (*efficiency*). This may indicate that our metrics capture different cognitive processes. Although SemNets are increasingly used in cognitive neuroscience, their exact meaning for human cognition is still uncertain. Further research is needed to directly explore if and how distinct SemNets metrics capture specific aspects of cognitive processing.

Finally, a critical question that our findings leave open relates to the causal link between restructuring and solving. We provided evidence that remoteness-based SemNets changes are involved in solving a future analogous problem and differentiate an active solving of the problem from a mere exposure to its solution (while solution-based SemNets changes do not). Even if the current study design does not address the causality question, these arguments support the idea that at least remoteness-based restructuring precedes solving and cannot be explained by a mere semantic priming of the solution.

## Conclusion

In this work, we directly related the restructuring of mental representations and complex human behavior, focusing on successful problem-solving and analogical transfer. Restructuring was operationalized as the changes in the organization of semantic associations using SemNets metrics. This approach allowed us to demonstrate that the local restructuring of problem-related semantic representations was associated with the successful solving of this problem. Additionally, local SemNets changes were also associated with the solving success of a semantically unrelated, analogous problem, suggesting its potential role in analogical transfer. SemNets changes differed depending on whether the participants solved the problem or were given the solution, or whether they solved the problem with insight or not. Together, our findings indicate that network science measures of SemNets changes capture a restructuring of semantic representations that are critical to problem-solving. The approach that we proposed to quantify restructuring could be used in further studies to explore its neural correlates and to test whether factors that have been shown to modulate problem-solving success, such as sleep[84–86], influence restructuring. Beyond problem-solving, our method could be converted as a tool to measure the restructuring of mental representations in various situations (e.g., when some mental associations are harmful, such as in phobia).

## Data availability

The study reported in this article was not formally preregistered. Experimental material is available in the Supplementary Information associated with this article. The datasets generated and/or analyzed during the current study (main manuscript and Supplementary Information) is available in a public persistent repository (https://osf.io/pk9nc/?view_only= 4c4432106624412391bb4f1016a79ac5). Data sharing will be anonymized and will not include participants' demographic information.

## Code availability

We used open software and toolboxes available online: semantic networks metrics were computed using the Brain Connectivity Toolbox (BCT)[68], and statistical analyses were done using Rstudio. We specified the statistical functions that we used in the *Methods* section. Scripts written for statistical analyses and to generate figures (main manuscript and Supplementary Information) are available in a public persistent repository (https://osf.io/pk9nc/?view_only=4c4432106624412391bb4f1016a79ac5).

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

## Acknowledgements

This research was partly funded by the 'Agence Nationale de la Recherche' [grant number ANR-19-CE37-001-01], the 'Fondation pour la recherche medicale' [grant number DEQ20150331725], and from the program "Investissements d'avenir" [grant number ANR-10- IAIHU-06]. TB also received funding from the 'Assistance Publique des Hôpitaux de Paris'. MOT is funded by Becas-Chile of ANID (CONICYT). CL was funded by the Doctoral school ED3C and 'Société Française de Recherche et Médecine du Sommeil' (SFRMS). The funders had no role in study design, data collection, and analysis, decision to publish or preparation of the manuscript. Part of this work was carried out in the PRISME facility of ICM. We gratefully acknowledge Karim N'diaye for his help in the data collection. We thank all the participants in the study and Clothilde Chappé for piloting part of the material used in this study.

## Author contribution

Conceptualization: Y.N.K., D.O., E.V.; Methodology: T.B., Y.N.K., M.O.T., A.L.P., C.L., D.O., E.V.; Investigation: T.B., M.S., I.M., J.S.; Visualization: T.B.; Supervision: D.O., E.V.; Writing: T.B., Y.K., D.O., E.V.

## Competing interests

The authors declare no competing interests.
