## [Peer Review File · Communications Psychology]

26th Jan 23

Dear Dr Bieth,

Thank you for your patience during the peer-review process. Your manuscript titled "Dynamic changes in semantic memory structure support successful problem-solving and analogical transfer" has now been seen by 2 reviewers, whose comments are appended below. You will see that they find your work of some potential interest. However, they have raised quite substantial concerns that must be addressed. In light of these comments, we cannot accept the manuscript for publication, but would be interested in considering a revised version that fully addresses these serious concerns.

We hope you will find the Reviewers' comments useful as you decide how to proceed. Should additional work allow you to address these criticisms, we would be happy to look at a substantially revised manuscript. If you choose to take up this option, please highlight all changes in the manuscript text file, and provide a detailed point-by-point reply to the reviewers.

Editorially, we consider the referees' concerns about the lack of control conditions to be of the highest priority. We would only be able to consider a revised manuscript for further peer review if you provided additional empirical data to satisfy these requests for suitable control conditions. Ideally, these additional experiments would need to be preregistered, with a prior commitment to and justification of the selection of outcome measures that will be chosen to evaluate your hypothesis. Finally, we highlight the referees' joint calls for additional analyses to test your interpretation and the request to place the contribution your work makes appropriately into the existing literature.

I include below a list of requests that the journal places on all revised manuscripts and highlight in particular our requirements for statistics and statistics reporting (<https://www.nature.com/commspsychol/submit/submission-guidelines#statistical-guidelines>). Please ensure that your revised work fully complies with these guidelines.

If the revision process takes significantly longer than five months, we will be happy to reconsider your paper at a later date, provided it still presents a significant contribution to the literature at that stage. We would appreciate it if you could keep us informed about an estimated timescale for resubmission, to facilitate our planning. Of course, if you are unable to estimate, we are happy to accommodate necessary extensions nevertheless.

Please use the following link to submit your revised manuscript, point-by-point response to the Reviewers' comments with a list of your changes to the manuscript text (which should be in a separate document to any cover letter) and any completed checklist:

[link redacted]

Please do not hesitate to contact me if you have any questions or would like to discuss the required revisions further. Thank you for the opportunity to review your work.

Best regards,

Marike Schiffer

Marike Schiffer, PhD
Chief Editor
Communications Psychology

EDITORIAL POLICIES AND FORMATTING

Editorial Policy: Policy requirements (Download the link to your computer as a PDF.)

Furthermore, please align your manuscript with our format requirements, which are summarized on the following checklist:

Communications Psychology formatting checklist

and also in our style and formatting guide Communications Psychology formatting guide .

* **CODE AVAILABILITY:** All Communications Psychology manuscripts must include a section titled "Code Availability" at the end of the methods section. In the event of publication, we require that the custom analysis code supporting your conclusions is made available in a publicly accessible repository; please choose a repository that provides a DOI for the code; the link to the repository and the DOI must be included in the Code Availability statement. Publication as Supplementary Information will not suffice. We ask you to prepare and upload code at this stage, to avoid delays later on in the process.

* DATA AVAILABILITY:

All Communications Psychology research manuscripts must include a section titled "Data Availability" at the end of the Methods section or main text (if no Methods). More information on this policy, is available at <http://www.nature.com/authors/policies/data/data-availability-statements-data-citations.pdf>.

At a minimum the Data availability statement must explain how the data can be obtained and whether there are any restrictions on data sharing. Communications Psychology strongly endorses open sharing of data. If you do make your data openly available, please include in the statement:

We recommend submitting the data to discipline-specific, community-recognized repositories, where possible and a list of recommended repositories is provided at <http://www.nature.com/sdata/policies/repositories>.

If a community resource is unavailable, data can be submitted to generalist repositories such as [figshare](https://figshare.com) or [Dryad Digital Repository](https://www.dryad.org). Please provide a unique identifier for the data (for example a DOI or a permanent URL) in the data availability statement, if possible. If the repository does not provide identifiers, we encourage authors to supply the search terms that will return the data. For data that have been obtained from publicly available sources, please provide a URL and the specific data product name in the data availability statement. Data with a DOI should be further cited in the methods reference section.

REVIEWER EXPERTISE:

The reviewers are expert in cognitive psychology with a focus on problem solving/reasoning and methodological expertise in computational analysis approaches, including network analysis.

Reviewer #1 (Remarks to the Author):

Bieth and colleagues investigate one sub-process of (insightful) problem-solving called restructuring (part 1) in combination with analogical reasoning (part 2). To do this, the authors use verbal riddles and investigate to what degree the subjects' semantic network changes are a function of restructuring those riddles. They construct a subject's semantic network before and after the riddle solving attempt by having them rate the conceptual similarity between key words of the riddle, the solution and neutral words. Subsequently they construct several network measures (e.g. clustering coefficient) from those individualized conceptual networks and define restructuring as the difference of those measures from pre to post solution. They find that those network measures in interaction with a specific weighting of the most relevant nodes and edges (which was independently rated = impact rating) predict successful problem solving as well as the likelihood to solve a structurally similar problems via analogical reasoning.

In general the manuscript is well written (but see points below). Finding new measures to quantify restructuring is highly relevant as there is still little research as the authors also point out. However, I have my concerns to what degree the study actually measures restructuring. The biggest issue that puts their interpretation of the results into question is the fact that they fail to include a control condition that measures potential changes in network measures for problems that had to be solved and are structurally similar to the riddles but do not require restructuring – one possible example may be word analogy problems (football:field = swimmer: ? [this is just a first idea for demonstration purposes, I'm sure there are much better examples]). Without this control condition the difference in network measures that they find could merely be because of having solved the riddles but not due to restructuring. For example, a person that solved the man-in-the-bar riddle will very likely rate hick-ups and shotgun as more similar simply because s/he just solved the riddle and still mentally clearly represents the connection between both concepts and also puts attention to this relationship thanks due to recent solving. Hence, this may be some kind of priming, an increased attention or simply a solving-effect (all three things seem related but are not restructuring per se). Hence, the authors would have to show that solving e.g. the analogy problem does not lead to an increased semantic relationship between e.g. football and swimmer or field and pool to make the claim that they measured restructuring in the riddles.

This is specifically relevant because the authors criticize the so-called circular arguments in classical insight problem solving studies that define restructuring as the ability to solve insight tasks because they require restructuring. Ironically, Bieth and colleagues end up doing the same because restructuring the problem with their current approach cannot be differentiated from solving the problem.

I still think this is an interesting approach but given this heavy caveat all the authors can do at this point is to row back on their interpretation and rather claim that they find differences in network measures that correlate with problem solving.

There are some other issues that will be listed below:

- As the authors already mention in the limitation section the amount of overall problems that were solved is very very low for both experiments, which makes this method a bit questionable in general (i.e. whether those kind of riddles (including this kind of difficult level) are appropriate to investigate restructuring given most people don't show the desired behavior [restructuring]). Furthermore, the priming of the key words to solve the task thanks to the pre-rating may also be a major issue that would require controlling for.
- There are several network measures, the authors seem to have chosen only a subset. Was the study preregistered using those network measures? If not what about other common measures such as: Cost, Path length, Global/local efficiency, betweenness centrality? If they were not preregistered, the chosen network measures need to be controlled for multiple comparison as you expect an effect in any of those measures given the same interaction between impact rating and metric.
- The chosen statistical approach in general seems appropriate. However, the cleaner way to model the data is to do a mixed effects model where the tasks are not merged but this information is explicitly given to the model via fixed or random effects: Hence, given your design I would have expected that you model all available data together (for each outcome measure), i.e. you'd have an additional variable for the task (riddle1, riddle2) and whether the task is naïve, directed or spontaneous transfer.
- What about other creativity tasks? Does your metric*impact rating interaction effect survive when you model at least one creativity task in your design?
- What about the insight measures? They were not introduced in the introduction section but are the analyzed and reported in the methods/results section. According to the insight literature, insight is often a consequence of restructuring (Danek et al. etc). To further test your hypotheses that the

metric*impact rating interaction represents restructuring one would expect a three-way interaction with insight, i.e. the metric*impact rating interaction should be most predictive of solving the task when the participants had experienced an insight.

- I understand that this is a highly exploratory study but it would be appropriate to make some assumptions about the underlying cognitive mechanisms regarding restructuring in the semantic network. In line 380/1 you claim that your approach may provide a mechanistic explanation of how individuals solve new problems. However, I failed to read any attempts to describe the assumed restructuring mechanism.

- The data seemed to have come from a bigger study but it was not disclosed (in the main text) which study this is and which papers have already been published with this data set. This is highly relevant for scientific transparency.

- Line 353-358: For the first two points you need to use Bayesian statistics or equivalence testing if you want to make actual claims about null effects (see Lakens et al., 2020).

- Other new approaches of studies investigating restructuring were not mentioned in the introduction (e.g. Wu et al., 2013; Danek et al. 2014; Di Bernardi Luft et al., 2018; Becker et al. 2020) and should also be discussed in relation to the current study

Other minor issues:

- Language: there are a few sentences where a word is missing or there some other language related issues (e.g. line 81:82)

- What is RJT? (line 163). The authors should describe what those letters stand already the first time they use it in the results section.

- Is the data publically available? If yes, where?

- How did you chose which words are relevant from each task?

- Line: 236/237: The authors did not show that solving the riddles involves mostly insight solving -> all they showed was the classical effect that correct responses are more often associated with Eureka experiences than incorrect ones. That does not show how often people actually solved those tasks with insight.

Reviewer #2 (Remarks to the Author):

This paper described a study where participants solved 2 sets of riddle pairs (naïve and transfer conditions) and performed relatedness judgments on words from the two riddles (+ unrelated words) before and after the naïve riddle. Individual semantic networks were estimated based on these relatedness judgments. The results indicated that successful solving was associated with “restructuring”, as evidenced by changes in network metrics (weight, eigenvector centrality and efficiency). These changes also predicted transfer to the other riddle, and the effect was replicated in a new experiment to some extent (not for weight).

Overall, the paper is interesting and presents a quantitative method to assess how individuals solve problems from a representation perspective. However, I have some questions about novelty and the general framing/logic of the studies. Some of my questions are to do with reframing or acknowledging limitations, while others may need more work in terms of a follow-up control experiment to understand the processes involved in the task more deeply.

1. I believe there is related work by the authors on how networks change as a function of tasks in low/high creative individuals as well as via conceptual combinations. It made me wonder about two

things: (1) convergent vs. divergent validity, i.e., what are some tasks where you would not expect these network changes to occur? And (2) what is the novelty of the present work other than exploring the same methods in a new task? I do think there is merit in this work, but I think the paper could benefit from a clearer discussion of how this work is truly adding to the literature. Some of my other comments below elaborate on issue (1).

2. One concern I have is something the authors touch upon in the limitations but do not directly examine. What if the “restructuring” is simply a levels-of-processing/exposure effect and doesn’t actually reflect restructuring per se? For example, as the authors suggest, would you expect the same type of effect if the riddle did not require restructuring but still required the use of some concepts vs. others from the RJT? The current approach seems to lack a bit of specificity in terms of what it predicts and what counts as restructuring. Another control experiment would be to provide the participants with the solution immediately and then measure pre/post network changes – since they did not necessarily come up with the solution themselves, would you still expect their networks to have changed? Basically, I am unclear on whether the change in network metrics is truly a “restructuring” or simply an outcome of how some concepts may be more salient than others after having solved the riddle (which could be achieved without solving it as well, by mere exposure).

3. The transfer finding is interesting here – especially since the RJT did not include words from the analogous riddle, so there does appear to be something in the structure of the original riddle that the participants are picking up on – I wonder if there was a way to capture this abstraction / higher order process via networks as well, i.e., is it possible they did some type of mapping of concepts from one riddle to the other? If the participants did a combined RJT for riddles 1 and 2 after solving both riddles, there might be a way to capture this forging of new links potentially. Another possible control experiment to run here to make sure that transfer is truly happening as one might expect is to do the same study but pair up riddles that do not have a direct overlap (e.g., Zoe/Daniel) – would you expect changes in the network to predict solving rates for this unrelated riddle in the spontaneous or direct transfer conditions?

4. It is also important to address some questions from the process perspective, i.e., while estimates at the representation level could be informative, ultimately the problem-solving task involves some type of processing – what is this process and how would the current approach be able to explain this process? Even if we believe that the “efficiency” of the network changes, how does this happen? Given that there are approaches for simulating how activation spreads within a network (e.g., spreadR), there may be ways to model this process. The process modeling would also force the authors to be specific about their predictions and assumptions, e.g., what qualifies as a ‘eureka’ moment vs. not a ‘eureka’ moment, and what kinds of processes mediate these experiences, which ultimately cause restructuring in the network.

5. Finally, while the general network approach is interesting, the authors do not seem to motivate why some metrics might be expected to significantly predict solving success or transfer, and more importantly, which ones you wouldn’t expect to predict the same. For example, why would one expect efficiency to be predictive of restructuring but not weight? Overall, it seems a bit of a “kitchen sink” approach where a variety of metrics were examined without properly motivated hypotheses about the explicit measures.

We thank the Editor and the Reviewers for their overall positive assessment of our study, their constructive feedback, and their help improving our manuscript. We address all the comments in details below. Each comment is written in italic, our response is in plain text, and the citations from the revised manuscript are highlighted in blue. We also made minor changes to further improve the manuscript, all highlighted in yellow.

Reviewer #1

R1 General comments to the Author

Bieth and colleagues investigate one sub-process of (insightful) problem-solving called restructuring (part 1) in combination with analogical reasoning (part 2). To do this, the authors use verbal riddles and investigate to what degree the subjects' semantic network changes are a function of restructuring those riddles. They construct a subject's semantic network before and after the riddle solving attempt by having them rate the conceptual similarity between key words of the riddle, the solution and neutral words. Subsequently they construct several network measures (e.g. clustering coefficient) from those individualized conceptual networks and define restructuring as the difference of those measures from pre to post solution. They find that those network measures in interaction with a specific weighting of the most relevant nodes and edges (which was independently rated = impact rating) predict successful problem solving as well as the likelihood to solve a structurally similar problems via analogical reasoning.

In general the manuscript is well written (but see points below).

R1 Comment #1

Finding new measures to quantify restructuring is highly relevant as there is still little research as the authors also point out. However, I have my concerns to what degree the study actually measures restructuring. The biggest issue that puts their interpretation of the results into question is the fact that they fail to include a control condition that measures potential changes in network measures for problems that had to be solved and are structurally similar to the riddles but do not require restructuring – one possible example may be word analogy problems (football:field = swimmer:? [this is just a first idea for demonstration purposes, I'm sure there are much better examples]). Without this control condition the difference in network measures that they find could merely be because of having solved the riddles but not due to restructuring. For example, a person that solved the man-in-the-bar riddle will very likely rate hick-ups and shotgun as more similar simply because s/he just solved the riddle and still mentally clearly represents the connection between both concepts and also puts attention to this relationship thanks due to recent solving. Hence, this may be some kind of priming, an increased attention or simply a solving-effect (all three things seem related but are not restructuring per se). Hence, the authors would have to show that solving e.g. the analogy problem does not lead to an increased semantic relationship between e.g. football and swimmer or field and pool to make the claim that they measured restructuring in the riddles.

This is specifically relevant because the authors criticize the so-called circular arguments in classical insight problem solving studies that define restructuring as the ability to solve insight tasks because they require restructuring. Ironically, Bieth and colleagues end up doing the same because restructuring the problem with their current approach cannot be differentiated from solving the problem.

I still think this is an interesting approach but given this heavy caveat all the authors can do at this point is to row back on their interpretation and rather claim that they find differences in network measures that correlate with problem solving.

We thank Reviewer #1 for their helpful feedback. In our paper, we initially used the term ‘restructuring’ interchangeably to define: i) any changes in semantic network (SemNets) metrics after compared to before the problem presentation and ii) a cognitive change in the mental representation of the problem. We agree that equating changes in SemNets with a cognitive restructuring was debatable and that some of the changes that we observed could indeed be due to a pre-activation of the solution (even if our results on analogical transfer made this alternative hypothesis less likely). Thus, we took two major steps to better characterize what these changes in problem-related SemNets capture: whether they reflect a true restructuring of the problem (i.e., a change in the problem representation) or can be explained by a mere pre-exposure to the solution (i.e., priming effect).

First, we reframed our analysis plan with the aim of distinguishing solution-based changes in SemNets (that could reflect a priming effect) from associative changes that could be instrumental in finding the solution (‘true’ restructuring). To do so, we explored two types of local changes in SemNets: i) changes based on the *impact rating* (as in the first version of our paper), which might reflect an optimal problem representation as *impact rating* directly measures how useful an edge or a node is for solving the problem (solution-based restructuring); and ii) changes based on the *semantic distance* between nodes (remoteness-based restructuring). This *semantic distance* variable represents how much two words of a pair used in the RJT are inherently related, independently of the riddle. It was computed at the group level as the median of all participants’ baseline pre-RJT rating (i.e., before the presentation of the riddle) for each given pair (all studies combined, Zoe: n = 193; Bar: n = 34 participants; Car: n = 38; Daniel: n = 34 participants). In line with the associative theory of creativity, which states that insight and creative problem-solving imply combining elements related to the problem in a new way or creating novel and unexpected connections between remote concepts in semantic memory (Chu & MacGregor, 2011; Mednick, 1962; Schilling, 2005), we expected that solving the riddle would require to connect concepts that are usually semantically distant.

By adding both *impact rating* and *semantic distance* in the model, the common variance between the two variables is left out in the model residues, allowing us to distinguish changes in SemNets that are ‘purely’ driven by the solution (i.e., pre-exposure effect measured by the *impact rating*) and those that capture the creative restructuring process (i.e., combining semantically remote concepts, measured by the *semantic distance*).

We found that both solution- and remoteness-based SemNet changes were associated with solving a problem and an analogous one (Studies 1 and 2), suggesting that both pre-exposure and true restructuring explain our results.

Second, we ran a new, control experiment (Study 3). We explored whether changes in SemNets’ metrics differed whether participants found the solution by themselves or not. We expected that only the participants who solved the riddle themselves would show a remoteness-based restructuring. We collected data from 47 new participants. They followed the same experimental procedure as in Study 2, with one exception: at the end of the 10-min allotted time to solve the riddle, the solution was given to participants who failed to solve it

by themselves, before completing the second RJT. Hence here, the non-solver group became a ‘solution group’, which differs from solvers as they did not actively restructure the problem associations. We compared changes in SemNets between solvers and participants to whom the solution was given.

Interestingly, we observed two distinct types of SemNet changes whether the participants solved the problem by themselves or not. The solver group showed a remoteness-based restructuring, whereas the solution group showed a solution-based restructuring. This dissociation posited the restructuring as an active process involved in problem-solving and not only a result of a pre-exposure to the solution. These findings support theoretical studies that highlighted restructuring as a problem-solving mechanism involving combining unusual associations of ideas.

This new result seconds another strong argument already discussed in the manuscript, i.e., the relationship between problem-related restructuring and the solving of an analogous one. Here, changes in SemNets were measured before the presentation of the analogous riddle. Hence, SemNet changes related to the naive problem were not impacted by working on the analogous one. In addition, the riddle in the *transfer* condition did not have any semantic relationships with either the naive riddle or its riddle-specific word list. Solving of the analogous problem could not be driven by a mere semantic priming or post-solution memory of the riddle concepts.

We updated the Abstract, Introduction, Method, Results, and Discussion sections to reflect our novel analysis strategy and include Study 3. Overall, we think our findings and conclusions are greatly improved by these revisions and we thank the reviewers and editor for their suggestions.

Finally, as suggested by Reviewer #1 and as we discussed in the initial version of our manuscript, another interesting, control condition could use a problem that does not require restructuring to be solved. While such a ‘non-restructuring needed’ control condition would give valuable insights into the specificity of our SemNets measures, we favored the ‘solution given’ control condition because: i) it allowed us to dissociate pre-exposure vs. restructuring effect, using the exact same material, ii) choosing a problem that does not necessitate restructuring is not trivial as we do not know for sure which problem requires restructuring or not to be solved (the uncertainty about the mechanisms underlying successful problem solving was precisely the reason why we ran the current study), iii) even if the perfect ‘non-restructuring needed’ problem exists, we would need to adapt it for SemNets building (i.e., finding 20 words, related and unrelated to the problem), which is a long process, requiring multiple steps of piloting. To put more emphasis on this potential limitation, we added a sentence in the discussion (p.16):

“Fourth, we only explored problems that are supposed to involve restructuring to be solved. Further studies with a larger variety of riddles, including a control riddle that does not require restructuring, would be helpful to conclude on the specificity of SemNet restructuring for creative problem-solving.”

R1 Comment #2

As the authors already mention in the limitation section the amount of overall problems that were solved is very very low for both experiments, which makes this method a bit questionable

in general (i.e. whether those kind of riddles (including this kind of difficult level) are appropriate to investigate restructuring given most people don't show the desired behavior [restructuring]). Furthermore, the priming of the key words to solve the task thanks to the pre-rating may also be a major issue that would require controlling for.

We fully agree with Reviewer #1 that the low problem-solving rate led to an unbalanced statistical design that may have impacted our results. As replication is important in science, in particular in the context of a new method and novel results, we have made efforts to replicate our main result with Study 2. This replication was conducted in a larger and independent sample of new participants. With this successful replication, we believe that our conclusions are more robust and reliable. In addition, each significant model showed a high *balanced accuracy*, indicating that these models were powerful enough to predict problem-solving considering the unbalanced design. Finally, the results related to analogous problem-solving are much less concerned by this issue, as the groups are more comparable in terms of sample size. Finally, we believe that the additional control group provided by Study 3 further reinforces our conclusions.

To fully address the Reviewer's point, we revisited the limitation paragraph to emphasize the concern raised by Reviewer #1 (p.16):

“Third, the solving rate of the riddles in the *naive* condition was overall low. It may be partially due to the difficulty of the riddle or to the additional cognitive load induced by having to report their ideas in real-time and answering questions about their attentional focus during the reflection period. The low solving rate led to an unbalanced statistical design that may have impacted our results. However, our main results (Study 1) were replicated in an independent larger study (Study 2), reinforcing the reliability of the results. In addition, we checked that the *balanced accuracy* of significant models was high, indicating that our models remained powerful when considering the unbalanced design.”

We indeed did not control for a potential priming effect by the first RJT in Studies 1 and 2. However, Study 3 partially addresses this concern by comparing SemNet changes in solvers and non-solvers provided with the solution. Furthermore, even if we did not specifically control this effect in Studies 1 and 2, the issue is alleviated by several facts. First, the solving rate is low, suggesting that being exposed to the word list is not helpful. Second, only one participant among the two samples (n = 250) found the solution after the post-RJT (0.4%). Third, during the RJT, there were 190 word pairs, with the majority being either neutral or misleading for solving the riddle, thus ‘hiding’ the helpful pairs.

We highlighted these points in the discussion section related to limitation as follows (p.16):

“Second, one might argue that some word pairs during the RJT primed the riddle's solution and gave participants hints. This is unlikely given the low solving rate and since only one participant among the two *first* samples (n = 250) found the solution after the post-RJT (0.4%). In addition, among the 190 word-pairs used during the RJT, most of them were either neutral or misleading for solving the riddle (see distribution of *impact rating* in **Figures S4 and S5**), which make helpful word pairs less salient.”

R1 Comment #3

There are several network measures, the authors seem to have chosen only a subset. Was the study preregistered using those network measures? If not what about other common measures such as: Cost, Path length, Global/local efficiency, betweenness centrality? If they were not preregistered, the chosen network measures need to be controlled for multiple comparison as you expect an effect in any of those measures given the same interaction between impact rating and metric.

We agree with Reviewer #1 that many network measures exist and we understand that choosing only a subset could seem arbitrary. However, we carefully selected our metrics based on the creativity literature, specifically those implemented in the brain connectivity toolbox, to limit the number of statistical tests. As mentioned in the *Methods* section (p. 21-22), we used *efficiency* and *clustering coefficient* because these metrics have shown promising correlations with creativity measures in previous studies (Bernard et al., 2019a; Kenett et al., 2014a; Ovando-Tellez, Benedek, et al., 2022; Ovando-Tellez, Kenett, et al., 2022; Ovando-Tellez et al., 2023). We chose *eigenvector centrality* as the centrality measure, based on a previous study showing that *eigenvector centrality* has higher reliability in cognitive research compared to other centrality measures (Christensen et al., 2018). Finally, we used *weight* to have a less transformed (and in turn more easily understandable) metric.

While we initially considered the metrics proposed by Reviewer #1, we did not choose them for several reasons. *Path length* is redundant with *efficiency*, as *efficiency* is the inverse of *path length* (He et al., 2021). *Betweenness centrality* (or other measures of centrality such as *degree* or *closeness*) shows less reliability compared to *eigenvector centrality* (as developed before). *Cost* is not a metric implemented in the Brain Connectivity Toolbox.

That said, we did not preregister the study and understand Reviewer #1's concerns. Thus, we used a correction for multiple comparisons to address the reviewer's point. As our metrics were correlated to each other, we used a correction adapted for non-independent tests (Derringer, 2018). This method allowed us to calculate the effective number of tests (M_{eff}) among our four correlated variables (*weight*, *efficiency*, *clustering coefficient*, and *eigenvector centrality*). We did not apply corrections for Studies 2 and 3, as the metrics used were predetermined by Study 1 (Bieth et al., 2021).

We revisited our manuscript to provide detailed methodology concerning the correction for multiple comparisons we did (p.24) and mentioned the surviving results in the result section.

R1 Comment #4

The chosen statistical approach in general seems appropriate. However, the cleaner way to model the data is to do a mixed effects model where the tasks are not merged but this information is explicitly given to the model via fixed or random effects: Hence, given your design I would have expected that you model all available data together (for each outcome measure), i.e. you'd have an additional variable for the task (riddle1, riddle2) and whether the task is naïve, directed or spontaneous transfer.

Thanks for this important comment. Our main goal was to investigate whether solving a riddle was associated with specific SemNets changes. To do so, we used nonlinear mixed effects models, with one model per metric computed. For each model, all available data were considered (i.e., 190 or 20 values per individual per riddle for edge and node metrics, respectively). Each outcome measure (i.e., Δ Metric and solving) was related to a value of *impact rating* and *semantic distance*. Because *impact rating* and *semantic distance* were riddle-specific, our model already takes into account the potential variability between riddles. We agree that adding a riddle variable to the model instead of pooling the four riddles together would be theoretically ideal to test a potential difference between riddles, but this would complicate our model, with quadruple interactions (e.g., *impact rating* x *semantic distance* x Δ Metric x riddle), making the results harder to interpret. As our main goal was not to compare the restructuring effect between riddles, we still believe that our model choice is the most appropriate in our situation. Similarly, adding a variable for the task condition (*naive*, *transfer*) would have even more complexity in the model, with a quintuple interaction. Besides, we consider the restructuring effect on problem-solving and analogous problem-solving as two distinct questions and our initial goal was not to compare them.

R1 Comment #5

*What about other creativity tasks? Does your metric*impact rating interaction effect survive when you model at least one creativity task in your design?*

We thank the Reviewer for this suggestion. As detailed in *Supplementary Data SI2*, we found that several measures of creativity were significantly correlated with problem-solving abilities (the more creative a participant was, the higher the solving rate). These measures included two measures related to another insight problem-solving task that required forming and combining remote semantic associations, as in the Remote Associates task (*CAT-dist* and *CAT-index*), and one measure related to divergent thinking (*ATTA-index*).

We now include new analyses to explore whether the significant relationships that we found between SemNet changes and problem-solving survive if we include *ATTA-index*, *CAT-dist* or *CAT-index* in the models. We explored solution-based (*impact rating*) and remoteness-based (*semantic distance*) SemNet changes separately and repeated the models for each measure of creativity.

Briefly, we found that all significant interaction effects of Δ Metric and *impact rating* or Δ Metric and *semantic distance* on problem-solving remained significant when each creativity measure was added to the model.

In summary, these analyses suggested that creative abilities did not explain relationship between restructuring and problem-solving, indicating that SemNet changes may reflect the creative process rather than individual creative traits. We added these new analyses in *Supplementary Data SI2*. We referred to these new analyses in the revisited manuscript in the *Results* section (p .9):

“In addition, the significant relationship that we found between SemNet changes and problem-solving remained significant when creative measures were added to the model (see *Supplementary Data SI2*), indicating that our findings reflect the creative process rather than individual creative traits.”

and in the *Methods* section (p.24):

“Finally, to ensure that the significant relationships that we observed between SemNet changes and problem-solving were not influenced by individual creative abilities, we ran the same models with each creativity score added as a predictor. These control analyses investigated whether the results remained significant after adding creative measures in the statistical models (see *Supplementary Data SI2*).”

R1 Comment #6

*What about the insight measures? They were not introduced in the introduction section but are the analyzed and reported in the methods/results section. According to the insight literature, insight is often a consequence of restructuring (Danek et al. etc). To further test your hypotheses that the metric*impact rating interaction represents restructuring one would expect a three-way interaction with insight, i.e. the metric*impact rating interaction should be most predictive of solving the task when the participants had experienced an insight.*

We agree with Reviewer #1 that insight is classically associated with restructuring in the literature, particularly in the context of ill-defined problems (as mentioned in the introduction). As suggested by Reviewer #1, we now provide an additional analysis to explore if restructuring (measured by changes in SemNet) was different whether participants solved the problem with or without Eureka. We used nonlinear mixed-effects models to predict insight solving based on the report of Eureka (versus non-insight solving). Currently, the subjective report of Eureka experience during problem-solving is the most common measure used to study insight (Laukkonen & Tangen, 2018). This analysis could not be done in Study 1 alone because of the low solving rate of the riddles ($n = 14$). Hence, we combined the solvers of Zoe riddle from Studies 1 and 2 in order to increase the sample size ($n = 37$ solvers).

We found that solvers with insight had higher remoteness-based SemNet changes than solvers without insight. In other words, participants brought distant associations closer together when a problem was solved with a Eureka compared to when it was solved without a Eureka. Of note, there were no other significant effects, in particular, no significant solution-based SemNet changes that explained insight problem-solving.

This result supports insight theories which state that insight problem-solving involves a restructuring resulting from a combination of remote and unexpected associations of ideas. Recent publications had challenged this theory, arguing that insight problem-solving did not always required restructuring (Becker et al., 2020, 2021; Cushen & Wiley, 2012). Our result suggesting a remoteness-based restructuring associated with insight problem-solving brings new light on the neuroscience of insight. In particular, it empirically supports Schilling’s theory of insight (Schilling, 2005) and introduces new methodological tools that could be applied in the future to better understand the neurocognitive mechanisms of insight.

This new analysis is now detailed in the revisited manuscript in the *Methods* section (p.25) and Results section (p.12) for Study 2. We discussed this new result as follows (p. 15):

“Here, we found that **problem-solving with insight** was especially linked with a local increase in *efficiency*, *clustering* and *weight* in the edges **and nodes** that are initially semantically distant. **This finding** substantiates Schilling’s hypothesis of a fast decrease in path length within individual’s network representations **when solving a problem with insight.**”

R1 Comment #7

I understand that this is a highly exploratory study but it would be appropriate to make some assumptions about the underlying cognitive mechanisms regarding restructuring in the semantic network. In line 380/1 you claim that your approach may provide a mechanistic explanation of how individuals solve new problems. However, I failed to read any attempts to describe the assumed restructuring mechanism.

This is a fair point. To our knowledge, our study is the first to propose a tool to quantify semantic restructuring of problem-related elements by comparing changes in SemNets between two time points (before and after attempting to solve the problem). We argue that such changes in SemNets could already be a mechanistic explanation of how individuals solve new problems, at least at the cognitive level. In addition, we believe that the new analyses that allow us to distinguish solution-based and remoteness-based SemNet changes provide a deeper understanding of the restructuring mechanism promoting problem-solving.

With the new analyses, we provide more precise insight into these mechanisms. We found that SemNet changes was not only based on the *impact rating*, but also on the *semantic distance*. Importantly, we observed different behaviors whether participants found the solution by themselves or not. Thus, SemNet changes suggest that the restructuring associated with problem-solving is related to the combination of semantically remote elements. This interpretation is in line with theories of insight problem-solving and creative problem-solving, emphasizing the formation of semantically remote associations in memory (Chu & MacGregor, 2011; Mednick, 1962; Schilling, 2005). However, we acknowledge that our method measured SemNet organization at two time-points without telling us what was happening between them and how the observed changes occurred. Since we have now a method for measuring restructuring, our study paves the way to investigate further the neural and cognitive mechanisms underlying restructuring.

We edited the manuscript to clarify our claim and question the causality of our results in the Discussion as follows (p. 14-16):

“We found that successful **problem-solving** was especially linked with a local increase in *efficiency and weight* in the edges between words relevant to the solution and words that were initially semantically distant. **These findings can be interpreted in two, not mutually exclusive ways. First, solution-based SemNet changes could suggest a restructuring that better integrates the solution into the mental representation of the problem, while remoteness-based changes indicate a restructuring that brings closer together concepts that were initially remote. Alternatively, these SemNet changes could reflect a restructuring that moves away word pairs that were solution-irrelevant and word pairs that were initially close. Indeed, we could hypothesize that strong but misleading word associations should be inhibited in order to create a new link^{13,64-66} and solve the problem. The inhibition of non-relevant and obvious association of ideas (that often lead to a mental impasse) has been proposed as a first necessary step to initiate restructuring^{64,67}. Further work is needed to characterize more comprehensively the cognitive processes underlying the restructuring evidenced in our study.**”

[...]

“Some limitations must be mentioned. First, **our methodology quantifying SemNet restructuring at two different time-points did not allow us to characterize the**

full mechanisms leading to SemNet changes, not provide a full picture of the dynamic changes occurring during creative problem-solving and whether these changes represented long-term structural changes.

[...]

“Finally, a critical question that our findings leave open relates to the causal link between restructuring and solving. We provided evidence that remoteness-based SemNet changes are involved in solving a future analogous problem and differentiate an active solving of the problem from a mere exposure to its solution (while solution-based SemNet changes do not). Even if the current study design does not address the causality question, these arguments support the idea that at least remoteness-based restructuring precedes solving and cannot be explained by a mere semantic priming of the solution.”

R1 Comment #8

The data seemed to have come from a bigger study but it was not disclosed (in the main text) which study this is and which papers have already been published with this data set. This is highly relevant for scientific transparency.

We apologize if a lack of clarity in our manuscript has caused any a misunderstanding. The current study only describes new datasets, and no previous data were re-analyzed. Some of the tasks used in this study were developed in previous studies from our lab and used to evaluate a distinct group of participants. For instance, Ovando-Tellez, Kenett, et al., 2022 used the RJT with another word list, and Bendetowicz et al., 2017 developed and used the CAT. The riddles and the RJT material were specifically designed for the current study.

R1 Comment #9

Line 353-358: For the first two points you need to use Bayesian statistics or equivalence testing if you want to make actual claims about null effects (see Lakens et al., 2020).

We agree with Reviewer #1 that our claims about null effects were overstated. We thus reformulated the conclusion and interpretation of our non-significant results when appropriate.

R1 Comment #10

Other new approaches of studies investigating restructuring were not mentioned in the introduction (e.g. Wu et al.,2013; Danek et al.2014; Di Bernardi Luft et al.,2018; Becker et al.2020) and should also be discussed in relation to the current study

As Rewiever #1 suggested, we now emphasize in the introduction and discussion those previous works related to restructuring in problem-solving. Changes are presented in the introduction (p.4) and discussion (p.14) sections.

R1 Comment #11

Language: there are a few sentences where a word is missing or there some other language related issues (e.g. line 81:82)

We apologize for these typos. We have carefully double-checked our manuscript to correct all typos. All changes resulting from this proof-reading are highlighted in yellow in the revised manuscript.

R1 Comment #12

What is RJT? (line 163). The authors should describe what those letters stand already the first time they use it in the results section.

We thank the Reviewer for having spotted this point (RJT = Related Judgment Task). We double-checked that every acronym that we used has been previously defined.

R1 Comment #13

Is the data publically available? If yes, where?

The experimental material is available in the *Supplementary Data* associated with this article. The data sets generated and/or analyzed during the current study will be available in a persistent OSF repository after acceptance. Data will be anonymized and will not include participants' personal information. We used open softwares and toolboxes available online: semantic networks metrics were computed using the Brain Connectivity Toolbox (BCT), and statistical analyses were done using Rstudio. We specified the statistical functions that we used in the Method section. Custom scripts written to use BCT or Rstudio scripts will be available on request to the corresponding authors.

All this information is displayed in the Data availability and Code availability section.

R1 Comment #14

How did you chose which words are relevant from each task?

The selection of the riddle-specific word lists used for the RJT followed a procedure described in the *Methods* section (p.19-20).

To briefly summarize our procedure, three of the co-authors, who are experts in creativity research (TB, DO, EV), first independently proposed up to 30 words for each riddle (blind procedure). Each of them adopted the approach used by Durso et al., consisting in selecting words that were explicitly stated in the problem (e.g., bar and shotgun in the Bar riddle), related to the solution (e.g., remedy and relieve in the Bar riddle), or usually associated with the problem but not with the solution (e.g., drunk and loaded in the Bar riddle).

Second, the three experts shared their respective lists and reached a consensus for 20 words per riddle. During this selection process, we were careful to avoid words that were not frequent (lexical frequency higher than one million occurrences in the lexicon database, <http://www.lexique.org/>), too closely related to the solution (e.g., *game* in the Zoe riddle), too strongly associated with other ones (e.g., *bar* and *barman* in the Bar riddle), or too distant from all of the other words of the list. We retained 20 words (rather than 14 in the Durso et al. study) to obtain a larger SemNet, considering that adding a word increased considerably the number of word pairs in the RJT and, thus, time on task. Distributions of *impact ratings* confirmed that our riddle-specific RJT material captures word associations of varying degrees of relevance for solving the riddle (**Figures S4 and S5**).

R1 Comment #15

Line: 236/237: The authors did not show that solving the riddles involves mostly insight solving -> all they showed was the classical effect that correct responses are more often associated with Eureka experiences than incorrect ones. That does not show how often people actually solved those tasks with insight.

We used chi-squared test to assess whether problem-solving (yes or no) and insight solving (with or without Eureka) were two independent variables. Hence, we calculated the proportions of insight solving, non-insight solving, non-insight non-solving, and non-insight solving. We found a significant effect indicating that problem-solving and insight were not independent.

We acknowledge that this analysis does not allow us to conclude that solving the riddles involves mostly insight solving. To avoid confusion, we have removed the sentence making that claim. Of note, when solving rates were reported in the manuscript, we systematically specified the exact number of solvers and the number of Eureka reports in parentheses.

Reviewer #2

R2 General comments to the Author

This paper described a study where participants solved 2 sets of riddle pairs (naïve and transfer conditions) and performed relatedness judgments on words from the two riddles (+ unrelated words) before and after the naïve riddle. Individual semantic networks were estimated based on these relatedness judgments. The results indicated that successful solving was associated with “restructuring”, as evidenced by changes in network metrics (weight, eigenvector centrality and efficiency). These changes also predicted transfer to the other riddle, and the effect was replicated in a new experiment to some extent (not for weight).

Overall, the paper is interesting and presents a quantitative method to assess how individuals solve problems from a representation perspective. However, I have some questions about novelty and the general framing/logic of the studies. Some of my questions are to do with reframing or acknowledging limitations, while others may need more work in terms of a follow-up control experiment to understand the processes involved in the task more deeply.

R2 Comment #1

I believe there is related work by the authors on how networks change as a function of tasks in low/high creative individuals as well as via conceptual combinations. It made me wonder about two things: (1) convergent vs. divergent validity, i.e., what are some tasks where you would not expect these network changes to occur? And (2) what is the novelty of the present work other than exploring the same methods in a new task? I do think there is merit in this work, but I think the paper could benefit from a clearer discussion of how this work is truly adding to the literature. Some of my other comments below elaborate on issue (1).

Thank you for highlighting these points. Network science methodology is gaining interest in many fields of neuroscience research and in particular in the cognitive neuroscience. They have been mainly used to explore the organization of semantic memory and how this organization is related to creativity abilities (Benedek et al., 2017; Bernard et al., 2019a; He et al., 2021; Kenett et al., 2014a; Ovando-Tellez, Benedek, et al., 2022; Ovando-Tellez, Kenett, et al., 2022; Ovando-Tellez et al., 2023; Yan et al., 2021). In other words, previous studies highlighted that the organization of semantic memory network built at a given moment (e.g., highly or weakly connected network) was related to trait creative abilities (e.g., high or low divergent thinking abilities) at the inter-individual level. In the current study, we measured changes in the organization of SemNet at the individual level between two relevant time-points, and related these changes to problem-solving. Hence, rather than quantifying the organization of SemNet at one given moment, we measured changes in SemNets between before and after working on a problem. We investigated if these SemNet changes were associated with successful problem-solving at the individual level, whether this individual is inherently creative or not.

In summary, our work not only used a new task but also pushed network science methodology towards a novel direction compared to previous research. We clarified this point in the introduction (p. 5) and discussion (p. 14).

As mentioned by the Reviewer, we did not specifically explore the sensitivity and specificity of our result. With Study 3, we showed that SemNet changes can occur when the solution was given to non-solver participants (see R1 comment 1). However, these SemNet

changes were significantly different than those observed in participants who found the solution by themselves. In addition, we provided additional analyses showing that changes in SemNet related to one given problem were not associated with other problems that had no analogous similarities (see R2 comment 3). Combined, these results supposed a specific signature of SemNet changes associated with solving a problem and its analogous.

An alternative control condition could use a problem that does not require restructuring (e.g., well-defined problem). In this case, we would assume that solving such a problem will not be significantly associated with remoteness-based changes in SemNets as opposed to solving our riddles. Although this question is interesting, we did not run such a control condition for several reasons (see our response to R1 comment 1). We emphasized this point in the limitation section (p. 16).

R2 Comment #2

One concern I have is something the authors touch upon in the limitations but do not directly examine. What if the “restructuring” is simply a levels-of-processing/exposure effect and doesn’t actually reflect restructuring per se? For example, as the authors suggest, would you expect the same type of effect if the riddle did not require restructuring but still required the use of some concepts vs. others from the RJT? The current approach seems to lack a bit of specificity in terms of what it predicts and what counts as restructuring. Another control experiment would be to provide the participants with the solution immediately and then measure pre/post network changes – since they did not necessarily come up with the solution themselves, would you still expect their networks to have changed? Basically, I am unclear on whether the change in network metrics is truly a “restructuring” or simply an outcome of how some concepts may be more salient than others after having solved the riddle (which could be achieved without solving it as well, by mere exposure).

We understand Reviewer #2’s concerns about the interpretation of the relationship between SemNet changes and problem-solving. We thank the reviewer for the suggestion of an additional control task. We implemented it in the revised version of the manuscript (Study 3). This point has also been raised by Reviewer #1. We now provide additional analyses aiming to specifically address this issue (please see our reply to R1 comment 1 for the details).

R2 Comment #3

The transfer finding is interesting here – especially since the RJT did not include words from the analogous riddle, so there does appear to be something in the structure of the original riddle that the participants are picking up on – I wonder if there was a way to capture this abstraction / higher order process via networks as well, i.e., is it possible they did some type of mapping of concepts from one riddle to the other? If the participants did a combined RJT for riddles 1 and 2 after solving both riddles, there might be a way to capture this forging of new links potentially. Another possible control experiment to run here to make sure that transfer is truly happening as one might expect is to do the same study but pair up riddles that do not have a direct overlap (e.g., Zoe/Daniel) – would you expect changes in the network to predict solving rates for this unrelated riddle in the spontaneous or direct transfer conditions?

We thank Reviewer #2 for this insightful comment. The relationship we found between problem-related SemNet changes and solving an analogous one is quite new. It opens the door for further studies to better understand the meaning of this restructuring in the context of

analogy transfer. As mentioned by Reviewer #2, we think that this restructuring process may allow the formation of an abstract schema that leads to analogy transfer. Unfortunately, we could not formally test the hypothesis of an alignment between concepts from riddle 1 and 2 by building a global SemNet. This was primarily due to time constraints, as the process would involve incorporating the word lists of each riddle. Rating the relatedness of 40 words (i.e., 780 word pairs) for this task would have required more than 65 minutes (without breaks).

We followed the Reviewer's suggestion and performed a control analysis to test the specificity of the SemNet changes in our existing dataset. We investigated if changes in SemNets related to one riddle predicted solving rates for another riddle which had not semantic or structural relationships (e.g., Zoe/Daniel). In practice, we tested whether the changes in SemNets metrics related to the first riddle that participants encountered could predict the solving of one riddle participants were asked to solve later in the experiment (i.e., the riddle in the *naive* or *transfer* condition; these riddles belong to the second couple of riddles, with no analogical similarities to the first riddle).

Briefly, we found that changes in SemNets related to the first riddle were not significantly associated with solving another riddle that showed no analogical similarities. It suggests that restructuring a considered problem may be truly instrumental for successful analogical transfer, and not simply reflect higher individual abilities (i.e., an ability to easily change their associations of ideas, which would have allowed them to solve any problems, analogous or not). We thank again the reviewer for this suggestion, which reinforces our claim that an analogous transfer truly happens and that restructuring had an active role in this process.

We added these new analyses in *Supplementary Data S14*, which are mentioned in the revisited manuscript as follow (p. 11):

"In addition, we ran similar analyses on a non-analogous problem as a control. Using similar statistical models, we did not find a significant relationship between changes in a given problem-related SemNet and the solving of another problem that had no analogical similarities with the initial riddle (see *Supplementary Data S14*)."

R2 Comment #4

It is also important to address some questions from the process perspective, i.e., while estimates at the representation level could be informative, ultimately the problem-solving task involves some type of processing – what is this process and how would the current approach be able to explain this process? Even if we believe that the “efficiency” of the network changes, how does this happen? Given that there are approaches for simulating how activation spreads within a network (e.g., spreadR), there may be ways to model this process. The process modeling would also force the authors to be specific about their predictions and assumptions, e.g., what qualifies as a ‘eureka’ moment vs. not a ‘eureka’ moment, and what kinds of processes mediate these experiences, which ultimately cause restructuring in the network.

The point raised by Reviewer #2 is highly relevant. In our study, we measured changes in SemNet organization between before and after attempting to solve a problem, but we do

not claim that we characterized the sub-processes that underlie the restructuring that we captured.

The Reviewer's idea to use exploratory models such as spreadR is very interesting, but difficult to apply in our dataset for two main reasons: i) such models are typically used in a unique static SemNet and are thus not tailored to capture dynamic changes between two SemNets as in our study; ii) the small size of our individual SemNets (20 nodes) is not optimal to investigate how activation spreads in the SemNet.

However, we performed novel analyses to better understand the processes underlying problem-related restructuring. We now provide a new experiment that allows us to better characterize the SemNets changes that specifically predict successful solving compared to being given the solution (see R1 comment 1). We also add an analysis on insight solving that identify the same type of SemNet changes that specifically predict insight solving compared to non-insight solving (see R1 comment 6). These specific changes include an increased weight of the links between word pairs that were initially remote (remoteness-based SemNet changes). Conversely, changes that were focused on the parts of the SemNets relevant to the solution were less specific: they occurred when participants solve the problem by themselves, but also when they do not, and the solution was given to them (solution-based SemNet changes). Overall, these new data go a step forward in characterizing restructuring, and we assume that remoteness-based SemNet changes reflect a restructuring cognitive mechanism supporting problem-solving. This mechanism seems to involve the combination of remote associates. We agree that our study does not allow to conclude on the processes by which this new combination occurs, and whether it is an associative or more controlled process. Its relationship with insight solving might reflect that associative processes occur. Conversely, the interaction between Δ Metric and *semantic distance* in predicting solving may also reflect that smaller changes in initially very close word pairs predict the failure to solve the problem. Thus, our result may also capture that a lack of changes in strong but misleading word associations impairs problem-solving (Luft et al., 2018; Ohlsson, 1984, 1992).

Our study offers the tools to investigate these hypotheses in further studies to go deeper in understanding the mechanism of restructuring. We revised the manuscript in the Discussion section (p. 14-16) as follows:

“These findings can be interpreted in two, not mutually exclusive ways. First, solution-based SemNet changes could suggest a restructuring that better integrates the solution into the mental representation of the problem, while remoteness-based changes indicate a restructuring that brings closer together concepts that were initially remote. Alternatively, these SemNet changes could reflect a restructuring that moves away word pairs that were solution-irrelevant and word pairs that were initially close. Indeed, we could hypothesize that strong but misleading word associations should be inhibited in order to create a new link^{13,64–66} and solve the problem. The inhibition of non-relevant and obvious association of ideas (that often lead to a mental impasse) has been proposed as a first necessary step to initiate restructuring^{64,67}. Further work is needed to characterize more comprehensively the cognitive processes underlying the restructuring evidenced in our study.”

[...]

“Some limitations must be mentioned. First, our methodology quantifying SemNet restructuring at two different time-points did not allow us to characterize the full mechanisms leading to SemNet changes, not provide a full picture of the dynamic changes occurring during creative problem-solving and whether these changes represented long-term structural changes.”

R2 Comment #5

Finally, while the general network approach is interesting, the authors do not seem to motivate why some metrics might be expected to significantly predict solving success or transfer, and more importantly, which ones you wouldn't expect to predict the same. For example, why would one expect efficiency to be predictive of restructuring but not weight? Overall, it seems a bit of a “kitchen sink” approach where a variety of metrics were examined without properly motivated hypotheses about the explicit measures.

We agree with Reviewer #2 that our study would benefit from a more theory-guided approach to help generate specific hypotheses about how a given metric predicts a particular aspect of the solving process. The field of cognitive network science is increasingly growing, and more and more work is demonstrating the theoretical significance of network metrics in cognition, specifically memory and language (Siew et al., 2019). To avoid too much ‘fishing’, we grounded our choice of metrics on previous work and used a limited set of metrics. We chose the SemNet metrics that have been previously linked to creative abilities as a trait (Benedek et al., 2017; He et al., 2021; Kenett et al., 2014a; Ovando-Tellez, Benedek, et al., 2022; Ovando-Tellez, Kenett, et al., 2022; Ovando-Tellez et al., 2023; Yan et al., 2021), and could thus potentially also explain the creative process as well.

Each metric that we selected quantifies a different characteristic of SemNets (edge metric, node metric, centrality metric). For example, much work has shown how more creative individuals have a SemNet with lower distances between concepts. The metric of *efficiency* is the inverse of *path length* between nodes and has been shown to be a more suitable measure in individual-based SemNets such as we analyze in our paper (He et al., 2021). Furthermore, as we hypothesize SemNet restructuring related to successful riddle solving, based on Schilling’s theory (Schilling, 2005), we expect to find changes in network metrics at the local, node and edge levels. This is why we also examine centrality measures, that have been shown to be a significant metric in memory research (Siew et al., 2019). Finally, we kept *weight* to have a less transformed (and in turn more easily understandable) metric. As mentioned in Reviewer #1 comment 3, we applied a correction for multiple comparisons to reduce the risk of false positives (also please refer to the reply to Reviewer 1 regarding the choice of metrics).

It is not entirely clear why some SemNet metrics reached significance and others did not. Interestingly, we found that prediction of insight (versus non insight) was significant for node metrics whereas prediction of solving (versus non-solving) was significant for edge metric. It is still uncertain whether these differences are meaningful for understanding the underlying related cognitive processes. More work should directly explore if and how distinct SemNet metrics capture specific aspects of cognitive information processing. Otherwise, these differences might be related to variations in statistical power.

We added this point in the limitation section, as follows (p. 16):

“Fifth, it is not entirely clear why some SemNet metrics reached significance in predicting solving success and others did not (see **Table S8**). The differences in the results of Studies 1 and 2 could be explained because *semantic distance* was computed based on a larger dataset for the Zoe riddle and thus may be more reliable than for the other riddles. In most of our analyses, significant results were mainly found for edge metrics (*weight* and *efficiency*), and less consistently for node metrics (*eigenvector centrality* and *clustering coefficient*). This might be due to a lower statistical power for node metrics, which were based on far less data points ($n = 20$ points/participant) than edge metrics ($n = 190$ point/participants). However, we found significant prediction of insight solving (versus non-insight solving) mainly with node metrics (*clustering coefficient* and *eigenvector centrality*) whereas significant prediction of solving (versus non-solving) was significant with edge metrics (*efficiency*). This may indicate that our metrics capture different cognitive processes. Although SemNets are increasingly used in cognitive neuroscience, their exact meaning for human cognition is still uncertain. Further research is needed to directly explore if and how distinct SemNet metrics capture specific aspects of cognitive processing.”

Reviewer #3

R3 General comments to the Author

This high-quality and interesting paper purportedly investigates restructuring during problem solving, in relation to both solving an initial problem and transferring a (similar) solution to an analogous problem. Restructuring is an important concept often (vaguely) asserted to occur during creative problem solving, and rarely actually demonstrated or measured. To further fill this gap, two experiments assess individual participants' solving and semantic networks, the latter through use of relatedness judgment task (RJT). The study advances past work in which semantic relatedness of problem (and solution) related concepts was assessed at the group level, in a 1994 study by Durso et al., even borrowing some (translated?) riddles. To outline, each participant performed the RJT on a set of concepts underlying a riddle & solution, attempted to solve the base riddle, then performed the RJT again, allowing the experimenters to assess change in the semantic network. Then participants attempted to solve an analogous (target) riddle. After some other tasks (including creativity tasks presumably meant to assess what individual factors contributed to solving), this was repeated for a second pair of riddles (base and target of the analogous pair). Assessment of the semantic networks computed through the RJT demonstrated that, on an individual basis, changes in an individual's semantic network were related to solving both the base and target riddle. This linkage between solving a problem and how people conceive of relations between problem elements is potentially important, and certainly novel at the level of individuals.

Most of the concerns I have about the study could be addressed through the writing, rather than requiring any additional data collection. But I think they could be conceptually important.

R3 Comment #1

One minor issue is the terminology about the degree to which network structure is changing versus dynamic access to that structure. It's unlikely that a single riddle changes an individual's long-term semantic network. The authors use the term dynamic(s) just a couple of times, and some of their remaining language could be interpreted as if long term structural changes are occurring. They could simply emphasize that they mean larger changes to the access of terms along links in the structure, and presumably smaller (very small?) long term changes.

We agree with Reviewer #3 that the SemNet changes we observed were contextual and probably not associated with long term changes. For instance, the words 'shotgun' and 'threat' will still be strongly related even after solving the Bar riddle. When we used the word dynamic, we referred to a change in SemNet organization between two time points. We believe this term remains meaningful as it allows to distinguish our current approach from previous approaches, which only explored how SemNet organization at one time point was associated with creative traits.

To take into account the reviewer's comment, we limited the use of 'dynamic' in the revisited manuscript and emphasized this point in the discussion (p. 15-16):

"Some limitations must be mentioned. First, our methodology quantifying SemNet restructuring at two different time-points did not allow us to characterize the full mechanisms leading to SemNet changes, not provide a full picture of the dynamic changes occurring during creative problem-solving and whether these changes represented long-term structural changes."

The reason I think this is important relates to other concerns I have – that their relatedness judgment task is potentially priming solutions, or that the ability of the riddle to prime the RJT (so how ready their networks were prior to the riddle) is what really is predicting solving AND change in RJT. This concern is probably at least partially ameliorated because they assessed RJT prior to initial solving of the base target (and initial network was not as related to initial solving).

Reviewer #3 proposed that seeing solution-relevant word pairs during the RJT could have helped participants in solving both the problem and the analogous one. It is true that we did not control this point. However, words that composed the riddle-specific word list were related to the problem, but they were not all related to the solution. The distributions of *impact ratings* showed that our material used in the RJT captures word associations of varying degrees of relevance for solving the riddle (from misleading associations to helpful associations for solving the riddle). Hence, participants did not only see solution-relevant word pairs during the RJT. In addition, as developed in R1 comment 2, the low solving rate of the riddles (even after completing two times the RJT) suggests that seeing some solution-relevant word pairs did not give hints to participants on the riddle solution. Finally, the words used in RJT were specific to each riddle. In the experimental design, participant did a RJT before and after a riddle and, in a second step, had to solve an analogous one (without doing the corresponding RJT for the analogous riddle). Hence, the RJT cannot induce a priming effect on solving the analogous riddle, since the words used in the RJT were specifically related to the first riddle and had no semantic relationship with the analogous riddle.

*Some of their language might confuse directionality of purported mechanisms. They repeatedly say that changes in the semantic memory network **predicted** solving – but the change was assessed after the solving, so directionality is not clear. It's possible that solving changed the network, or assessment of the network, essentially priming the task. If it was restructuring driving both - or better for their conclusions, restructuring of the network preceding and enabling solving – that's great. But we don't really know for certain. I'm fine with their argument, but they should allow for other interpretations. At the least they should change most of their uses of "predict" to "associated with".*

We agree with Reviewer #3 that our statistics does not allow us to conclude about prediction in a causal sense. Hence, we changed the term 'prediction' used in our conclusions, and reformulated sentences in the revised manuscript. However, when we referred to the statistical analyses, we used the term 'prediction' to emphasize which variables were the independent or dependent variables. We also discussed the directionality of our results in the limitation section.

Similarly, when they discuss the alleged effect of semantic network change on solving of the target (analogous) riddle, it could be interpreted as saying that the target riddle was primed by the RJT, and more in some subjects than others.

We also agree with Reviewer #3 that several interpretations could be put forward regarding the association that we found between SemNet changes and problem-solving. However, as developed in R1 comment 1, we provided a new experiment and analyses to better describe our main result. Briefly (please see R1 comments 1 and 7), we found different types of SemNet changes whether participants found the solution by themselves or not. We also found that restructuring problem-related concepts was associated with solving an analogous one. These arguments strongly suggested that remoteness-based SemNet changes were not the result of a priming, an increased attention, or a pre-exposure to the solution, but rather an active process leading to problem-solving.

Important in this vein: I believe they reported that transfer success was the same, whether an individual solved the base problem or was presented with the solution. But, if restructuring, and the changes in the semantic network, underlay solving of base and of successful transfer, it seems unexpected that merely hearing the solution to the base problem would equally enable solving the target (analogous) riddle. In other words, many of these issues revolve around the idea that assessing the semantic network itself changed the semantic network, and /or altered (generally facilitated) solving. The RJT contained only the puzzle-specific word pairs you generated prior to experiment, correct? So, couldn't this set of words be strengthening the connection b/w the RJT and the solving task as well as between the two puzzles (base and target)? PLEASE discuss this finding in relation to your theoretical interpretation.

We agree with Reviewer #3 that it is surprising that transfer success was not higher in individuals who solved the initial problem themselves than in the ones who were presented with the solution. This absence of difference may result from low statistical power (low solving rate). Alternatively, it could be that restructuring occurring during the first problem is not the only possible mechanism for transfer success and that initial non-solvers successfully solved the analogous problem in a different way. We emphasized this point in the discussion, leaving the door open for further studies to elucidate it (p. 14-15):

“In addition, we also replicated the classical analogy transfer effect shown in previous studies²⁰⁻²⁵ with new riddles that are different from the ones used in the classic tumor-fortress problem paradigm²⁰⁻²⁴. Our design allowed us to identify a novel behavioral signature of analogy transfer, evidenced by shorter solving times for the analogous compared to the naïve problem. Our findings converge with the literature suggesting that transfer to novel problems depends on the structure of the mental representation of the initial problem. Our results further extend this literature by providing empirical evidence that this restructured representation better integrates concepts related to the solution and/or bridges closer previously remote elements of the problem. Overall, future studies are needed to clarify the mechanisms of analogy transfer and elucidate whether the specific SemNet changes that we measured led to the formation of an analogy schema or an abstract representation that facilitated analogical transfer, or whether these SemNet changes reflect a higher individual ability to reorganize one’s semantic associations.”

R3 Comment #2

Another issue regarding their methods to assess analogical transfer is their comparison of the “spontaneous” versus “directed” transfer conditions. These are entirely confounded with time,

since the cue that they should use the prior riddle to help solve the current one was given after 8 minutes of the “spontaneous condition”.

The cue indicating to participants that there is a link between the two riddles was given after four minutes if participants did not find the solution. In this case, they were given four additional minutes to find the solution in the light of this cue.

We agree with Reviewer #3 that the distinction we did between *spontaneous* and *directed transfer* was ambiguous. In the revised manuscript, we chose to keep only analyses related to *spontaneous transfer*. We believe that the *directed transfer* condition is not totally relevant for the question of analogous problem-solving and presents some methodological issues (in particular, participants included in the analyses of *directed transfer* also included those who initially failed in solving the problem in *spontaneous transfer* condition).

In the revisited manuscript, we now refer to the *transfer* condition as the *spontaneous transfer* corresponding to the four first minutes during which participants had no more indication than the analogous problem statement. The method and result sections have been revised accordingly.

R3 Comment #3

Speaking of analogical transfer, there is another writing issue: Because this paper includes strong focus on analogical transfer, it should include more discussion of past results suggesting strong limits to such transfer. I.e., transfer occurs best (perhaps only) when there are strong surface similarities between base and target, or when the relevance of the analogical base is specifically indicated (or, readily deduced by the problem solver). E.g., consider Reed et al 1974 Cognitive Psychology; Holyoak & Koh, 1987 Memory & Cog [cited but not discussed in regards to limits of transfer], etc.. The “directed transfer” condition is strongly related to the issue.

As suggested Reviewer #3, we revisited the introduction (p. 3) and discussion (p. 14-15) of the manuscript related to the analogy transfer. We also discussed the mentioned references.

R3 Comment #4

One other important issue is related to the many measures and network metrics examined. Some of the factors mentioned in methods are barely described in analyses (did Eureka or attention focus affect transfer, or semantic network change?). And they explored several network metrics in parallel (weight, efficiency, centrality, clustering coefficient). The authors should address the multiple comparison issues. Furthermore, a brief explanation on the cognitive meaning of those metrics will be beneficial to readers who are unfamiliar with semantic network.

We thank Reviewer #3 for these suggestions. We added a relevant analysis concerning the relationship between restructuring (i.e., SemNet changes) and the Eureka report (see R1 comment 6). In this analysis, we considered only the correct responses. We used attention focus probes only to be sure that participants were actively working on the problem during the allotted time. We did not analyze the effect of attention on problem-solving as it was out of the scope of our aims.

We understand Reviewer #3’s concerns about the multiple comparisons. All network measures we examined have been previously applied in cognitive research and focus on

different theoretical aspects related to memory and language (Siew et al., 2019). We applied a correction for multiple comparisons in Study 1 (see R1 comment 3), which was indeed exploratory. Studies 2 and 3 used the a priori variables and analyses derived from Study 1.

We explained as much as possible the cognitive meaning in the Methods section (p.22) based on the existing literature.

“We used a limited set of SemNet metrics, that have been commonly used in cognitive research and have been previously linked to creative abilities as a trait^{43,45,46,61–63}. We hypothesized that they could thus potentially also explain the creative process as well. These metrics all quantified a different characteristic of SemNets (edge metric, node metric, centrality metric).”

“We computed several metrics for each node (n = 20) and edge (n = 190) in the SemNets, including: (1) the *weight*, that is the brute strength of association extracted from RJT ratings (the higher the *weight* is, the more associated two nodes are) and is thus easily interpretable; (2) the *efficiency*, that is the inverse shortest path length between two nodes (the higher the *efficiency* is, the more efficient the connection between two nodes is) and has been shown to be a more suitable measure than path length in individual-based SemNets such as the ones we analyzed in our study⁵⁸; (3) the *clustering coefficient*, that is the degree to which neighbor nodes are also connected to one another (the higher the *clustering coefficient* is, the more interconnected words are); and (4) the *eigenvector centrality*, that is a self-referential measure of centrality that considers the centrality of neighbor (the higher the *eigenvector centrality* is, the more central and influential the node is) and is the centrality measure with the highest reliability compared to other centrality measures in cognitive networks⁶⁶. In previous studies, *efficiency*, *clustering coefficient*, and *eigenvector centrality* have all shown promising correlations with creativity measures^{54,55,60,80,81}. For example, more creative individuals often exhibit a more connected (higher *clustering coefficient*) and efficient (higher *efficiency*) SemNet than less creative individuals^{50,51,53,54,59}. This suggests that ideas in more creative people were more interconnected and closer, leading to more flexible and efficient spontaneous association of ideas.”

R3 Comment #5

Below, I discuss some general areas that could be improved. Although these include methodological concerns, I believe the research is still meaningful, just that some of the issues should be addressed in the text (perhaps in the supplement for some).

Certain aspects of the material and design lead to some caveats or hedges (or skepticism from some readers). The riddles themselves, in my opinion, aren't great problems, as they each tend to rely on a particular trick, rather than a satisfying novel semantic relation that fulfills many constraints. The problems may fit, in some ways, as "ill-defined problems", but these aren't equivalent to all creatively solved problems. Individually the riddles satisfy at least some of the criteria for this domain (they're not so much ill-constrained, as the authors contend, as misdirected; but that's true of a number of classic puzzles in the area). However, more of an issue arises concerning analogical transfer. Each problem is presented soon after the analogical base (with an intervening task that is also related, although not problem solving per se). Thus, it fits with the literature suggesting that transfer to novel problems depends on knowledge that the form of the solution should transfer. In this case, there is a "trick" to one puzzle and the next puzzle uses the same trick (the scenario is a game; or what seems like rude

behavior is helping with a condition). Participants may be on the lookout for the trick, hence more likely to solve second problem.

We thank the reviewer for this interesting comment. We believe that our problems reached the typical criteria corresponding to creative problem-solving: the solution is both original (not the first that comes to mind) and adapted to the context. Moreover, we showed that solving our problems was significantly correlated with other measures of creativity such as another insight problem-solving requiring forming and combining distant semantic association (remote associates test), but also with commonly used divergent thinking test (Torrance tests). We also provided additional analyses showing that these relationships with creative abilities did not significantly affect the link between restructuring and problem-solving (see R1 comment 5). Finally, we found that solving our problems was often associated with a Eureka report. This phenomenon is classically used to investigate insight problem-solving, which is considered as participating in creative abilities.

It is true that our problems required to think out of the box to find the solution. This can be interpreted as a “trick” common to one puzzle and the analogous one as suggested by the Reviewer#3. Most previous studies in the field mainly focused on the fortress and tumor paradigm (Barnett & Ceci, 2002; Cushen & Wiley, 2018; George & Wiley, 2018; Gick & Holyoak, 1980, 1983; Reed et al., 1974). In this paradigm, the “trick” would be the scenario of a convergence. Unlike these studies, our results demonstrate the role of analogical transfer in four new combinations of couples of riddles. We found that analogical transfer was associated with both remoteness-based and solution-based SemNet changes. It remains difficult to detect the information related to the “trick” with the current material. However, we will consider this insightful comment for further studies.

We clarified this point in the discussion section (p. 15).

R3 Comment #6

Equal transfer whether solve base or have solution given to them? This seems a pretty big deal. How did it affect their semantic net? (contrast with Durso or other work on transfer)

Please see the response to R3 comment 1 that addresses this question.

R3 Comment #7

Changes in SemNet (liens 247-262)

I suggest reordering section a little bit: Would be helpful to state in plain English first, prior to the analyses. Emphasize that the analysis is done on the CHANGE in scores (if that is indeed the case) after solving the naïve (or “base”) problem. And perhaps spell out what you’re looking for, prior to giving the data, which are relatively complicated. I.e., if I understand, you’re looking for a strong relation (beta weight) between solving and the subsequent change in SemNet. But your text actually implies that the change in SemNet “predicts” higher solving, which sounds like it’s coming before (thus implying transfer, but you’re still looking at naïve/base here, right?). In a way, it’s more like solving predicts (temporally) the change in SemNet, no? (lines 263-266 interesting, but again “predict” seems misleading)

We thank the Reviewer for this useful suggestion. We now state explicitly the expected results before each analysis that we report.

We agree that our predictions do not reflect temporality between changes in SemNets and solving success. Because it is difficult to ask participants to assess semantic associations at the same time as they are trying to solve a problem, we indeed performed the second RJT after the active solving phase (naive problem). Our idea was that this second RJT would represent the novel mental representation of the problem (compared to the pre-RJT) and that changes in SemNets would reflect the restructuring process that leads to the solution rather than be driven by the solution itself. In other words, we expected the changes in SemNets to precede the solution, not the other way around. This assumption was indeed debatable (as also pointed out by the other reviewers) and the results of our new control experiment (solution group) suggest that changes actually happen in both directions (before and after the solution). In the revised version of the manuscript, we toned down our claim of directionality and discussed the different hypotheses in the light of additional analyses (see R1 comment 1). Even if the term ‘predict’ is often used when reporting the results of a mixed model, we decided to rephrase this term in the result and discussion sections to avoid confusion, as suggested by Reviewer #3.

R3 Comment #8

“Predict” works better for describing the relation between individual SemNet changes and solving of the analogous problem, which comes later. The argument is a good one, but there still may be a couple potential issues that, while they don’t prohibit publication, need to be discussed. Mainly: to what extent is the purported link b/w SemNet changes & analogous solving actually driven by initial solving OR by increased priming of the solution (the concept being transferred) due to both solving and the RJT. Specifically, participants who solve the base problem are more aware of certain elements, and the potential relations between them (what looks like restructuring), so when they take the second RJT those concepts or pairings are both more primed by their prior experience AND more priming for future solving. Such an idea doesn’t completely negate the idea that change in semantic networks could be important, but ...

Please see the response to R3 comment 1 that addresses this point. Note that this comment appears truncated.

R3 Comment #9

The exclusion of participants (for good reasons, such as they’d heard a puzzle before) should be mentioned in the Participants section, so we know how many people were contributing to final data analysis. The main paper (I’ll double check the Supplement) confused me as to how many people were excluded completely or in part due to RJT, to yield final # of participants. Also, lines 654-666, does the 22 correct responses and 120 non responses refer to puzzle solving or RJT, which the rest of this paragraph is discussing. Either way, that seems to be not a great ratio.

We specified the size of the final sample in the Participants section (p.17), indicating that details about exclusions are available later in the Methods (p.22).

In the Supplementary Data, Table S12 provided information on the number and reasons for excluding participants' data: (i) cases where participants already knew the riddle (first column) and (ii) because of technical issues related to the tasks or to the SemNet preprocessing (third column). The second column corresponds to the number of analyzed participants' data for the behavioral analyses (such analysis includes some participants who do not pass the criteria for the SemNet analysis) and the fourth column provides the number of analyzed participants' data for the SemNet analyses. The resulting sample included 22 solvers and 120 non-solvers distributed over 80 participants (two data points per participant as participants had to solve two riddles during the experiment). Table S13 exposes the sample size but at the individual level (i.e., how many participants who solved either one or two riddles were finally included).

We agree that our statistical design was unbalanced, an issue which may have impacted our results. However, we presented two arguments in response to R1 comment 2, supporting the robustness of our results despite this unbalanced design.

R3 Comment #10

Were you worried that instructing participants to report their thoughts and processes could cause a verbal overshadowing effect (decrease solving) or otherwise alter their natural solving behavior?

It is indeed possible that asking participants to report their thoughts during the solving phase negatively impacted problem-solving. However, thinking of a riddle for 10min straight might be tough, especially for those who have no idea of the solution. We thus decided to use probes to limit mind-wandering and ensure that participants remain actively focused on the problem. We now included this limitation in the Discussion (p. 16):

“Third, the solving rate of the riddles in the *naive* condition was overall low. It may be partially due to the difficulty of the riddle or to the additional cognitive load induced by having to report their ideas in real-time and answering questions about their attentional focus during the reflection period.”

R3 Comment #11

When reporting Eureka, the criterion “not the direct result of cognitive effort” seems a bit strict, but we can hope the participants understood this as a graded contrast to Analytic, rather than a strict requirement.

We acknowledge that it might sound strict. We will consider rephrasing the definition of Eureka we gave to the participants in our future studies.

R3 Comment #12

9 judges is a small N for determining the baseline relatedness of word /puzzle pairs in the RJT

We agree with Reviewer #3 that nine judges to determine the impact rating is quite small. However, we found a strong reliability between raters with a interclass coefficient (ICC)

higher than .80 for all riddles (Zoe: ICC = .93; Car: ICC = .84; Bar: ICC = .90; and Daniel: ICC = .94).

R3 Comment #13

When were the creativity tasks given, in relation to first & 2nd solving attempts, and SJT tasks? (wondering about timing b/w 1st & 2nd solving and RJT, and whether subs are likely to pick up on the how these components are related).

Participants had to solve two couples of analogous riddles. Within each couple, participants completed two RJT that were before the presentation of the riddle in the *naive* condition and after trying to solve it. Between the first and second couple of riddles, the participantd performed the creativity tasks. Those tasks had no relationship with the riddles. In addition, after completing the second RJT, the solution to the problem in *naive* condition was given to the participants. They were asked to solve the analogous problem after completing an independent creativity task.

The overall design of the experiment was detailed in the Figure 1A, and the position of the creativity tasks was explained in the legend. Finally, we specified in the Methods section (p. 23) that creativity tasks did not interfere with riddle solving as they were semantically unrelated.

R3 Comment #14

Need to spell out relatedness judgment task before first use of RJT abbreviation

We thank the Reviewer for spotting this point. We double-checked that every acronym is defined before it is used.

R3 Comment #15

p.1, line 77: most of à many

line 85: It ultimately à In theory, restructuring ultimately...

p4 line 119: not sure why refs broken up 42-44, 44-47

p17 line 601: missing a close parenthesis,), after "study¹²"

p 20 line 772: "Since our approach was unprecedented" seems a bit strong. It's definitely a novel twist and a potentially valuable contribution, but Durso performed somewhat similar work at group level. I think just saying relatively novel is sufficient, and adding in that replication is always good

Supplement, baseline SemNet, both experiments, I'd say "whether or not they subsequently solved the riddle" (instead of "whether they solved or not the riddle")

We apologize for these errors. We have carefully double-checked our manuscript and made sure to correct all typos. All changes resulting from this proof-reading are highlighted in yellow in the revisited manuscript.

References

- Barnett, S. M., & Ceci, S. J. (2002). When and where do we apply what we learn? : A taxonomy for far transfer. *Psychological Bulletin*, 128(4), 612-637. <https://doi.org/10.1037//0033-2909.128.4.612>
- Beaty, R. E., Kenett, Y. N., Hass, R. W., & Schacter, D. L. (2023). Semantic memory and creativity : The costs and benefits of semantic memory structure in generating original ideas. *Thinking & Reasoning*, 29(2), 305-339.
- Becker, M., Kühn, S., & Sommer, T. (2021). Verbal insight revisited—Dissociable neurocognitive processes underlying solutions accompanied by an AHA! experience with and without prior restructuring. *Journal of Cognitive Psychology*, 33(6-7), 659-684. <https://doi.org/10.1080/20445911.2020.1819297>
- Becker, M., Wiedemann, G., & Kühn, S. (2020). Quantifying insightful problem solving : A modified compound remote associates paradigm using lexical priming to parametrically modulate different sources of task difficulty. *Psychological research*, 84(2), 528-545.
- Bendetowicz, D., Urbanski, M., Aichelburg, C., Levy, R., & Volle, E. (2017). Brain morphometry predicts individual creative potential and the ability to combine remote ideas. *Cortex*, 86, 216-229. <https://doi.org/10.1016/j.cortex.2016.10.021>
- Benedek, M., Kenett, Y. N., Umdasch, K., Anaki, D., Faust, M., & Neubauer, A. C. (2017). How semantic memory structure and intelligence contribute to creative thought : A network science approach. *Thinking & Reasoning*, 23(2), 158-183. <https://doi.org/10.1080/13546783.2016.1278034>
- Bernard, M., Kenett, Y. N., Tellez, M. O., Benedek, M., & Volle, E. (2019a). Building individual semantic networks and exploring their relationships with creativity. *CogSci*, 138-144.
- Bernard, M., Kenett, Y. N., Tellez, M. O., Benedek, M., & Volle, E. (2019b). *Building individual semantic networks and exploring their relationships with creativity*. 138-144.
- Bieth, T., Kenett, Y., Ovando-Tellez, M., Lopez-Persem, A., Lacaux, C., Oudiette, D., & Volle, E. (2021). *Dynamic changes in semantic memory structure support successful problem-solving*.
- Christensen, A. P., Kenett, Y. N., Aste, T., Silvia, P. J., & Kwapil, T. R. (2018). Network structure of the Wisconsin Schizotypy Scales–Short Forms : Examining psychometric network filtering approaches. *Behavior Research Methods*, 50(6), 2531-2550. <https://doi.org/10.3758/s13428-018-1032-9>
- Chu, Y., & MacGregor, J. N. (2011). Human performance on insight problem solving : A review. *The Journal of Problem Solving*, 3(2), 6.
- Cushen, P. J., & Wiley, J. (2012). Cues to solution, restructuring patterns, and reports of insight in creative problem solving. *Consciousness and Cognition*, 21(3), 1166-1175. <https://doi.org/10.1016/j.concog.2012.03.013>
- Cushen, P. J., & Wiley, J. (2018). Both attentional control and the ability to make remote associations aid spontaneous analogical transfer. *Memory & cognition*, 46(8), 1398-1412.
- Derringer, J. (2018). *A simple correction for non-independent tests*.
- George, T., & Wiley, J. (2018). Breaking past the surface : Remote analogical transfer as creative insight. *Insight*, 143-168.
- Gick, M. L., & Holyoak, K. J. (1980). Analogical problem solving. *Cognitive Psychology*, 12(3), 306-355. [https://doi.org/10.1016/0010-0285\(80\)90013-4](https://doi.org/10.1016/0010-0285(80)90013-4)
- Gick, M. L., & Holyoak, K. J. (1983). Schema induction and analogical transfer. *Cognitive psychology*, 15(1), 1-38.
- He, L., Kenett, Y. N., Zhuang, K., Liu, C., Zeng, R., Yan, T., Huo, T., & Qiu, J. (2021). The relation between semantic memory structure, associative abilities, and verbal and figural creativity. *Thinking & Reasoning*, 27(2), 268-293. <https://doi.org/10.1080/13546783.2020.1819415>
- Kenett, Y. N., Anaki, D., & Faust, M. (2014a). Investigating the structure of semantic networks in low and high creative persons. *Frontiers in Human Neuroscience*, 8. <https://doi.org/10.3389/fnhum.2014.00407>
- Kenett, Y. N., Anaki, D., & Faust, M. (2014b). Investigating the structure of semantic networks in low and high creative persons. *Frontiers in human neuroscience*, 8, 407.
- Laukkonen, R. E., & Tangen, J. M. (2018). How to detect insight moments in problem solving experiments. *Frontiers in psychology*, 9, 282.
- Luft, C. D. B., Zioga, I., Thompson, N. M., Banissy, M. J., & Bhattacharya, J. (2018). Right temporal alpha oscillations as a neural mechanism for inhibiting obvious associations. *Proceedings of the*

- National Academy of Sciences*, 115(52), E12144-E12152.
- Mednick, S. A. (1962). The associative basis of the creative process. *Psychological Review*, 69, 220-232.
- Ohlsson, S. (1984). Restructuring revisited : II. An information processing theory of restructuring and insight. *Scandinavian journal of psychology*, 25(2), 117-129.
- Ohlsson, S. (1992). Information-processing explanations of insight and related phenomena. *Advances in the psychology of thinking*, 1, 1-44.
- Ovando-Tellez, M., Benedek, M., Kenett, Y. N., Hills, T., Bouanane, S., Bernard, M., Belo, J., Bieth, T., & Volle, E. (2022). An investigation of the cognitive and neural correlates of semantic memory search related to creative ability. *Communications biology*, 5(1), 1-16.
- Ovando-Tellez, M., Kenett, Y. N., Benedek, M., Bernard, M., Belo, J., Beranger, B., Bieth, T., & Volle, E. (2022). Brain connectivity-based prediction of real-life creativity is mediated by semantic memory structure. *Science Advances*, 8(5), eabl4294. <https://doi.org/10.1126/sciadv.abl4294>
- Ovando-Tellez, M., Kenett, Y. N., Benedek, M., Bernard, M., Belo, J., Beranger, B., Bieth, T., & Volle, E. (2023). Brain Connectivity-Based Prediction of Combining Remote Semantic Associates for Creative Thinking. *Creativity Research Journal*, 1-25. <https://doi.org/10.1080/10400419.2023.2192563>
- Reed, S. K., Ernst, G. W., & Banerji, R. (1974). The role of analogy in transfer between similar problem states. *Cognitive Psychology*, 6(3), 436-450. [https://doi.org/10.1016/0010-0285\(74\)90020-6](https://doi.org/10.1016/0010-0285(74)90020-6)
- Schilling, M. A. (2005). A « small-world » network model of cognitive insight. *Creativity Research Journal*, 17(2-3), 131-154.
- Siew, C. S., Wulff, D. U., Beckage, N. M., & Kenett, Y. N. (2019). Cognitive network science : A review of research on cognition through the lens of network representations, processes, and dynamics. *Complexity*, 2019.
- Yan, T., Zhuang, K., He, L., Liu, C., Zeng, R., & Qiu, J. (2021). Left temporal pole contributes to creative thinking via an individual semantic network. *Psychophysiology*, 58(8), e13841.

31st Jan 24

Dear Dr Bieth,

Thank you for your patience during the peer-review process. Your manuscript titled "Dynamic changes in semantic memory structure support successful problem-solving and analogical transfer" has now been seen by 2 of the previous reviewers, and I include their comments at the end of this message. They find your work improved but raised some important remaining points. We remain interested in the possibility of publishing your study in Communications Psychology, but would like to consider your responses to these concerns and assess a revised manuscript before we make a final decision on publication.

We therefore invite you to revise and resubmit your manuscript, along with a point-by-point response to the reviewers. Please highlight all changes in the manuscript text file.

Editorially, we consider emphasize that the journal has strict requirements regarding statistics reporting and interpretation <https://www.nature.com/commpsychol/submit/submission-guidelines#statistical-guidelines>

I highlight in particular that we do not permit the interpretation of non-significant findings derived from null-hypothesis significance tests and require provision of Bayes Factors or equivalence tests to support claims of no difference or no effects. Similarly, we do not permit rhetorical contrasting between significant and non-significant findings to claim a difference in differences. This requires interaction contrasts. Finally, we ask that you make greater use of display items to show the nature of the data and also respond to Reviewer #3's request for an improved presentation.

I am attaching an Editorial Requests Table that details critical reporting requirements for the revised manuscript. Please attend to each item and ensure your manuscript is fully compliant. We are requesting that your manuscript aligns with these requirements as this facilitates the evaluation of your manuscript, reducing delays in re-review and potential future acceptance. If your revised manuscript is not aligned with these requests on major issues, such as those concerning statistics, it may be returned to you for further revisions without re-review. Additional information can be found in our style and formatting guide Communications Psychology formatting guide.

Please use the following link to submit your

- revised manuscript,
- point-by-point response to the referees' comments,
- cover letter (as a separate document),
- the Editorial Policy Checklist (see below),
- the Reporting Summary (see below), and
- the completed Editorial Request Table (attached):

[link redacted]

** This url links to your confidential home page and associated information about manuscripts you

may have submitted or be reviewing for us. If you wish to forward this email to co-authors, please delete the link to your homepage first **

Best wishes,

Marike

Marike Schiffer, PhD
Chief Editor
Communications Psychology

REVIEWER REPORTS:

Reviewer #2 (Remarks to the Author):

I acknowledge the authors' efforts in addressing all of my points, and they have successfully resolved most of them.

There are a few but still relevant issues left (mostly interpretative ones):

1) R1 Comment #1 Response: I appreciate the authors' additional efforts to address my concerns regarding the interpretability of changes in problem-related Semantic Networks (SemNets). Despite these efforts, I maintain the view that demonstrating the relationship between restructuring and SemNet changes requires a proper control condition—a problem without restructuring. While I acknowledge the challenges and time constraints in creating such problems for the current manuscript, considering it as an exploratory study with a novel approach, I find it acceptable if the restructuring is not shown in the most rigorous manner.

An alternative analysis strategy proposed by the authors in Study 3 involves providing non-solvers with the solution and introducing a semantic distance measure, assuming that restructuring involves connecting semantically distant concepts. For a more compelling demonstration of actual restructuring, it would be crucial to show that solvers, compared to non-solvers, connect more solution-relevant semantically distant concepts. This necessitates a three-way interaction: $isSolved = \Delta Metric * IR * SD$. However, the absence of a positive three-way interaction raises concerns about the specificity of the semantic distance measure to solution-relevant concepts.

In summary, merely showing that solvers connect any semantically remote concepts may not be sufficient to convincingly demonstrate successful restructuring unless these concepts are also solution-relevant. If the results do not reveal a three-way interaction but only two independent two-way interactions ($\Delta\text{Metric} * \text{IR} + \Delta\text{Metric} * \text{SD}$), it is imperative to discuss this limitation as a caveat before exploring alternative explanations. Moving on to then clarifying why the two two-way interactions may still support the hypotheses (as you already partially do), even in the absence of a three-way interaction, would enhance the reader's understanding and strengthen the argument as you already partially do ("We found that successful problem-solving was especially linked with a local increase in efficiency and weight in the edges between words relevant to the solution and [independent of solution-relevant words] words that were initially semantically distant. These findings can be interpreted in two, not mutually exclusive ways. First,..."). But what is still missing here, imo, is to discuss the unique characteristics of the chosen riddles (+giving an example) and make a point about how connecting solution-relevant words may not necessarily be the most semantically remote ones explaining the missing three-way interaction.

2) Response to Comment #5 but similar to point 1) above: Regarding the correlation between riddle-solving and creativity measures, this is a nice result confirming that the riddles relate to creative problem solving. But what I am surprised about is that the relationship between the $\Delta\text{Metric} * \text{impact rating}$ interaction (your measure for restructuring) and problem-solving was unaffected by the inclusion of creativity measures. The authors interpret this finding in the sense that SemNet changes may reflect the creative process rather than individual creative traits. While this is a possibility, this finding is still counterintuitive if one assumes that a) restructuring is one core aspect of creative problem solving and b) your $\Delta\text{Metric} * \text{impact rating}$ interaction measures restructuring because the creativity measures and restructuring are supposed to share significant variance in regards to (creative) problem-solving ability. Hence, one would have expected that the relationship between $\Delta\text{Metric} * \text{impact rating}$ interaction and problem-solving ability becomes non-significant after entering the creativity measures or one would have expected a triple interaction (more creative ability [CAT and/or ATTA]+ more restructuring [$\Delta\text{Metric} * \text{impact rating}$ interaction]) is associated with higher problem solving ability. This should at least be briefly mentioned.

Note, from this whole reasoning, the triple interaction between ΔMetric , impact rating, and ATTA_index on problem-solving (assuming it goes into the right direction, but without visual display of the results, see point 3) it is difficult to know) is exactly what one would expect and therefore a more convincing result than the results showing no affect of creativity measures on the relationship between $\Delta\text{Metric} * \text{impact rating}$ interaction and problem solving ability. Hence, when interpreting the results with the creativity measures I would also put more emphasize on this nice three-way interaction.

3) It would also hugely help the reader to more quickly understand your main results if you provided plots of your interaction effects for them especially when you report the three-way-interactions. You can easily produce interaction plots with glmer models with the "ggeffects" library: `ggpredict` (your_model, c('change_metric', 'impact_rating', 'semantic_distance')) %>% plot() + ggplot2::theme_classic()

4) Study1: It is not clear to me why you have two random effects in your models + (1|Subject) + and what the -1 in the second random term is supposed to mean. The second terms makes the model a lot more complex. I also could not find any explanation towards this and why it is necessary to model the data as such and not just use the much simpler random subject intercept (1|Subject). Does the

significance of the results depend of this $(-1+\Delta\text{Metric}|\text{Subject})$ random term?

5) Abstract: You have not accounted for multiple comparisons in your analysis of network parameters in studies 2 and 3, justifying it by the prior selection of relevant parameters in study 1, which is acceptable. However, the omission of correction implies an expectation of an effect in every network parameter when attempting to replicate results in studies 2 and 3 to support your hypotheses. Given that replication is successful for some network measures but not for others, the evidence presented lacks full persuasiveness. It is acknowledged that this is an exploratory study, but this necessitates a toned down interpretation of the main results, particularly in the abstract where space limitations prevent a thorough explanation of the somewhat inconsistent findings, as is done in the discussion. Therefore, the statement in the Abstract: "We found that restructuring a problem-related SemNet was associated with...." should be revised to convey a more nuanced perspective, such as "Some of our network measures provide initial evidence that restructuring a problem-related SemNet was associated with...".

6) Title: You have not delved into the specific dynamics of how changes in semantic memory structure contribute to effective problem-solving (and analogical transfer). With only two time points—pre and post solution—it would be more appropriate to omit the term "dynamic" from the title.

7) Reporting summary: "hierarchical and complex designs, identification of the appropriate level for tests and full reporting of outcomes" n/a -> should be put to "confirmed" because you used hierarchical models (random effects models are hierarchical models too) but see point 4)!

8) Minor language typos:

- "Contrary to the previous result, this result indicates that remoteness-based SemNet changes (i.e., based on the semantic distance) was higher in the solver group compared to the solution group." -> do you mean non-solver group instead of "solution group"?

- "This triple interaction effect was not significant for the other metrics (efficiency: $\beta = -0.02$, $p = .43$; clustering coefficient: $\beta = 0.20$, $p = .34$; and eigenvector centrality: $\beta = -0.04$, $p = .68$) (all statistical analyses are available in Table S7)." -> statistical analyses

- "First, solution-based SemNet changes could suggest a restructuring that better integrates the solution into the mental representation of the problem, while remoteness-based changes indicate a restructuring that brings closer together concepts that were initially remote." -> reStructuring

Finally, it would greatly help me if you could directly paste all the relevant changes made in the manuscript below your response. This would eliminate the need for me to search through the manuscript to identify alterations related to your response. There have been instances, such as in Comment #3, where the response directed me to check the manuscript with a specified page number, but the manuscript did not have page numbers.

Reviewer #3 (Remarks to the Author):

I appreciate the effort that authors have put in to address different issues pointed out by the

reviewers. Overall, I think this revised manuscript is much stronger and makes clearer claims about the hypothesized constructs. The control experiment (Study 3) is particularly interesting as it provides evidence that solvers and non-solvers who are given the solution are likely showing different types of associations to solution-relevant concepts.

My remaining comments are very minor.

Not sure if this would be possible, but it would be interesting to report for each riddle, exactly which "remote" associations were rated higher (or brought closer) when the riddle was solved vs. when the solution was simply presented (in Study 3). This might help make the claims a bit more clearer as I do think the paper is a bit dense with all the metrics. For instance, it is very hard to grasp the key two-way interaction between metric x impact rating and metric x semantic distance and examples(or figures) would really help elucidate this pattern a lot better.

Overall, I think examples could be really useful throughout the paper. For example, when you discuss weight, efficiency, etc., these metrics seem very abstract and it would be good to ground them in some way using examples of words via tables or figures.

Table S8 should be moved to the main text - this is an overall summary of the key findings that would be helpful for the reader. I might even add a verbal description of what the effect means as an additional column for each study.

There are several typos in the ms - hopefully this can be corrected in proofing.

EDITORIAL POLICIES

We ask that you ensure your manuscript complies with our editorial policies and reporting requirements.

To that end, we require revised manuscripts to be accompanied by two completed items: a reporting summary that collects information on study design and procedure, and an editorial policy checklist that verifies compliance with all required editorial policies.

- Nature Research Reporting Summary
- Editorial Policy Checklist

All points on the policy checklist must be addressed. Your revised manuscript can only be sent back to the referees if these checklists are completed and uploaded with the revision.

Notes: If you have submitted a Stage 1 Registered Report, Review, Primer, Comment, or Perspective you do not need to submit these forms. If you have already submitted these forms, you may disregard this request.

* TRANSPARENT PEER REVIEW: Communications Psychology uses a transparent peer review system. This means that we publish the editorial decision letters including Reviewers' comments to the authors and the author rebuttal letters online as a supplementary peer review file. However, on author request, confidential information and data can be removed from the published reviewer reports and rebuttal letters prior to publication. If your manuscript has been previously reviewed at another journal, those Reviewers' comments would not form part of the published peer review file.

If you experience problems in linking your ORCID, please contact the Platform Support Helpdesk.

Reviewer #2**R2 General comments to the Author**

I acknowledge the authors' efforts in addressing all of my points, and they have successfully resolved most of them. There are a few but still relevant issues left (mostly interpretative ones).

R2 Comment #1

I appreciate the authors' additional efforts to address my concerns regarding the interpretability of changes in problem-related Semantic Networks (SemNets). Despite these efforts, I maintain the view that demonstrating the relationship between restructuring and SemNet changes requires a proper control condition—a problem without restructuring. While I acknowledge the challenges and time constraints in creating such problems for the current manuscript, considering it as an exploratory study with a novel approach, I find it acceptable if the restructuring is not shown in the most rigorous manner.

*An alternative analysis strategy proposed by the authors in Study 3 involves providing non-solvers with the solution and introducing a semantic distance measure, assuming that restructuring involves connecting semantically distant concepts. For a more compelling demonstration of actual restructuring, it would be crucial to show that solvers, compared to non-solvers, connect more solution-relevant semantically distant concepts. This necessitates a three-way interaction: $isSolved = \Delta Metric * IR * SD$. However, the absence of a positive three-way interaction raises concerns about the specificity of the semantic distance measure to solution-relevant concepts.*

*In summary, merely showing that solvers connect any semantically remote concepts may not be sufficient to convincingly demonstrate successful restructuring unless these concepts are also solution-relevant. If the results do not reveal a three-way interaction but only two independent two-way interactions ($\Delta Metric * IR + \Delta Metric * SD$), it is imperative to discuss this limitation as a caveat before exploring alternative explanations. Moving on to then clarifying why the two two-way interactions may still support the hypotheses (as you already partially do), even in the absence of a three-way interaction, would enhance the reader's understanding and strengthen the argument as you already partially do ("We found that successful problem-solving was especially linked with a local increase in efficiency and weight in the edges between words relevant to the solution and [independent of solution-relevant words] words that were initially semantically distant. These findings can be interpreted in two, not mutually exclusive ways. First,..."). But what is still missing here, imo, is to discuss the unique characteristics of the chosen riddles (+giving an example) and make a point about how connecting solution-relevant words may not necessarily be the most semantically remote ones explaining the missing three-way interaction.*

We agree with Reviewer #2 that it will be interesting to add a control experiment with a problem that does not necessitate restructuring. As mentioned in the last round of revision, we chose an alternative strategy (Study 3) because of the practical and methodological challenges in designing such a control experiment.

Furthermore, we agree that it would be intuitive to hypothesize that problem-solving is associated with SemNet changes targeting associations of ideas that are both solution-relevant and semantically remote. As highlighted by Reviewer #2, we did not find a significant three-way interaction (i.e., $\Delta Metric \times impact\ rating \times semantic\ distance$) (but note that we

have a statistical tendency for *efficiency*, with a significant *p*-value that did not survive after correction for multiple comparisons).

However, we found significant positive effects of the interaction between Δ Metric and *impact rating* in one hand, and between Δ Metric and *semantic distance* on the other hand. Importantly, these two individual results were obtained from the same statistical model (and not from two distinct models). By adding both *impact rating* and *semantic distance* factors in a unique mixed-effects statistical model, the common variance between the two variables is left out in the model residues, allowing us to distinguish changes in SemNets that are purely driven by the solution-relevant edges or node, and those driven by the semantic remoteness of the associations. In sum, our results do not allow us to conclude whether *impact rating* and *semantic distance* have an opposite or synergic effect on SemNet changes, but we can put forward that these two variables captured different information considering each other. Indeed, even if the two variables share a common variance, the two-way interaction effects remained individually significant suggesting that they (at least partially) independently contribute to the relationship with problem-solving.

This interpretation is supported when observing the *impact rating* distribution as a function of *semantic distance* (**Figures S2 and S3**). The correlation is not strong. We can see word pairs that are solution-relevant but semantically close (for instance, *to write – ground* for the Zoe riddle, or *relieved – remedy* for Bar riddle) and word pairs that are solution-irrelevant but semantically distant (for instance, *to fly – space*, or *to die – bar* for the Bar riddle). Finally, this is also clearly demonstrated by the dissociation effect between remoteness-based and solution-based restructuring observed in Study 3.

We understand Reviewer #2's concerns that changes in remote associations are more difficult to interpret for problem-solving if they do not specifically target solution-relevant nodes/pairs. We believe that remoteness-based restructuring may capture more specifically the creative restructuring process (because they reflect how remote concepts are initially combined). Such type of restructuring may create shortcuts between different SemNet communities that may be relevant for successful problem-solving.

We specifically now discuss this point in the discussion section and rephrased our conclusion as follows (p. 23):

“SemNets changes associated with problem-solving success were related to local effects that focused **separately** on both solution-based and remoteness-based concepts (Studies 1 and 2). We found that successful problem-solving was **especially** linked with a local increase in *efficiency and weight* in the edges between words relevant to the solution and words that were initially semantically distant. **In studies 1 and 2, we did not find three-way interaction effects that might have suggested a synergic or opposite effect of solution-based and remoteness-based SemNets changes on problem solving success. These results may suggest that the restructuring of solution-relevant words may not necessarily involve the most semantically remote ones explaining the missing three-way interaction. However, our significant two-way interaction effects may reflect that solution-based and remoteness-based SemNets changes captured distinct processes.**”

We also provide examples of word pairs that are solution-relevant but semantically close, and conversely (p. 10):

“We observed that solution-relevant edges or nodes were usually also semantically remote (and conversely), but the correlation was weak. Thus, some edges could be solution-relevant but semantically close (for instance *to write – ground* for the Zoe riddle, or *relieved – remedy* for the Bar riddle), and other edges could be solution-irrelevant but semantically distant (for instance *to fly – space* for the Zoe riddle, or *to die – bar* for the Bar riddle). This suggests that *impact rating* and *semantic distance* variables could capture different effects.”

Finally, we clarify in the Method section the relevance of including both *impact rating* and *semantic distance* in the statistical model, and the three-way interaction effect meaning (p. 13):

“By adding both *impact rating* and *semantic distance* in statistical models, the common variance between the two variables is left out in the model residues, allowing us to distinguish changes in SemNets that are purely driven by the solution and those likely to capture more specifically the creative restructuring process (because they reflect how initially remote concepts are combined).”

“Finally, a Δ Metric x *impact rating* x *semantic distance* interaction effect explores whether solution-based and remoteness-based restructuring have a synergistic or opposite effect.”

R2 Comment #2

*Regarding the correlation between riddle-solving and creativity measures, this is a nice result confirming that the riddles relate to creative problem solving. But what I am surprised about is that the relationship between the Δ Metric**impact rating* interaction (your measure for restructuring) and problem-solving was unaffected by the inclusion of creativity measures. The authors interpret this finding in the sense that SemNet changes may reflect the creative process rather than individual creative traits. While this is a possibility, this finding is still counterintuitive if one assumes that a) restructuring is one core aspect of creative problem solving and b) your Δ Metric**impact rating* interaction measures restructuring because the creativity measures and restructuring are supposed to share significant variance in regards to (creative) problem-solving ability. Hence, one would have expected that the relationship between Δ Metric**impact rating* interaction and problem-solving ability becomes non-significant after entering the creativity measures or one would have expected a triple interaction (more creative ability [CAT and/or ATTA]+ more restructuring [Δ Metric**impact rating* interaction] is associated with higher problem solving ability. This should at least be briefly mentioned.*

*Note, from this whole reasoning, the triple interaction between Δ Metric, *impact rating*, and ATTA_index on problem-solving (assuming it goes into the right direction, but without visual display of the results, see point 3) it is difficult to know) is exactly what one would expect and therefore a more convincing result than the results showing no affect of creativity measures on the relationship between Δ Metric**impact rating* interaction and problem solving ability. Hence, when interpreting the results with the creativity measures I would also put more emphasize on this nice three-way interaction.*

The aim of the study was to explore the restructuring process as a cognitive mechanism involved in problem-solving. As we used new material (riddles), we explored the relationship between creativity measures and problem-solving to confirm that our materials capture something related to creativity (i.e., creative problem-solving). Although problem-solving ability was associated with creativity, what we wanted to describe was what happened in individual SemNets in solvers compared to non-solvers, irrespective of creativity traits. We used mixed-effects regression model to consider inter-individual variability in our analyses.

In the previous revision, a Reviewer asked if creativity measures influenced the relationship that we observed between SemNet changes and problem-solving. Hence, we provided additional analyses showing that including creativity measures in the statistical model did not modify our results concerning SemNet changes. In contrast, we observed that the relationship between creativity and problem-solving became non-significant. The fact that these significant effects became non-significant when adding factors related to restructuring (i.e., Δ Metric, *semantic distance*, and *impact rating*) suggests a mediation effect. For instance, the interaction effect of Δ Metric (*weight*) and *impact rating* on problem-solving captured all the variance, leaving the main effect of *CAT-dist* non-significant. In this case, the restructuring factor (interaction effect of Δ Metric and *impact rating*) may play a role of mediator in the relationship between creativity and problem-solving abilities.

Supporting the idea that the restructuring effect we found reflected a cognitive mechanism, we found that riddle-specific SemNets was not significantly associated with the solving of other problems.

The three-way interaction effect we observed between *ATTA-index*, Δ Metric, and *impact rating* on problem-solving was not in the direction proposed by Reviewer #2. Although difficult to interpret, if we reduced the *ATTA-index* into a binary variable, we observed that individuals with higher *ATTA-index* (lower creativity abilities) had a more robust interaction effect of Δ Metric and *impact rating* on problem-solving (**Figure S7** and **Supplementary Data S11**, see below). This result may be counterintuitive. However, problem-solving may require additional creative cognitive processes than the ones involved in *ATTA*, aimed to capture divergent thinking abilities (such as convergent thinking to match with problem statement constraint). In addition, the two-way interaction effect between Δ Metric and *impact rating* on problem-solving stayed significant even when the three-way interaction between *ATTA-index*, Δ Metric, and *impact rating* on problem-solving was.

We specified this interpretation in **Supplementary Data S11** as follows (p. 24):

“These results suggest that the significant relationships we observed between SemNets changes and problem-solving were not merely due to individual differences in creative abilities. In addition, we found a significant three-way interaction effect of Δ Metric, *impact rating*, and *ATTA_index* on problem-solving. It suggests that the effect of solution-based SemNet changes (solution-based nodes becoming more central) on successful solving was higher for individual with lower divergent thinking abilities (**Figure S7**), which could be counterintuitive. However, the two-way interaction effect between Δ Metric and *impact rating* on problem-solving stayed significant even when the three-way interaction between *ATTA-index*, Δ Metric, and *impact rating* on problem-solving was. We could speculate that problem-solving requires a balance between divergent thinking (to generate alternative problem representation and solutions) and convergent thinking (to respect constraints related to the problem statement). This

hypothesis is supported by previous study showing that RAT solving requires a divergent and then a convergent process ⁶ or other suggesting that divergent thinking and convergent problem-solving are not strongly related ⁷.

Figure S7. Relationship between ATTA creativity measure, restructuring, and problem-solving. Three-way interaction effect of Δ Metric, impact rating, and semantic distance on problem-solving for eigenvector centrality. Graphs represent the Δ Metric weighted by the impact rating (Δ Metric \times impact rating) as a function of solving success for individuals with low ATTA-index (i.e., low divergent thinking abilities, in red) and high ATTA-index (i.e., high divergent thinking abilities, in blue). Lines represent the fitting curves of the interaction effect (Δ Metric \times impact) in predicting problem-solving with the same color code.”

R2 Comment #3

It would also hugely help the reader to more quickly understand your main results if you provided plots of your interaction effects for them especially when you report the three-way-interactions. You can easily produce interaction plots with glmer models with the "ggeffects" library: `ggpredict(your_model, c('change_metric', 'impact_rating', 'semantic_distance')) %>% plot() + ggplot2::theme_classic()`

We thank the reviewer for this useful suggestion. We now provide additional figures (Figure 3, p. 20, and Figure 4, p. 21) providing a clearer understanding of the results when three-way interaction effects were significant.

Figure 3. Impact rating and semantic distance capture different SemNets changes related to insight problem-solving (Study 2). Three-way interaction effects of Δ Metric, *impact rating*, and *semantic distance* on insight problem-solving (compared to non-insight problem-solving) for *efficiency*. Graphs represent the Δ Metric weighted by the *impact rating* (Δ Metric x *impact rating* – solution-based restructuring, in red) or the Δ Metric weighted by the *semantic distance* (Δ Metric x *semantic distance* – remoteness-based restructuring, in blue) as a function of insight report when solving the riddle. Lines represent the fitting curves of the interaction effect (Δ Metric x *impact rating* in red, and Δ Metric x *semantic distance* in blue) in predicting insight problem-solving. A solid line indicates that a significant effect ($p < .05$) was found.

Figure 4. Impact rating and semantic distance capture different SemNets changes related to problem-solving (Study 3). Three-way interaction effects of Δ Metric, *impact rating*, and *semantic distance* on problem-solving (compared to non-solvers with solution) for *weight*. Graphs represent the Δ Metric weighted by the *impact rating* (Δ Metric x *impact rating* – solution-based restructuring, in red) or the Δ Metric weighted by the *semantic distance* (Δ Metric x *semantic distance* – remoteness-based restructuring, in blue) as a function of solving success. Lines represent the fitting curves of the interaction effect (Δ Metric x *impact rating* in red, and Δ Metric x *semantic distance* in blue) in predicting problem-solving. A solid line indicates that a significant effect ($p < .05$) was found.

We also provide a new figure that illustrates our findings (Figure 5, p. 22). It shows the different patterns of SemNets changes according to the group (non solver, solver, and solution groups), the word pair relevance for problem-solving (*impact rating*), and the word pair semantic remoteness (*semantic distance*).

p. 21-22:

“An illustration of the results is provided for summarizing our findings (Figure 5), showing the different patterns of SemNets changes according to the group (non solver, solver, and solution groups), the word pair relevance for problem-solving (*impact rating*), and the word pair semantic remoteness (*semantic distance*).

Figure 5. SemNets changes at the group level regarding *impact rating* and *semantic distance* of word pairs. Each row represents a word pair ($n = 190$) that are sorted by increasing *impact rating* (A) or *semantic distance* (B). $\Delta Weight$ are averaged across participants according to the group (non solver, solver, or solution groups). We combined the three study and used only the Zoe riddle. Color code indicates averaged $\Delta Weight$ value from negative (in red) to positive value (in green). We used *weight* for this figure as it is less transformed (and in turn more easily understandable) metric.”

R2 Comment #4

Study1: It is not clear to me why you have two random effects in your models + (1|Subject) + and what the -1 in the second random term is supposed to mean. The second terms makes the model a lot more complex. I also could not find any explanation towards this and why it is necessary to model the data as such and not just use the much simpler random subject intercept (1|Subject). Does the significance of the results depend of this (-1+ $\Delta Metric$ |Subject) random term?

We thank the reviewer for pointing this lack of clarity. In Study 1, we included two random effects in our statistical models. The first one (i.e., (1|Subject)) allowed us to consider the random effect on the intercept, whereas the second one (i.e., (-1 + $\Delta Metric$ |Subject)) allowed us to consider the random effect on slope. We used “(1|Subject) + (-1 + $\Delta Metric$ |Subject)” to represent the random effect in the formula instead of “(1+ $\Delta Metric$ |Subject)” because we did not know whether the random effects between intercept and slope were dependent or independent (-1 indicates in the formula that we removed the intercept).

By default, we considered that the two effects were independent. Indeed, we have no argument to claim that subjects would differ similarly on both the slope and intercept (if one subject has a higher intercept than another one, we don't see why they would also have a higher slope).

To test whether this strategy affected the results, we ran again the models without the random effect on slope ($-1 + \Delta\text{Metric}|\text{Subject}$). The results remained significant for model predicting problem-solving (*weight*: $\Delta\text{Metric} \times \text{impact rating}$, $p = .02$; *efficiency*: $\Delta\text{Metric} \times \text{impact rating}$, $p = .006$ and $\Delta\text{Metric} \times \text{semantic distance}$, $p = 0.001$; *eigenvector centrality*: $\Delta\text{Metric} \times \text{impact rating}$, $p = .01$) and analogous problem-solving (*weight*: $\Delta\text{Metric} \times \text{semantic distance}$: $p = .01$; *efficiency*: $\Delta\text{Metric} \times \text{impact rating}$, $p = .048$ and $\Delta\text{Metric} \times \text{semantic distance}$, $p = .01$). Only the $\Delta\text{Metric} \times \text{impact rating}$ interaction effects on analogous problem-solving for *weight* became non-significant, but trending, when removing the random effect on slope ($p = .057$).

As we believe the statistical models that we used were more appropriate, we did not change our statistical design in the main manuscript. We can add them on request for the editor to the supplementary material. Nevertheless, we updated the method section to clearly explain the meaning of the random factors included in the model (p. 12):

“Participants were entered as a random-effect factor in the model on both the intercept ($1|\text{Subject}$) and slope ($\Delta\text{Metric}|\text{Subject}$). By default, we considered that the two random effects were independent as we have no argument to claim that subject would differ similarly on both slope and intercept (i.e., $(1|\text{Subject}) + (-1 + \Delta\text{Metric}|\text{Subject})$). These random effects allow us to take into account the repeated measures across subjects as a random-effect factor (Subject, maximum two riddles per subject) and inter-individual variability.”

R2 Comment #5

Abstract: You have not accounted for multiple comparisons in your analysis of network parameters in studies 2 and 3, justifying it by the prior selection of relevant parameters in study 1, which is acceptable. However, the omission of correction implies an expectation of an effect in every network parameter when attempting to replicate results in studies 2 and 3 to support your hypotheses. Given that replication is successful for some network measures but not for others, the evidence presented lacks full persuasiveness. It is acknowledged that this is an exploratory study, but this necessitates a toned down interpretation of the main results, particularly in the abstract where space limitations prevent a thorough explanation of the somewhat inconsistent findings, as is done in the discussion. Therefore, the statement in the Abstract: "We found that restructuring a problem-related SemNet was associated with..." should be revised to convey a more nuanced perspective, such as "Some of our network measures provide initial evidence that restructuring a problem-related SemNet was associated with..."

We agree with Reviewer #2 that our claims about our results were overstated in the abstract. We thus reformulated the abstract taking into account the word number limitation:

Abstract (p. 2):

“Our results provide initial evidence that problem-related SemNets restructuring may be associated with the successful solving of this problem and an analogous one.”

R2 Comment #6

Title: You have not delved into the specific dynamics of how changes in semantic memory structure contribute to effective problem-solving (and analogical transfer). With only two time points—pre and post solution—it would be more appropriate to omit the term "dynamic" from the title.

We have removed the term “dynamic” from the title, as Reviewer#2 suggested.

R2 Comment #7

Reporting summary: "hierarchical and complex designs, identification of the appropriate level for tests and full reporting of outcomes" n/a -> should be put to "confirmed" because you used hierarchical models (random effects models are hierarchical models too) but see point 4)!

Reviewer #2 is correct. Hence, we changed the reporting summary and confirmed that we used hierarchical and complex designs.

R2 Comment #8

Minor language typos:

- *"Contrary to the previous result, this result indicates that remoteness-based SemNet changes (i.e., based on the semantic distance) was higher in the solver group compared to the solution group." -> do you mean non-solver group instead of "solution group"?*
- *"This triple interaction effect was not significant for the other metrics (efficiency: $\beta = -0.02$, $p = .43$; clustering coefficient: $\beta = 0.20$, $p = .34$; and eigenvector centrality: $\beta = -0.04$, $p = .68$) (all statistical analyses are available in Table S7)." -> statistical analyses*
- *"First, solution-based SemNet changes could suggest a restructuring that better integrates the solution into the mental representation of the problem, while remoteness-based changes indicate a restructuring that brings closer together concepts that were initially remote." -> restructuring*

Finally, it would greatly help me if you could directly paste all the relevant changes made in the manuscript below your response. This would eliminate the need for me to search through the manuscript to identify alterations related to your response. There have been instances, such as in Comment #3, where the response directed me to check the manuscript with a specified page number, but the manuscript did not have page numbers.

We apologize for these typos. We have carefully double-checked our manuscript to correct all typos. All changes resulting from this proofreading are highlighted in yellow in the revised manuscript.

Concerning the first bullet point, we confirmed that the sentence was right. In study 3, we compared SemNet changes between individuals who solved themselves the riddle (solver group) and those who did not but had the solution before computing the second SemNet (solution group).

As Reviewer #2 soundly requested, we pasted all changes made in the manuscript directly after our responses to the reviewers and ensured that page numbers refer to the revised manuscript.

Reviewer #3

R3 General comments to the Author

I appreciate the effort that authors have put in to address different issues pointed out by the reviewers. Overall, I think this revised manuscript is much stronger and makes clearer claims about the hypothesized constructs. The control experiment (Study 3) is particularly interesting as it provides evidence that solvers and non-solvers who are given the solution are likely showing different types of associations to solution-relevant concepts.

My remaining comments are very minor.

We thank the Reviewer for their positive evaluations of our revised manuscript and Study 3.

R3 Comment #1

Not sure if this would be possible, but it would be interesting to report for each riddle, exactly which "remote" associations were rated higher (or brought closer) when the riddle was solved vs. when the solution was simply presented (in Study 3). This might help make the claims a bit more clearer as I do think the paper is a bit dense with all the metrics. For instance, it is very hard to grasp the key two-way interaction between metric x impact rating and metric x semantic distance and examples (or figures) would really help elucidate this pattern a lot better.

Overall, I think examples could be really useful throughout the paper. For example, when you discuss weight, efficiency, etc., these metrics seem very abstract and it would be good to ground them in some way using examples of words via tables or figures.

We apologize for the lack of clarity concerning our results. We now provide in our revised manuscript:

- Additional figures to make the results easier to understand, in particular for the three-way interaction effects (**Figure 3** and **4**, p. 20-21) and another one summarizing our findings (**Figure 5**, p. 22; see Reviewer #2 comment 3).
- Examples showing word pairs with opposite *impact rating* and *semantic distance* variables (i.e., solution-relevant but semantically close, or solution-irrelevant but semantically distant; see Reviewer #2 comment 1).

R3 Comment #2

Table S8 should be moved to the main text - this is an overall summary of the key findings that would be helpful for the reader. I might even add a verbal description of what the effect means as an additional column for each study.

We moved **Table S8** to the main manuscript (called now **Table 1**) when summarizing our results in the discussion section (p. 23). As suggested by Reviewer #3, we added a description of our results to help readers understand our conclusions.

		Solution-based restructuring (Δ Metric x IR)	Remoteness-based restructuring (Δ Metric x SD)	
Exp. 1	PS	W, Eff, EV	Eff	 •Strengthening/centralizing high-IR edges/nodes is positively related to PS. •Strengthening high-SD edges is positively related to PS.
	APS	W, Eff	W, Eff	 •Strengthening high-IR or high-SD edges is positively related to APS.
Exp. 2	PS	CC	W, Eff	 •Interconnecting high-IR nodes is positively related to PS. •Strengthening high-SD edges is positively related to PS.
Exp. 1+2	IPS	-	W, CC, EV	 •Strengthening/interconnecting/centralizing high-SD edges/nodes is positively related to IPS.
Exp. 3	PS	-	W	 •Strengthening high-SD edges is positively related to PS.
	Solution	W, Eff	-	 •Strengthening high-IR edges is positively related to giving the solution •These effects for high-SD and high-IR edges were significantly opposite.

Table 1. Summary of the results from the three studies. For each analysis, we report the dependent variables (PS: naive problem-solving, APS: analogous problem-solving, IPS: insight problem-solving) and the metrics (W: *weight*, Eff: *efficiency*, CC: *clustering coefficient*, EV: *eigenvector centrality*) for which the interaction effect of Δ Metric and *impact rating* (*IR*, solution-based restructuring) or Δ Metric and *semantic distance* (*SD*, remoteness-based restructuring) was positive and significant.

R3 Comment #3

There are several typos in the ms - hopefully this can be corrected in proofing.

We apologize for these typos. We have carefully double-checked our manuscript to correct all typos. All changes resulting from this proof-reading are highlighted in yellow in the revised manuscript.

27th Mar 24

Dear Dr Bieth,

Your manuscript titled "Changes in semantic memory structure support successful problem-solving and analogical transfer" has now been seen by our reviewers, whose comments appear below. In light of their advice I am delighted to say that we are happy, in principle, to publish a suitably revised version in Communications Psychology under the open access CC BY license (Creative Commons Attribution v4.0 International License).

We therefore invite you to revise your paper one last time to address the remaining concerns of our reviewers and a list of editorial requests. At the same time we ask that you edit your manuscript to comply with our format requirements and to maximise the accessibility and therefore the impact of your work.

EDITORIAL REQUESTS:

SUBMISSION INFORMATION:

OPEN ACCESS:

Communications Psychology is a fully open access journal. Articles are made freely accessible on publication under a CC BY license (Creative Commons Attribution 4.0 International License). This license allows maximum dissemination and re-use of open access materials and is preferred by many research funding bodies.

For further information about article processing charges, open access funding, and advice and support from Nature Research, please visit <https://www.nature.com/commspsychol/article-processing-charges>

At acceptance, you will be provided with instructions for completing this CC BY license on behalf of all authors. This grants us the necessary permissions to publish your paper. Additionally, you will be asked to declare that all required third party permissions have been obtained, and to provide billing information in order to pay the article-processing charge (APC).

* TRANSPARENT PEER REVIEW: Communications Psychology uses a transparent peer review system.

On author request, confidential information and data can be removed from the published reviewer reports and rebuttal letters prior to publication. If you are concerned about the release of confidential data, please let us know specifically what information you would like to have removed. Please note that we cannot incorporate redactions for any other reasons.

* CODE AVAILABILITY: All Communications Psychology manuscripts must include a section titled "Code Availability" at the end of the methods section. We require that the custom analysis code supporting your conclusions is made available in a publicly accessible repository at this stage; please choose a repository that generates a digital object identifier (DOI) for the code; the link to the repository and the DOI must be included in the Code Availability statement. Publication as Supplementary Information will not suffice.

* DATA AVAILABILITY:

[link redacted]

Best regards,

Marike

Marike Schiffer, PhD
Chief Editor
Communications Psychology

REVIEWERS' COMMENTS:

Reviewer #2 (Remarks to the Author):

My points have been more or less sufficiently answered (points #1 and #2 less exactly so). I am grateful to the authors for their effort to incorporate my suggestions into their discussion and enhancing the readability of the manuscript. My gratitude also extends to the reviewer who took the

time to directly insert the changes made to the manuscript in their response.

However, there remains a minor yet significant issue I'd like to address. It concerns the definition of "restructuring," the main concept of this manuscript, for which the authors refer to Mednick (1962) and Schilling (2004).

1) The paper by Mednick (1962) does not explicitly mention the term "restructuring" as a standalone concept, rendering it an unsuitable reference for this context, and it should be removed.

2) The interpretation of "restructuring" by Schilling (2004) diverges significantly from its traditional (Gestalt related) understanding within the framework of Representational Change Theory, as outlined by Ohlsson (1984), Knoblich et al., 1999; Öllinger et al., 2014 and others. Schilling's definition is broader, considering any recognition of similarities between previously unrelated concepts as constituting restructuring, without necessitating these concepts to be directly relevant to solving the problem at hand. While this broader definition aligns more closely with the findings of the authors, who did not observe a three-way interaction among semantic distance, solution relevance (impact rating), and change metric, it is crucial for clarity's sake to inform the readers that a more lenient interpretation of restructuring is being used. Without this clarification, readers familiar with the Representational Change Theory expect to see a three-way interaction (semantic distance * solution relevance [impact rating] * change metric) and likely find the results insufficient for what the paper aims to show.

Reviewer #3 (Remarks to the Author):

I appreciate the authors effort to address the remaining comments, especially R2's interpretative comments, which I agree are important. I have no further comments.

Reviewer 2

R2 General comments to the Author

My points have been more or less sufficiently answered (points #1 and #2 less exactly so). I am grateful to the authors for their effort to incorporate my suggestions into their discussion and enhancing the readability of the manuscript. My gratitude also extends to the reviewer who took the time to directly insert the changes made to the manuscript in their response.

However, there remains a minor yet significant issue I'd like to address. It concerns the definition of "restructuring," the main concept of this manuscript, for which the authors refer to Mednick (1962) and Schilling (2004).

R2 Comment #1

The paper by Mednick (1962) does not explicitly mention the term "restructuring" as a standalone concept, rendering it an unsuitable reference for this context, and it should be removed.

We agree with Reviewer #2 that the paper by Mednick does not explicitly mention the term "restructuring". Hence, we removed Mednick's citation p.3 (ligne 64).

R2 Comment #2

*The interpretation of "restructuring" by Schilling (2004) diverges significantly from its traditional (Gestalt related) understanding within the framework of Representational Change Theory, as outlined by Ohlsson (1984), Knoblich et al., 1999; Öllinger et al., 2014 and others. Schilling's definition is broader, considering any recognition of similarities between previously unrelated concepts as constituting restructuring, without necessitating these concepts to be directly relevant to solving the problem at hand. While this broader definition aligns more closely with the findings of the authors, who did not observe a three-way interaction among semantic distance, solution relevance (impact rating), and change metric, it is crucial for clarity's sake to inform the readers that a more lenient interpretation of restructuring is being used. Without this clarification, readers familiar with the Representational Change Theory expect to see a three-way interaction (semantic distance * solution relevance [impact rating] * change metric) and likely find the results insufficient for what the paper aims to show.*

Thanks for this comment. To clarify this point, we now: i) explicitly mention the Representational Change Theory and the definition of restructuring for the tenants of this theory, ii) how Schilling theory extends the restructuring concept more broadly to include any atypical associations without necessitating the concepts to be relevant to solving the problem.

We changed the manuscript as follows (p.3 and p.4):

“According to the Representational Change Theory¹⁰, solving such problems involves restructuring the initial problem mental representational space^{5,9}, which presumably entails combining elements related to the problem in a new way. In theory, restructuring allows one to change perspective, reframe the problem, or escape its implicitly imposed constraints¹¹, leading to creative associations^{6,9}.”

[...]

“In a theoretical paper, Schilling proposed that insight solving occurs as a result of a recombination of the SemNet associations through the creation of new or unexpected links between remote nodes in semantic memory³⁵. This hypothesis echoes the concept of restructuring developed by the tenants of the Representational Change Theory¹⁰, but extends it more broadly to the formation of any atypical association without necessitating the concepts to be directly relevant to solving the problem at hand. This idea aligns with current theories on the dynamic nature of semantic memory⁵⁷, but only scarce empirical studies tested it (but see⁵⁸).”

Reviewer 3

R3 General comments to the Author

I appreciate the authors effort to address the remaining comments, especially R2's interpretative comments, which I agree are important. I have no further comments.

We appreciate the Reviewer's positive feedback.